# An 85-year record of glacier change and refined projections for Kennicott and Root Glaciers, Alaska

Albin Wells [1] ✉, Brandon S. Tober [1], Sarah F. Child[2], David R. Rounce [1], Michael G. Loso[3], Chad P. Hults[4], Martin Truffer [5,6], John W. Holt[7,8] & Michael S. Christoffersen[5,9,10]

Long-term historical records of glacier mass change are key to advancing understanding of glaciers' response to climate change and improving predictions of their future. Here, we use historical aerial photographs and new bed topography measurements to provide an 85-year record of glacier change on Kennicott and Root Glaciers in Alaska. At the glacier terminus, little change is observed in the two decades prior to 1957, followed by ongoing and accelerating mass loss with dynamically driven spatial variability. Glacier projections, constrained by these mass loss estimates, predict that Kennicott Glacier will lose $38 \pm 14\%$ to $63 \pm 18\%$ of its mass by 2100, relative to 2000, and Root Glacier will lose $38 \pm 11\%$ to $58 \pm 12\%$, depending on the emissions scenario. These results differ by up to 22% from similar predictions made by projections calibrated from the past two decades of glacier change only. This highlights the importance of long-term glacier mass-loss records that help us better project far-reaching consequences of climate change related to sea level rise, water resources, natural hazards, climate, and culture.

The recent ubiquity of satellite observations in glacier monitoring has enabled two decades of spatially-distributed mass loss estimates, with the largest changes occurring in Alaska (e.g., 2000–2019)[1]. The widespread thinning of Alaskan glaciers has impacts on water resources and sea level rise, with critical implications for ecosystems and human society[2–6]. Still, the models for estimating future glacier change are shrouded with uncertainty due to the relatively short time periods of these continuous observations (two decades) in comparison with glacier response times (decades to centuries)[7].

Regional and global glacier evolution models are currently calibrated with mass change observations since the 2000s[8–10] or older records that are only available for a small subset of monitored

glaciers[11–14]. While recent advances in glacier observations are providing unprecedented insight into quantifying present-day changes[15], all glacier evolution models currently suffer from over-parameterization issues (i.e., a lack of sufficient observations to constrain model parameters) and nonlinear feedbacks that encumber historical reconstructions and future estimates. Fortunately, long-term mass balance data–which include glaciological (in situ) observations and geodetic mass balance from digital elevation model (DEM) differencing–provide valuable constraints to overcome these challenges and improve glacier projections[16,17].

Historical aerial photographs combined with structure-from-motion photogrammetry provide unique opportunities to extend

[1]Department of Civil and Environmental Engineering, Carnegie Mellon University, Pittsburgh, PA, USA. [2]Cooperative Institute for Research in Environmental Sciences, University of Colorado, Boulder, CO, USA. [3]Wrangell-St. Elias National Park and Preserve, Copper Center, AK, USA. [4]National Park Service, Alaska Regional Office, Anchorage, AK, USA. [5]Geophysical Institute, University of Alaska Fairbanks, Fairbanks, AK, USA. [6]Department of Physics, University of Alaska Fairbanks, Fairbanks, AK, USA. [7]Lunar and Planetary Laboratory, University of Arizona, Tucson, AZ, USA. [8]Department of Geosciences, University of Arizona, Tucson, AZ, USA. [9]Department of Geosciences, University of Alaska Fairbanks, Fairbanks, AK, USA. [10]School of Earth and Atmospheric Sciences, Georgia Institute of Technology, Atlanta, GA, USA. ✉e-mail: awwells@cmu.edu

records of DEMs and orthophotos of glaciers well before the satellite era. Such historical records have been used to obtain velocity records[18–20] before Landsat (1972) and reconnaissance satellite missions[21,22], and to quantify glacier mass change in Europe[23,24], western North America[25–27], and Antarctica[19,20]. Few studies assess changes in Alaska (Juneau Icefield[28]; Gulkana, Wolverine, and Lemon Creek glaciers[29]). Historical elevation contour maps from the 1950s and 1960s have also been used to quantify long-term glacier changes over larger regions in Alaska, although these contain large uncertainties (-15–45 m) such that uncertainty in 40 + years mass balance exceeds 20% of the elevation change signal[30,31]. Further studies have assessed glacier changes since the 1970s by processing film from the Corona and Hexagon military satellite mission[32–35]. Unfortunately, long-term measurements of Alaskan glaciers from historical film imagery remain largely untapped due to a lack of digitized scenes.

In this study, we leverage a suite of historical and modern datasets (Supplementary Table 1) to quantify changes since 1938 for two of Alaska's most accessible large valley glaciers–Kennicott Glacier and Root Glacier–located adjacent to the Kennecott Mines National Historic Landmark within Wrangell-St. Elias National Park and Preserve (Fig. 1). We produce several new datasets, including high-resolution DEMs from film in 1938, 1957, and 1978, as well as new surface velocity data from 1957–1962 from orthophotos. We complement these data with DEMs from 2004, 2012, and 2023 to obtain spatially distributed changes in elevation and glacier dynamics over time. We demonstrate the value of leveraging geodetic mass balance from historical DEMs to calibrate a glacier evolution model. We simulate projections through 2100 for various emissions scenarios, which reveal considerable changes in future estimates compared to model calibration using only present-day mass change. Not only do we present one of the oldest spatially-distributed records of multi-temporal elevation change for a glacier in Alaska, but, to our knowledge, we also produce the oldest spatially-distributed surface velocity map for a mountain glacier.

## Results and discussion

### Historical record reveals periods of equilibrium and accelerated mass loss

We assess changes in glacier geometry, thinning, and associated mass loss across various extents of Kennicott and Root Glaciers, including a specific focus on the terminus of Kennicott Glacier (below Root Glacier) where DEM coverage extends back to 1938 (Fig. 2g). Elevation change calculations reveal Kennicott and Root Glaciers were roughly in a state of equilibrium from 1938 to 1957, with mass loss accelerating thereafter (Fig. 2). Specifically, the lowest -5 km of Kennicott Glacier was balanced to slightly positive (0.62 ± 0.92 m yr⁻¹) from 1938 to 1957 (Supplementary Table 2). The greater ablation area thinned by

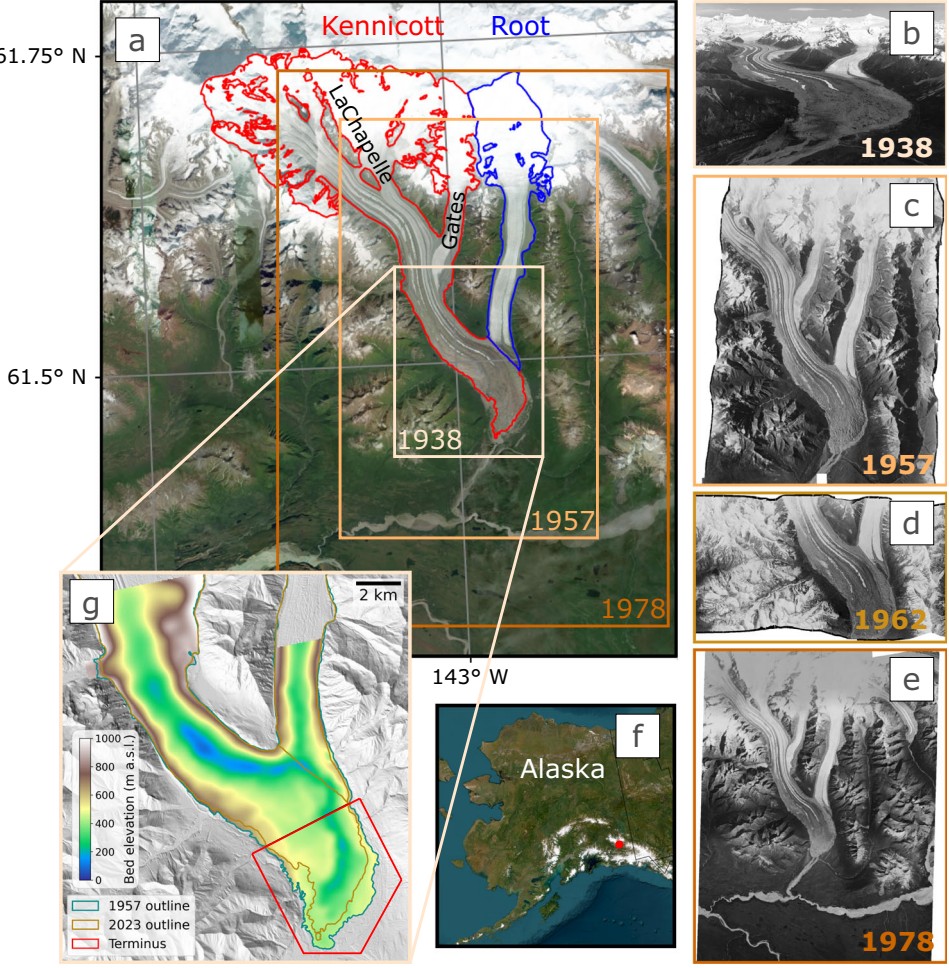

**Fig. 1 | Overview of historical datasets of Kennicott and Root Glaciers in Alaska.**
**a** Map of Kennicott Glacier and Root Glacier showing approximate historical DEM extents and the 2023 glacier outlines, plotted atop the Esri World Imagery basemap. The background map is the intellectual property of Esri and is used herein under license. Copyright © 2025 Esri and its licensors, all rights reserved. **b** Sample aerial photograph from the 1938 dataset (courtesy of

the Bradford Washburn Collection, University of Alaska Fairbanks). **c–e** Orthophotos derived from historical imagery for 1957, 1962, and 1978, respectively. **f** The study area location in Alaska. **g** Kennicott and Root Glacier bed topography shown atop the interferometric synthetic aperture radar (IFSAR) 2012 hillshade with 1957 and 2023 glacier outlines.

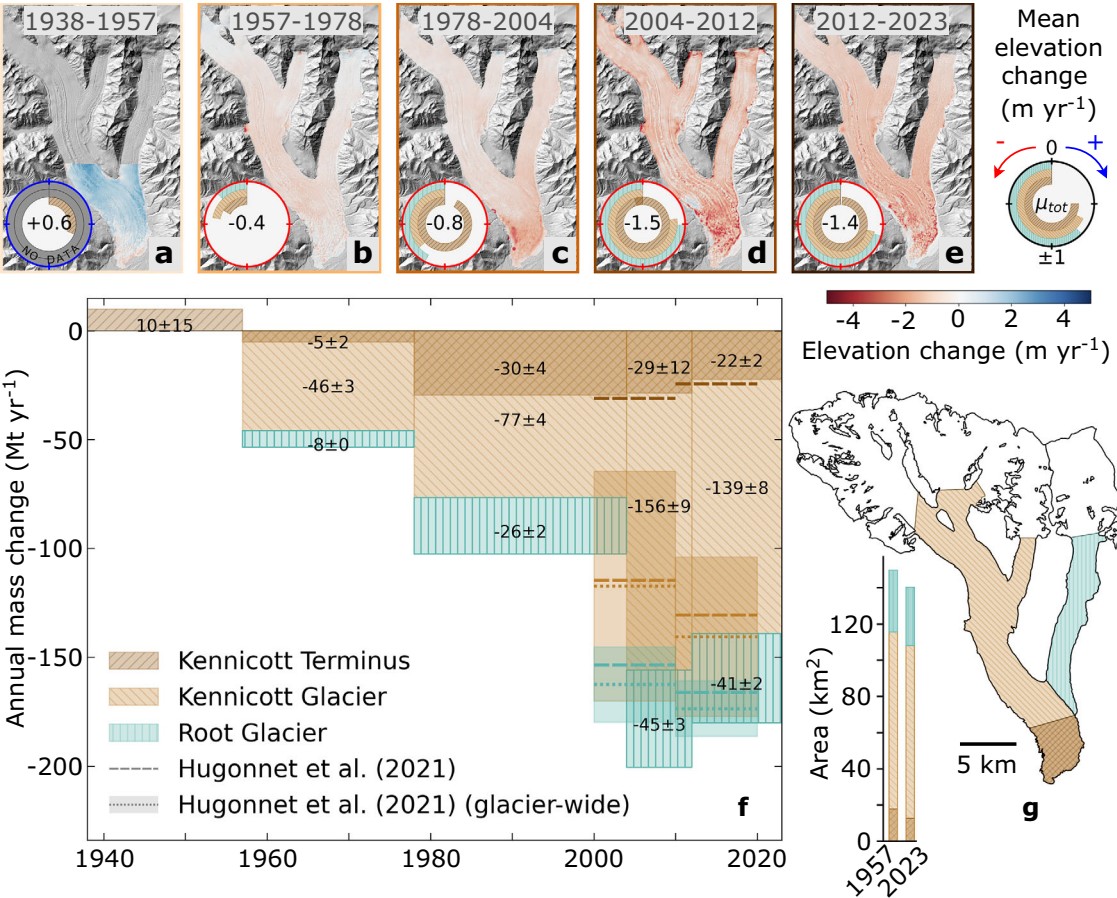

**Fig. 2 | Multi-decadal changes on Kennicott and Root Glaciers. a–e** Spatially-distributed elevation change rates from 1938–2023 overlaying the IFSAR DEM hillshade. Discs show the mean annual elevation change for the Kennicott Glacier terminus, Kennicott Glacier ablation area (labeled as "Kennicott Glacier"), and Root Glacier ablation area (labeled as "Root Glacier"). The value in the center of each disc represents the mean annual elevation change of the total measured area (Kennicott and Root Glaciers). **f, g** Annual mass changes (**f**) of the three areas (**g**) are shown in multi-decadal periods aligning with (**a–e**). The Hugonnet et al. (2021)[1] mass change is shown for both the area covered by the historical DEMs and glacier-wide. The area covered by historical DEMs post-1957 represents over 95% of glacier-wide mass loss (Supplementary Note 1). Large uncertainties exist in glacier-wide mass change from Hugonnet et al. (2021)[1].

0.44 ± 0.02 m yr$^{-1}$ on Kennicott Glacier from 1957–1978, which increased to 0.74 ± 0.03 m yr$^{-1}$ from 1978–2004 and to 1.43 ± 0.06 m yr$^{-1}$ from 2012–2023. A similar trend exists on Root Glacier, which thinned by 0.25 ± 0.02 m yr$^{-1}$ from 1957–1978, 0.85 ± 0.05 m yr$^{-1}$ from 1978–2004, and 1.41 ± 0.09 m yr$^{-1}$ from 2012–2023. Mass loss over this 66-year period corresponds to a 5.3 km$^2$ (30%) decrease in Kennicott Glacier's terminus area from 1957–2023 (Fig. 1g). Notably, the western side of the terminus had particularly thin ice in 2024 (< 30 m) and–given current thinning rates in this area (-3 m yr$^{-1}$; Fig. 2f)–will likely be ice-free in the next decade (Supplementary Fig. 1).

The historical elevation change record reveals roughly two decades of widespread thinning from 1957–1978 followed by anomalous terminus thinning on Kennicott Glacier relative to the rest of the glacier (Fig. 2). The rate of thinning at the terminus of Kennicott Glacier increased nearly six-fold (0.33 to 1.84 m yr$^{-1}$) from 1957–1978 to 1978–2004, compared to a relatively modest 16% increase across the rest of the ablation zone during the same period (0.46 to 0.53 m yr$^{-1}$). Mass loss at the terminus thus increased from accounting for less than 12% of total mass loss prior to 1978 to nearly 40% in the following decades (Fig. 2f). On Root Glacier–which has notably less debris cover than Kennicott Glacier (Fig. 1a)–the spatially-distributed thinning pattern is relatively consistent even as the magnitude of thinning increases over time (Fig. 2a–e). In contrast to Kennicott Glacier, thinning rates at the terminus of Root Glacier are similar to those up-glacier,

suggesting that glacier dynamics and debris-cover are critical drivers of spatial thinning patterns on Kennicott Glacier.

Since the turn of the 21st century (2004–2023), terminus thinning has remained steady while thinning across the rest of the ablation area has roughly doubled on Kennicott Glacier, compared to the previous two decades (1978–2004; Fig. 2f). Although thinning in both areas is slightly greater for the years 2004–2012 than 2012–2023, the multi-decadal trend of accelerated thinning from 1957 to 2023 is evidence of a continued change in climate. Our results show that relatively modest mass loss throughout the second half of the 20th century has been exacerbated by ongoing climate change. Kennicott and Root Glaciers have thus lost a combined 180 ± 10 Mt yr$^{-1}$ from 2012–2023 over the area covered by our DEMs, which represents over 95% of glacier-wide mass loss (Supplementary Note 1).

### Glacier slowdown drives present-day retreat on debris-covered terminus

Ice thickness distributions (1938–2023) derived from differencing DEMs and glacier bed topography, in combination with long-term velocity changes (1960–2017), reveal nearly six decades of changes in ice flux divergence and gravitational driving stresses on Kennicott and Root Glaciers. Both velocity (1960–2017; Fig. 3a) and driving stress (1938–2023; Fig. 3e) have decreased with thinning, with the largest changes occurring in the latter halves of the respective observational periods. In 2023, the lowest 1 km of Kennicott Glacier had virtually no

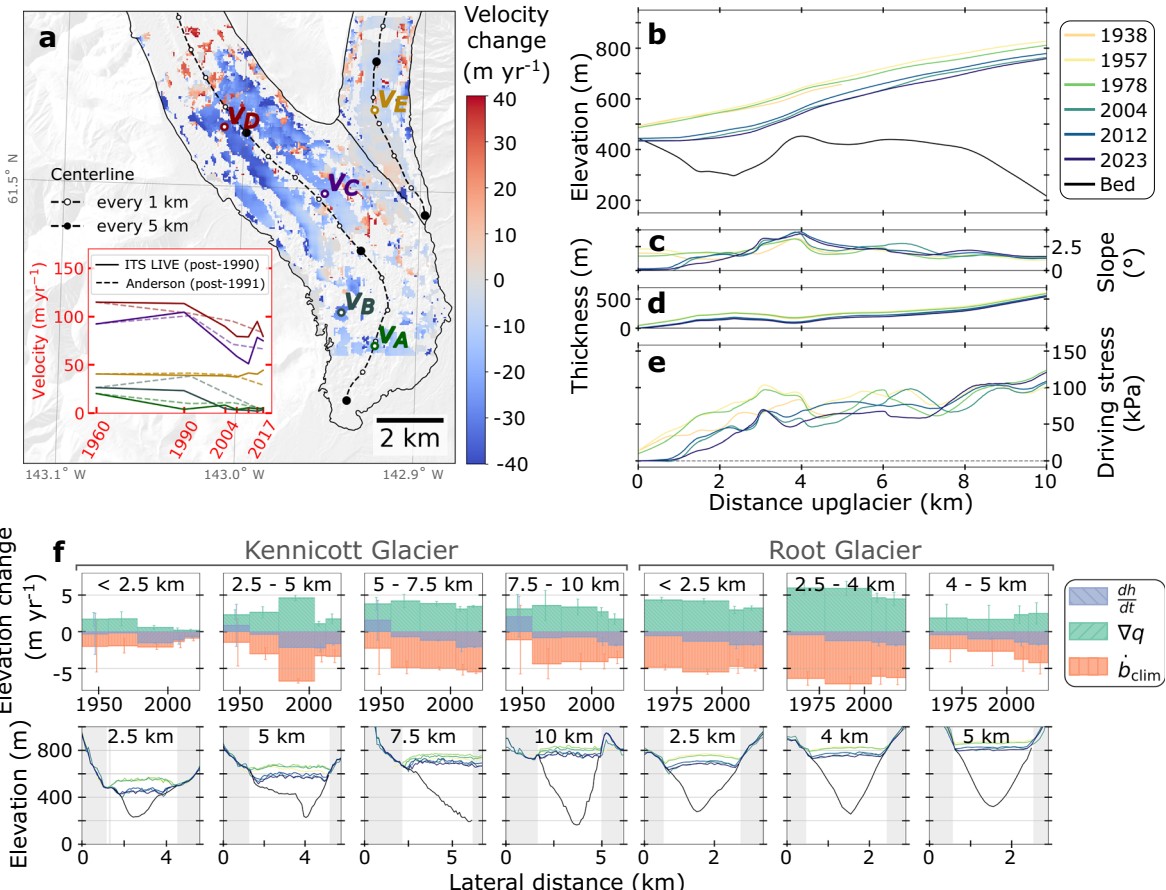

**Fig. 3 | Dynamical changes over the lower ablation area of Kennicott and Root Glaciers. a** Long-term change in velocity from 1960 to 2017. The 1960 velocity map is a result of feature-tracking using PyCorr with orthophotos from 1957 and 1962, which effectively tracked bare ice with largely unchanged features over the 5-year span (Supplementary Fig. 3). The data lies within the 1957 glacier outline and overlays the IFSAR DEM hillshade. Velocity magnitudes from this study (1960) and from ITS_LIVE and Anderson et al. (2021b)[36] (post-1990) are shown at five points on Kennicott and Root Glaciers. **b**–**e** Kennicott Glacier centerline (**b**) surface and bed

elevation, (**c**) surface slope, (**d**) ice thickness, and (**e**) driving stress for 1938, 1957, 1978, 2004, 2012, and 2023. The distance up-glacier corresponds to the centerline in (**a**), which begins at the 2023 glacier terminus. **f** Evolution of glacier surface elevation change ($\frac{dh}{dt}$), flux divergence ($\nabla \cdot q$), and climatic mass balance ($\dot{b}_{clim}$) as well as flux gate cross-sections on Kennicott (1938–2023) and Root (1957–2023) Glaciers for a total of seven flux zones denoted by the longitudinal distance from the terminus at the top of each panel. Bar whiskers represent uncertainty for each value (Methods).

driving stress (< 2.5 kPa) and was near stagnation (5 m yr⁻¹ velocity) due to a decline in slope and ice thickness: a noteworthy decrease from 45 kPa in 1957 (Fig. 3e) and 20 m yr⁻¹ in 1960 (Fig. 3a point $v_A$). The near-zero driving stress across the lowest 1 km of the glacier indicates that the glacier is no longer experiencing sufficient gravitational force to drive ice flow, such that the ice in these regions has effectively decoupled from the dynamic processes of the glacier, marking the onset of terminus stagnation and wastage.

A reduction in driving stress of ~30% (~30 kPa) is observed throughout the lowest 4 km of Kennicott Glacier since 1978, despite a slight surface steepening 2.5-4 km up-glacier. This reduction in driving stress corresponds with a shelf feature in the bed that extends across the center and western side of the glacier, which ice must flow over or around (Fig. 3b and Supplementary Fig. 1). Below this feature, decreased ice velocity (from ~26 m yr⁻¹ in 1960 to < 5 m yr⁻¹ in 2017) and corresponding decrease in ice flux due to a rising equilibrium-line altitude leads to drastic thinning, which has led to substantial reductions in driving stress into the lowest 4 km of Kennicott Glacier. In turn, stagnation prompts further complexities that either reduce (e.g., debris thickening, reduction in ice cliffs) or enhance (e.g., supraglacial pond expansion, increased surface relief) melt[36]. Above 4 km, driving stress shows little change over time; however, surface velocity has decreased up to 30% from 1960–2017 (Fig. 3a point $v_C$ and $v_D$). On Root

Glacier, no substantial dynamical changes have occurred since 1960 (Fig. 3a point $v_E$): velocity remains ~40 m yr⁻¹ and driving stress is largely unchanged (Supplementary Fig. 2).

For a glacier in equilibrium, the flux of ice into the ablation area offsets the climatic mass balance (i.e., the total of ablation, accumulation, and refreezing). We partition thinning signals into ice flux divergence and climatic mass balance to assess the relative contributions of flux divergence and enhanced melt on thinning rates across the lower ablation areas of Kennicott and Root Glaciers. For four flux zones on the lower 10 km of Kennicott Glacier, thinning increases over the 21st century (Fig. 3f). Thinning rates 7.5–10 km up-glacier more than doubled from 1978–2004 to 2012–2023, despite the climatic mass balance becoming slightly less negative, due to the flux divergence (emergence) halving over the same period. A similar trend is observed lower on the glacier: flux divergence decreased 2.5–5 km up-glacier by ~–2.9 m yr⁻¹ from 1978–2004 to 2012–2023, and reduced from 1.7 ± 0.7 m yr⁻¹ to 0.2 ± 0.2 m yr⁻¹ from 1938–1978 to 2004–2023 in the lowest 2.5 km. The historical time-series thus shows drastic changes in flux divergence on the Kennicott Glacier terminus since 1938, displaying a decadal-scale feedback where up-glacier thinning leads to reduced ice influx down-glacier regardless of surface melt at the terminus.

The temporal variations in elevation change, flux divergence, and climatic mass balance are considerably different on Root Glacier

(Fig. 3f). Flux divergence is constant over time 4–5 km up-glacier (–2.2 m yr$^{-1}$) and decreases marginally 2.5–4 km up-glacier (from 5.9 ± 0.8 to 4.5 ± 0.8 m yr$^{-1}$) and at the lowest 2.5 km (from 4.5 ± 0.4 to 3.2 ± 0.3 m yr$^{-1}$). Near the terminus (below 2.5 km up-glacier; with full debris cover), Root Glacier displays a similar but subtler pattern as Kennicott Glacier, where decreasing fluxes drive elevation changes amid relatively unchanging climatic mass balance. Above this zone (2.5–5 km up-glacier; primarily clean ice), uncertainties in flux divergence and climatic mass balance complicate the attribution of elevation changes, although the consistent velocity over time suggests the mechanism for thinning over the clean ice on Root Glacier is driven by surface melt processes rather than a change in glacier dynamics.

### Debris-cover evolution contributes to terminus wastage

Orthophotos display the reduction in clean ice extent over time, with the debris cover extent migrating up-valley by ~3 km from 1957–2023 on Kennicott Glacier (Fig. 2a and Supplementary Fig. 4). While the stagnant ice under relatively thick debris (>10 cm) has the largest melt rate on Kennicott Glacier due to ice cliffs and ponds[37], historical data suggest that the newly debris-covered areas (where the debris layer is initially thin and enhances melt) shift the region of greatest surface melt up-glacier. In turn, this leads to reduced ice fluxes down-glacier in subsequent decades. The location and timing of this debris cover evolution coincide with the timing of velocity slowdowns (Fig. 3a). In 1957, the reach of clean-ice bands on Kennicott Glacier was around 5 km up-glacier. Below this region, the glacier shows a consistent decrease in velocity over time, reducing ice flux into the terminus and subsequently contributing to substantial thinning 2.5–5 km up-glacier from 1978–2012 (Fig. 3f). As the reach of clean-ice bands migrated to ~8 km up-glacier in 2023, the symptoms of these changes are similarly evident over time: around 7.5–10 km up-glacier, decreasing velocities after 1990 preceded substantial thinning 5–7.5 km up-glacier post-2012. Ultimately, terminus stagnation and wastage leading to the formation of proglacial lakes (Supplementary Fig. 4) can increase thinning through a reduction in compressional flow at the terminus and frontal ablation[38], but we do not observe evidence of these mechanisms for enhanced thinning on Kennicott Glacier.

### Past glacier observations constrain future mass loss

Leveraging historical data for calibrating glacier evolution models is a considerable advance compared to prior studies that only use more recent glacier-wide mass balance data (e.g., 2000–2019)[1], hereafter referred to as "modern" data. Calibration with historical data from 1940–2004 (the data from DEMs produced with historical film imagery) ensures that the model aligns with long-term mass change, including being initialized at a period of relative equilibrium. The long-term record thus enables the model to be in a well-constrained present-day dynamical state suited for projecting changes through 2100. Using monthly air temperature and precipitation data from the European Center for Medium-Range Weather Forecasts (ECMWF) Reanalysis v5 (ERA5)[39], we simulate glacier mass balance and dynamics with the Python Glacier Evolution Model (PyGEM)[10]. PyGEM estimates melt using a temperature-index model that includes accounting for sub-debris melt[40], accumulation using temperature thresholds, and refreezing using mean annual air temperature. Simultaneously, it updates the glacier geometry annually using a flowline model based on the shallow-ice approximation[19]. Note that, when calibrating the model parameters with historical data, the glacier mass at the start of the 21st century is consistent with current multi-model ice thickness estimates[41] (Fig. 4a, b). In contrast, the model calibrated with only modern data overestimates mass loss from 1940 to 2023 by 32% and 54% on Kennicott and Root Glaciers, respectively, relative to multi-model ice thickness estimates (Fig. 4a, b).

Future glacier change was simulated from 2000–2100 using an ensemble of 12 general circulation models (GCMs) and four shared

socioeconomic pathways (SSPs) (Methods). In 2100, the model predicts Kennicott Glacier will lose 38 ± 14 to 63 ± 18% of its mass (21 ± 8 to 35 ± 10 Gt), relative to 2000, and Root Glacier will lose 38 ± 11 to 58 ± 12% (5.1 ± 1.5 to 7.7 ± 1.6 Gt), depending on the SSP scenario (Fig. 4). For both Kennicott and Root Glaciers, the model calibrated with historical data shows notably less mass loss in the first half of the 21st century compared to the model calibrated with modern data, followed by accelerated mass loss after 2050 (Fig. 4c, d). While the decrease in the rate of mass loss in the first half of the century is less pronounced for Kennicott Glacier (Fig. 4e) compared to Root Glacier (Fig. 4f), the difference in mass loss between the models with historical and modern calibrations prior to 2050 exceeds the GCM variability. By 2100, GCM spread plays a large role in model projections as the remaining glacier mass falls within GCM spread for both Kennicott and Root Glaciers (Fig. 4c, d boxplots), although significant differences are reported for particular GCM realizations ($p < 0.05$ in 2100 for all SSPs on both glaciers using a Wilcoxon signed rank-sum test).

From 2000–2100, the historically-calibrated model shows an average of 22% less mass loss on Kennicott Glacier compared to the model calibrated only with modern data, depending on the SSP scenario, which is a substantial reduction in projected mass loss. This is primarily due to differences in model parameters that alter the sensitivity of the climatic mass balance to future forcing, especially at the start of the 21st century (Fig. 4e). On Root Glacier, the mean difference in mass loss between the models calibrated with historical and modern data is 7%, which is an increase in the predicted mass loss since 2000. When the model is initialized in 1940, instead of 2000, similar results are found, despite the mass in 2000 being considerably different, especially for Root Glacier (Supplementary Fig. 5). Regardless of whether the historically-calibrated model predicts increased or decreased mass by 2100 compared to the model calibrated with modern data, both Kennicott and Root Glaciers show a more negative rate of mass loss at the end of the 21st century, implying a more dramatic disequilibrium between the glacier and the climate than previously projected (Fig. 4e, f). Both glaciers are thus expected to continue losing mass at a rapid rate beyond 2100.

In summary, observations over an ~85-year period on Kennicott and Root Glaciers show a clear shift from a likely near-equilibrium state prior to 1957, to ongoing and accelerating mass loss throughout the late 20th and into the 21st century. No clear evidence of mass loss is observed before 1957, while substantial mass loss occurs after 1978. Stark differences in the timing and pattern of thinning on the terminus of Kennicott and Root Glaciers emphasize the impact of debris cover on surface mass balance and ice fluxes. Under heavy debris on Kennicott Glacier, the timing of thinning aligns with reduced surface velocities, while thinning on Root Glacier corresponds to changes in the surface mass balance. Projections from a glacier evolution model that utilizes these data predict accelerating rates of mass loss through at least 2050 before leveling out (on Kennicott Glacier) or becoming less negative (on Root Glacier) by the end of the century. While the detailed pattern of mass loss varies between the two glaciers, these improved projections ultimately suggest that 38 to 60% of Kennicott and Root Glaciers' total mass in 2000 will be lost by 2100, depending on the emissions scenario. Our results reinforce observations of long-term accelerated mass loss and rising equilibrium-line altitudes across regions of Alaska[28,29,31,42,43] and project this acceleration to continue through much of the 21st century–decades later than previous studies suggest[10]. Given that Kennicott and Root Glaciers are experiencing thinning rates similar to (albeit slightly higher than) other large glaciers (>50 km²) in the Wrangell Mountains[1], we hypothesize that these projections are indicative of expected changes across the broader region.

As archived Cold War-era documents continue to become declassified, troves of Alaskan historical images are becoming publicly available. Once digitally scanned, we have shown that such images provide unprecedented insights into mechanisms of long-term glacier

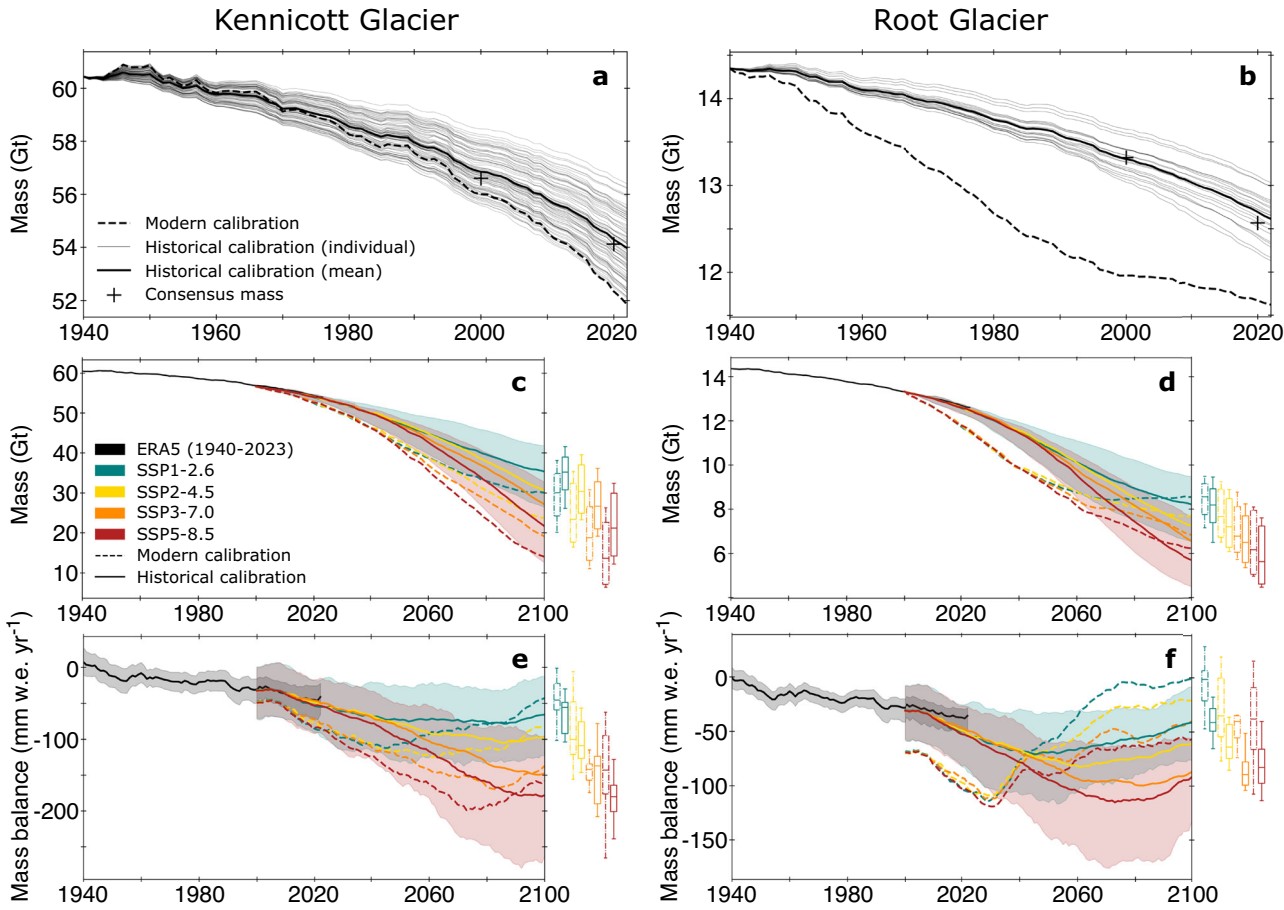

**Fig. 4 | Modeled glacier mass reconstructions and projections of Kennicott and Root Glaciers. a, b** Kennicott (**a**) and Root (**b**) glacier mass from 1940–2023 using PyGEM with ERA5 reanalysis data. Results show all individual runs and the mean of optimal parameter sets that align with multi-decadal mass balance dating back to 1938, and the model calibrated only with modern data from 2000–2019 following a Markov chain Monte Carlo approach (Rounce et al., 2020). Current multi-model "consensus" mass estimates[41] are shown in 2000 and adjusted to 2020 using mass change rates from Hugonnet et al. (2021)[1]. **c, d** Kennicott (**c**) and Root (**d**) glacier annual mass from 1940–2023 (using mean values from **a, b**) and from 2000–2100 using different shared socioeconomic pathways (SSPs). The mean of 12 general circulation models (GCMs) for each SSP is plotted with a thick line. Results are

shown using the optimal parameter set using historical data (1940–2020) and only modern data (2000–2019)[1] following a Markov Chain Monte Carlo approach[10]. **e, f** Kennicott (**e**) and Root (**f**) glacier mass balance from 1940–2023 and from 2000–2100 for different SSPs. Mass balance is smoothed using an 11-year moving average. Shaded regions represent the minimum and maximum GCM realization for the SSP scenario, and the distribution of GCM realizations in 2100 for both the historical and modern calibration is displayed to the right of each plot, where box-plot edges represent data quartiles and the whiskers show the most extreme point 1.5 times the interquartile range (**c**–**f**). Changes in mass and mass balance between historical and modern model calibrations in 2100 are statistically significant ($p < 0.05$) for all SSPs on both glaciers (**c**–**f**).

mass change and provide valuable data that improve projections of glacier change in both the near and distant future. Such analyses not only enhance our understanding of glaciers' response to climate change but also better prepare society for the adverse impacts accompanying glacier mass loss. Thus, by leveraging historical data, we can better plan for future changes, including developing adaptation and mitigation strategies for the corresponding impacts on sea-level rise, ecosystems, water resources, and the communities that rely on these glaciers.

## Methods

All datasets used or produced in this study (Supplementary Table 1) are described below. A brief overview of previous work on Kennicott and Root Glaciers is provided in Supplementary Note 2.

### Historical DEMs

**Historical imagery.** DEMs were generated from digitized film negatives acquired in 1957 and 1978. The 1957 DEM was produced from 32 aerial photographs covering ~929 km², which were acquired during a joint U.S. Geological Survey and U.S. Air Force effort to map Alaska[44].

The 1978 DEM was generated from 23 aerial photographs covering ~1830 km², which were acquired as part of the NASA AMES High-Altitude Aerial Photography program[45]. Five photographs collected by the U.S. Air Force in 1962 (U.S. Air Force, 1963) were also processed to derive 1957–1962 velocities.

**Historical DEM generation.** Image preprocessing for historical imagery ensures that aerial photographs are suitable for stereo-parallax calculations (> 60% image overlap) and are standardized to fixed dimensions, which is required to extract precise elevation information from the stereo view. Structure-from-motion photogrammetric processing was carried out in Agisoft Metashape v1.8.2 (Agisoft, 2022). Internal camera geometry data was generated based on the known camera focal length and by establishing the principal point through measurements of the left-right and top-bottom distances between fiducial marks. By manually identifying the fiducial mark locations (Supplementary Fig. 6), the images were corrected for any inconsistencies in dimension caused by the digitization process. All images were resampled to fit the same geometry as the scanning resolution of 1000 dpi. These corrected digital images were subsequently used for

DEM generation (Supplementary Fig. 7). Camera calibration was performed with the Metashape default Brown-Conrady model to account for lens distortion, principal point offsets, and focal length variations. Given the detailed reporting on camera focal lengths for each air photo set, a single bundle adjustment was sufficient in minimizing reprojection errors. Focal length, principal point coordinates, and radial and tangential distortion coefficients were intrinsically estimated during bundle adjustment. After camera calibration, tie points were identified from internally aligned images and spurious points were manually removed.

Ground control points (GCPs) were selected using prominent stable-terrain features identifiable in both aerial imagery and a reference DEM or hillshade, such as mountain peaks, ridges, crags, and buildings. The GCPs were evenly distributed throughout the study area and included both high and low elevation values (Supplementary Fig. 8). Multi-decadal changes in GCP positions due to isostatic adjustment or plate tectonics were assumed to be negligible and well within the range of historical DEM uncertainties[46].

The GCPs and tie points were used to generate a dense point cloud and subsequent DEMs and orthomosaics for 1957, 1962, and 1978 (Fig. 1c–e). DEMs were assigned resolutions of 4.5, 5.3, and 12.7 m for the 1957, 1962, and 1978 imagery, respectively, based on aerial photograph quality, overlap, altitude, and scale. Snow cover, poor contrast, and limited coverage encumbered reconstructions of the accumulation area, which constrained the final DEMs to the ablation areas of Kennicott and Root Glaciers (Fig. 2a–e). Additional processing settings and input parameters are available (Supplementary Note 3 and Supplementary Table 3).

**1938 "Washburn" DEM.** A DEM from 1938 was produced by the National Park Service using oblique aerial photographs collected by Bradford Washburn, covering ~30 km$^2$ of the terminus of Kennicott Glacier from a camera with a lens that had a 304.8 mm focal length[47]. The DEM was derived from 600 dpi scans of the original (20.3 × 25.4 cm) negatives and processed following the methods outlined above. Due to the nature of these images, limited ground control was available: nine manually-selected ground-control points were identified. The point cloud was cleaned by manually removing clearly aberrant points and applying a denoising filter to remove all points with >1 m absolute error relative to a plane fit through the 30 closest-neighbor points. Considering the quality of the film, the gridded 1938 DEM was rasterized and exported with a conservative resolution of 10 m.

**Additional DEMs**
**ASTER DEM.** A 2004 DEM was derived from ASTER imagery using the MMASTER workflow[48] and had an acquisition date of 4 May 2004 and 100 m resolution. Over the study site, this DEM contained the most complete spatial coverage. Holes in the DEM were filled after co-registration using adjusted long-term (1978–2012) thinning rates[49] so that the mean difference between the filled pixels and the surrounding data gap pixels was zero (Supplementary Fig. 9). Given the ASTER DEM was acquired in late spring and the interferometric synthetic aperture radar (IFSAR) DEM was acquired in late summer, we applied a vertical offset based on monthly DEM predictions between May and September[1] to reduce the impact of seasonal elevation change in the thinning signal. The effective acquisition date of the ASTER DEM is thus September 2004.

**IFSAR DEM.** As part of the Alaska Mapping Initiative from 2010–2012, DEMs were produced over large parts of Alaska from IFSAR (an airborne, imaging radar) at native 5 m resolution (https://doi.org/10.5066/P9C064CO). For Kennicott and Root Glaciers, a 2012 DEM was generated with imagery acquired in late summer (between 14 August and 8 September). In this study, the IFSAR DEM was used as a reference elevation for all other DEM products due to the expansive off-glacier coverage and reported sub-meter accuracy, even on steep terrain[50]. We converted the IFSAR DEM vertical datum to WGS84 ellipsoid heights using GEOID2009 grid files (NOAA, https://vdatum.noaa.gov/download.php) to be consistent with the other datasets in this study.

**National Park Service DEM.** The National Park Service produced a 0.32 m resolution DEM, covering 426 km$^2$, from 1250 images collected on 1 August 2023 over the ablation areas of Kennicott and Root Glaciers using structure-from-motion photogrammetry of aerial photographs[51,52]. In this study, the DEM was downsampled to match the 5 m resolution of the reference IFSAR DEM.

**DEM co-registration**
All DEMs were co-registered to ensure consistent spatial alignment based on stable terrain. We used the National Land Cover Database (NLCD) land cover product[53] and Randolph Glacier Inventory v7.0 (RGI 7.0 Consortium, 2023) glacier outline to mask unstable terrain. A 200 m buffer was added to the glacier outline to account for past glacier extent. The NLCD 'shrubland' class was used for stable terrain as other classes such as 'barren' included unstable rock glaciers, debris cover, and loose, steep terrain. This filter thus facilitated co-registration over an expansive stable area of the study region.

DEM co-registration was implemented using the open-source DEM alignment utility within demcoreg[54], which follows from Nuth and Kääb[55]. In addition to a stable terrain mask, all terrain with a slope below 0.1 degrees or above 20 degrees was also masked out. The IFSAR product was used as the reference DEM.

The 1938 DEM initially contained biased elevation values due to the processing of oblique imagery. We removed this bias by fitting a first-order polynomial from differenced elevations over stable terrain (with a 2-sigma filter) using the SAGA polynomial regression function in QGIS and subtracting this from the DEM.

**Glacier outline digitization**
Glacier outlines were used for area, volume, and mass change calculations. Outlines were produced for three distinct extents (Kennicott Glacier, Root Glacier, and the terminus of Kennicott Glacier) and two time periods (1957 and 2023). Given the broad temporal span of DEMs, elevation change maps were used to delineate glaciated and off-glacier terrain. Orthophotos and surface slope were used to delineate the glacier margin when the elevation change signal was unclear. Accumulation areas were excluded during outline digitization due to poor image contrast on snow and a lack of suitable ground-control points which encumbered DEM generation in these regions: above the uppermost icefalls on Root Glacier, the Gates and LaChapelle tributaries, and the edge of the aerial photograph extent on the main branch of Kennicott Glacier (Fig. 1a). Nonetheless, our digitized outlines (Fig. 2g) represent over 95% of all mass loss (~40% of total glacier area) on Kennicott and Root Glaciers from 2000–2019[1].

Calculations that use glacier area require contemporaneous glacier boundaries with regards to the relevant dataset (e.g., Eqs. 1 and 3). Since no notable changes in extent were observed from 1938–1978, and no notable changes in extent were observed from 2012–2023, the 1957 glacier extent was used for the years 1938, 1957, and 1978 and the 2023 glacier extent was used for 2012 and 2023. For data between 1978 and 2012, an average extent was applied[56].

**Volume, elevation, and mass change**
Volume, elevation, and mass change rates were calculated from DEM differencing and digitized glacier outlines for 1938–1957, 1957–1978, 1978–2004, 2004–2012, and 2012–2023. The volume change rate is the sum of the elevation change rate for all pixels within the glacier

boundary:

$$\dot{V} = \sum_{p=1}^{N_p} (\dot{h}_p) \cdot A_p \qquad (1)$$

where $\dot{h}_p$ is the elevation change rate for a pixel $p$ in a glacier boundary containing a total of $N_p$ pixels, and a pixel area, $A_p$. The mass change rate is converted from volume change rate:

$$\dot{M} = \dot{V} \cdot \rho_{ice} \qquad (2)$$

using 900 kg m$^{-3}$ as the density of ice ($\rho_{ice}$). A density of 900 kg m$^{-3}$ was preferred to the commonly-used geodetic volume-to-mass conversion 850 kg m$^{-3}$ because our study is constricted to ablation areas over long temporal spans[57], and thus is not affected by snow or firn. The mean elevation change rate is the volume change rate divided by the area:

$$\dot{H} = \frac{\dot{V}}{\bar{A}} \qquad (3)$$

where the $\bar{A}$ is the average area during the time span when the volume change rate $\dot{V}$ is evaluated.

## Ice thickness and bed topography

Two ice-penetrating radar campaigns were conducted to obtain detailed observations of ice thickness and bed topography. In March 2024, the Groundhog radar[58] was towed 3 linear-km by ski across the lower Root Glacier in a common-offset configuration, with the transmitter and receiver separated each at the center of a 20 m half-wavelength dipole antenna. During May 2024, Blue System Integration's Air Ice-Penetrating Radar (AirIPR)–an adaptation of the radar described in Mingo and Flowers (2010)[59]–was utilized in an extensive helicopter-borne radar survey to acquire 530 linear-km of sounding data across both Kennicott and Root Glaciers. The AirIPR system was slung from a platform ~25 m below the helicopter and ~30 m above the glacier surface, with the transmitter and receiver each at the center of an 8 m half-wavelength V-dipole antenna[60].

Groundhog and AirIPR data were similarly processed by removing the mean trace, applying a bandpass filter in the fast-time direction, and performing Stolt migration[61] assuming a wavespeed in temperate ice of 169 m μs$^{-1}$ (relative permittivity of 3.15[62]). Groundhog data were also time shifted to account for the time delay between signal transmission and the start of each record that was triggered by the airwave. For both radar systems, point measurements of the two-way travel time delay in ice were provided by manually digitizing ('picking') glacier bed returns (as well as the surface return for AirIPR profiles) using the Radar Analysis Graphical Utility (RAGU)[63]. In total, 215 linear km of glacier bed returns were picked. Two-way travel time delays were converted to ice thickness values, assuming a wave speed of 169 m μs$^{-1}$. Groundhog data were corrected for the geometrical separation of the transmit and receive antennas[64]. Bed elevations were subsequently derived by subtracting the ice thicknesses from the 2023 National Park Service DEM. A crossover analysis[65] of nearly 300 radar profile intersections was used to quantify the accuracy of radar-derived measurements. This analysis reveals a combined median and interquartile range (IQR) disagreement in ice thickness of 6 and 16 m, respectively (Supplementary Table 4).

Bed elevation measurements were interpolated through Gaussian process regression, following the methodology presented in Tober et al. (2023)[66]. The historical glacier outline, radar-derived bed elevations, and proglacial lake depths acquired through bathymetry surveys[67] were all used as observed data points upon which the Gaussian process model was trained. Bed elevation predictions were made across a uniform 50 m grid, spanning from the up-glacier extent

of radar-derived bed elevations to the terminus (Supplementary Fig. 1). This interpolation method yields a probability distribution over beds with variable roughness, from which the mean was taken with uncertainty represented by twice the marginal standard deviation.

## Historical and modern velocity

The orthophotos from 1957 and 1962 were used to estimate a mean annual velocity near the Kennicott Glacier terminus over this time period. Feature-tracking was implemented with PyCorr[68] using orthophotos downsampled to 30 m with an increment size of 2 pixels, half source of 6 pixels, and half target of 20 pixels. Noise was filtered from the spatially-distributed velocity product by removing 2-sigma outliers and pixels flowing uphill (Supplementary Fig. 3).

Velocity products from the NASA ITS_LIVE program[69] were also used to assess changes in glacier dynamics. ITS_LIVE velocities are derived from Landsat imagery between 1985–2017; 2018 was excluded due to an episode of flow acceleration on the western half of the glacier near the terminus[70]. In this study, we used annual and multi-year averaged ITS_LIVE mosaicked velocity coincident with our elevation change calculations.

## Flux divergence and climatic mass balance

Spatially-distributed elevation change, ice thickness, and velocity were used to calculate the flux divergence and climatic mass balance for distinct zones near the glacier terminus (Fig. 3f). Ice thickness was calculated for each time period by subtracting the bed elevation from the DEM. The flux divergence for a given zone was calculated by differencing the volume of ice entering and leaving the zone:

$$\nabla \cdot \mathbf{q}_{zone} = \frac{(uH)_{in} - (uH)_{out}}{a_{zone}} \qquad (4)$$

where $u$ is depth-averaged velocity (assumed to be 0.9 times the surface velocity), $H$ is ice thickness, and $a_{zone}$ is the area of the flux zone. Assuming negligible basal melt, the flux divergence was summed with the surface elevation change to derive the climatic mass balance:

$$\frac{\dot{b}_{clim, zone}}{\rho} = \dot{H}_{zone} + \nabla \cdot \mathbf{q}_{zone} \qquad (5)$$

where $\dot{H}_{zone}$ is the mean elevation change rate in the flux zone.

The uncertainty of this method is dominated by flux zone placement; as such, sensitivity to flux zone placement is assessed by shifting flux zones from their original placements and taking the mean value of nine zones over an 800 m distance. We took the range of values as the uncertainty. This flux-gate method emulates previous literature[36] but is robust to specific flux gate placement, thus reducing bias and accurately capturing uncertainty in climatic mass balance results. This method ensures that localized uncertainties in velocity, ice thickness, and surface elevation change are consistently accounted for regardless of particular flux gate positions. We further incorporated uncertainty in the historical velocity product based on the proportion of linearly-interpolated data gaps for a given cross-section, and added this as a percentage uncertainty of flux divergence. We assumed no bias in ice thickness data due to the density of observations (Supplementary Fig. 1) and crossover accuracy of ice-penetrating radar results presented in this study (Supplementary Table 4), and we assumed no bias in elevation change products due to co-registration results (Supplementary Fig. 10).

## Driving stress

Gravitational driving stress was calculated along the centerline of Kennicott Glacier for each DEM (Fig. 3a). Individual surface slopes ($\alpha$) and ice thickness ($H$) were derived from each DEM and the bed

topography raster. The driving stress was calculated via:

$$\tau_d = \rho_{\text{ice}} \cdot g \cdot H \cdot sin(\alpha) \tag{6}$$

where $g$ is the acceleration due to gravity. Surface slope was obtained after applying a locally weighted scatterplot smoothing[64]. Derived longitudinal driving stresses were smoothed with a filter length 2–4 times the ice thickness to minimize potential effects of longitudinal stress gradient coupling[71]. Spatially-distributed driving stress (Supplementary Fig. 2) was calculated for each grid cell at 100 m resolution using the slope calculated from DEMs smoothed with a 2-sigma Gaussian filter.

## Uncertainty assessments

**DEM accuracy.** To evaluate the accuracy of each DEM relative to the reference IFSAR DEM, we calculated the median and normalized median absolute deviation (NMAD) of stable terrain, as they are robust to extreme values and represent DEM accuracy well[72]. The median provides an estimate of bias over the DEM spatial extent; the NMAD is an estimate of the absolute error for any pixel. Accuracy metrics are shown before and after co-registration (Supplementary Fig. 10).

**Spatially autocorrelated error.** We assessed the spatial autocorrelation of error for all DEMs by sampling empirical variograms of the elevation differences over stable terrain[73]. We sampled five unique empirical variograms with a sample size of 10,000 using the xDEM Python package (xDEM contributors, 2021)[74]. We subsequently fit a short-range (maximum pixel distance of 500 m) and long-range (no limit) spherical variogram model to the mean empirical variogram. The sill and range were used to assess the scale and extent of spatial autocorrelation in errors, reflecting the variance beyond which points are no longer correlated and the total distance over which points remain correlated, respectively.

**Area uncertainty.** We applied a buffer method[75] with a 50 m buffer (4x and 100x the 1957 and 2023 DEM pixel resolutions, respectively) to capture the uncertainty in glacier outline delineations. To address uncertainties resulting from a temporal mismatch between the glacier outline and a dataset, we used (1) temporally-resolved glacier area extents when possible and added no additional uncertainty, or (2) a mean area with the minimum and maximum area as the uncertainty[56].

**Volume, elevation, and mass change uncertainty.** Volume change uncertainty was calculated from the delineated and buffered glacier outlines. We assumed no systematic error resulting from co-registered DEM elevation change products, since co-registration removed biases, and any local errors in elevation change are negligible over large spatial extents. For analysis of the Kennicott Glacier terminus–a relatively small spatial extent–DEM errors were incorporated into the volume change uncertainty by taking stable terrain elevation change NMAD values as the vertical uncertainty in volume calculations.

Elevation change rate uncertainty was propagated from uncertainties in volume change rate ($\sigma_{\dot{V}}$) and glacier area ($\sigma_A$):

$$\sigma_{\dot{H}} = \dot{H} \cdot \sqrt{\left(\frac{\sigma_A}{A}\right)^2 + \left(\frac{\sigma_{\dot{V}}}{\dot{V}}\right)^2} \tag{7}$$

Mass change rate uncertainty was similarly propagated from uncertainties in volume change ($\sigma_{\dot{V}}$) and ice density ($\sigma_\rho = 50$ kg m$^{-3}$):

$$\sigma_{\dot{M}} = \dot{M} \cdot \sqrt{\left(\frac{\sigma_{\dot{V}}}{\dot{V}}\right)^2 + \left(\frac{\sigma_\rho}{\rho}\right)^2} \tag{8}$$

## Glacier evolution modeling

The past (1940–2023) and future (through 2100) evolution of Kennicott and Root Glaciers was simulated with the Python Glacier Evolution Model (PyGEM)[10].

**Historical glacier evolution reconstruction.** The historical reconstruction requires an estimate of the glacier geometry (mass and extent) from 1940 to initialize the simulation. Elevation change rates from 1957–2012 were therefore used to adjust the ~2000 multi-model ice volume[41] back to 1940. The multi-model product[41] was chosen due to its complete coverage and to facilitate comparisons with previous studies (i.e., Rounce et al., 2023[10]). Volume change in the accumulation areas of Kennicott and Root Glaciers was estimated using the 1957 U.S. Geological Survey topographic maps (Supplementary Fig. 11). Glacier-wide mass change from 1940–1957 was assumed to be negligible compared to the overall glacier mass, given no surface elevation change at the terminus.

PyGEM was run with the ECMWF Reanalysis v5 (ERA5) climate reanalysis data for each glacier from 1940–2023 for over 7500 parameter combinations of the three mass balance model parameters (precipitation bias, temperature bias, and the degree-day factor of snow) based on plausible parameter ranges for Kennicott and Root Glaciers[10]. Only parameter sets that aligned with historic observations from DEM differencing were retained. We calculated the root mean square error between modeled and observed ice thickness change for 10 m elevation bins and three time periods coinciding with DEM difference dates (1957–1978, 1978–2004, 2000–2019). A parameter set was retained if the root mean square error was in the bottom 10% of all parameter combinations and the glacier-wide mass balance was within the uncertainty range of observations (Supplementary Note 4). Ultimately, 1.2% ($n = 89$) and 0.3% ($n = 22$) of the model parameter combinations met this criteria for Kennicott and Root Glaciers, respectively (Supplementary Fig. 12). All retained parameter combinations were used to model the glacier from 1940–2023, from which the mean modeled mass change is used to estimate the historical glacier evolution (Fig. 4a, b). By using the long-term mass balance record for calibration, these parameter combinations are an inherent improvement over the model calibration with only data from the 21st century, as they capture climate sensitivity, which may impact calibration over shorter timescales (e.g., 2000–2019). The difference in elevation change rates between observations and the models calibrated with historical and modern data shows reduced bias when using historical data for calibration (Supplementary Fig. 13), and show a long-term rise in equilibrium-line altitude of ~250 m and ~150 on Kennicott and Root Glaciers, respectively, from 1940–2022 (Supplementary Fig. 14).

**Future glacier evolution.** Future glacier change was simulated from 2000–2100 for both glaciers using the calibrated sets of model parameters and an ensemble of 12 general circulation models (GCMs) from the Coupled Model Intercomparison Project Phase 6 (CMIP6) and four shared socioeconomic policies (SSPs), such that each model run had 48 individual realizations. The four SSPs (SSP1-2.6, SSP2-4.5, SSP3-7.0, and SSP4-8.5) correspond to approximately 1.8, 2.7, 3.8, and 4.5 °C warming by the end of the 21st century relative to preindustrial levels, respectively. Modeled glacier change using the retained parameter combinations as determined above from historical data was also compared to results using parameters calibrated with only modern data (2000–2019) using a Markov chain Monte Carlo approach[10]. We assessed 'significant' differences in modeled glacier changes using a Wilcoxon signed-rank test ($p < 0.05$) to emphasize differences in model outputs irrespective of GCM spread within an SSP scenario. Future simulations initialized in 1940 were also performed to quantify the uncertainty associated with the initial conditions.

**Model limitations**. On Kennicott and Root Glacier, PyGEM offers significant advancements over other large-scale glacier evolution models through a mass balance scheme that accounts for sub-debris melt using state-of-the-art debris thickness estimates and subsequent melt enhancement factors[40]. However, PyGEM does not explicitly account for the transient state of debris cover over time and the corresponding nonlinear feedback on sub-debris melt. Furthermore, the model does not account for proglacial lake evolution and frontal ablation on Kennicott Glacier, which could increase terminus retreat especially where the glacier bed is overdeepened (Supplementary Fig. 6). While the impact of proglacial lake evolution and frontal ablation can affect model results on decadal timescales[76], this ultimately is glacier specific and will have diminishing impact on projected glacier mass as the glacier retreats towards higher elevations[77]. Other potentially impactful factors on modeling surface melt include ice cliff, supraglacial pond, and surface relief feedbacks[36] or model parameters changing over time[78–80], although incorporating these requires the development of new model parameterizations, which is beyond the scope of this study.

## Data availability

All data used in this work are available with open access, and the source data are provided with this paper. The aerial photographs from 1938 can be obtained from the Bradford Washburn Collection at the University of Alaska Fairbanks, Alaska, and the Polar Regions Collections and Archives. Aerial photograph single frames (used for the 1957, 1962, and 1978 DEM) can be accessed from the U.S. Geological Survey via EarthExplorer (https://doi.org/10.5066/F7610XKM). The IFSAR DEM is available publicly online through the U.S. Geological Survey (https://doi.org/10.5066/P9C064CO). The historical topographical maps can be downloaded from the U.S. Geological Survey national maps service (apps.nationalmap.gov/downloader/). Climate data used for modeling are available online at the Copernicus Climate Change Service (C3S) Climate Data Store (cds.climate.copernicus.eu/). The 2000–2019 elevation change data are available from Hugonnet et al. (2021) (https://www.sedoo.fr/theia-publication-products/?uuid=c428c5b9-df8f-4f86-9b75-e04c778e29b9). All DEMs and orthophotos, velocity products, glacier outlines, radar-derived ice thickness and bed topography, and model outputs produced and used in this study are available from Zenodo (Wells et al., 2025), "Data products from an 85-year record of glacier change and impacts on future projections for Kennicott and Root Glaciers, Alaska", Zenodo, v2, doi: 10.5281/zenodo.14783252; https://zenodo.org/records/14783252).

## Code availability

The software used in this manuscript includes Python, QGIS, MicMac, and Agisoft Metashape Pro. Information regarding installation and system requirements are all available with these software packages. MicMac is available at https://github.com/micmacIGN/micmac. The PyGEM model code and documentation is available on GitHub (github.com/PyGEM-Community/PyGEM). Code used to produce results and plot figures is available on Zenodo (Wells, Albin (2025), "albinwwells/past_and_future_mb", Zenodo, v1;doi: 10.5281/zenodo.15786494; https://zenodo.org/records/15786494).

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

## Acknowledgements

A.W. and D.R.R. were supported by the National Aeronautics and Space Administration (NASA) awards 80NSSC20K1296 and 80NSSC24K1530. B.S.T. and D.R.R. were supported by the Department of the Interior (DOI) award P22AC02208. S.F.C. was supported by the U.S. National Science Foundation (NSF) grant ANS2135018. B.S.T., M.G.L., M.T., J.W.H., and M.S.C. were supported in conducting ice-penetrating radar surveys by the National Park Service Federal Lands Planning Program, DOI award P24AC00284. M.G.L. was supported in part by the National Park Service Inventory and Monitoring Program. This work began during the International Summer School in Glaciology in McCarthy, Alaska. We acknowledge the summer school faculty (especially Regine Hock) and the support from NASA, the International Association of Cryospheric Sciences, and the International Glaciological Society. We thank Lia Lajoie for the 1938 DEM, Romain Hugonnet for ASTER DEMs, Tim Smith (U.S. Geological Survey EROS) for information on the 1962 aerial photographs, and William Armstrong and Mark Fahnestock for discussions regarding velocities.

## Author contributions

Concept and design were done by S.F.C., M.G.L., D.R.R., A.W., and B.S.T.; data acquisition by S.F.C., M.G.L., C.P.H., D.R.R., B.S.T., and A.W.; bed data acquisition and processing by B.S.T., M.G.L., M.T., J.W.H., and M.S.C.; programming by A.W., B.S.T., and S.F.C.; formal analysis by A.W. and B.S.T.; writing and visualization by A.W.; funding acquisition and project administration by S.F.C., D.R.R., and M.G.L.; A.W., B.S.T., D.R.R., M.G.L., C.P.H., M.T., J.W.H., and M.S. contributed to editing and review.

## Competing interests

The authors declare no competing interests.
