## [Transparent Peer Review file · Nature Communications]

An 85-year record of glacier change and refined projections for Kennicott and Root Glaciers, Alaska

Corresponding Author: Mr Albin Wells

Version 0:

Reviewer comments:

Reviewer #1

(Remarks to the Author)

The manuscript presents long-term elevation change and velocity reconstructions of two Alaskan glaciers, Kennicott and Root glaciers. As of today, these are likely the longest complete reconstructions of mountain glacier changes, providing accurate elevation changes and velocities with high spatial resolution over the entire ablation zone. The authors identified a comprehensive list of resources (imagery, IFSAR) and calculated accurate elevations and velocities with reliable error estimates.

However, the manuscript falls short in analyzing the new record. The two sections on the observations (L86-L191) are limited to describing the changes (thinning, flux divergence, climatic mass balance, etc.) in different zones along the glaciers without providing insight into the processes driving these changes. For example, L154-L167 is simply a list of numbers that could be converted into a table.

The authors attribute the terminus waste to a decrease in upstream velocity caused by an upstream migration of a debris-free band (L193-L206). The idea of linking mass loss to changes in debris cover is not new. Anderson, L.S., et al., 2021 presented a detailed analysis to identify feedback mechanisms impacting the response of Kennicott Glacier to climate forcing, "Four additional feedbacks relating glacier thinning to melt changes are evident: the debris feedback (negative), the ice cliff feedback (negative), the pond feedback (positive), and the relief feedback (positive)." In this manuscript, very little (if any) new evidence is offered to confirm or elaborate on these feedback mechanisms. As shown here, by 2023, Kennicott Glacier calves into a large lake, and its terminus region became flat and likely floating (Extended Data Fig. 4, Fig. 3b). This result suggests complex interactions between frontal ablation, undercutting, ice cliff melt, and other factors. Several recent studies examined the dynamic response of lake-calving glaciers to climate warming (e.g., Sato et al., 2021). Moreover, Anderson, B. et al., 2021, examined the response of debris-covered and lake-calving glaciers to climate change, a scenario directly relevant to this study.

The new reconstruction offers an opportunity to study these complex processes on the Kennicott/Rott glacier system and thus lay the groundwork for improving the projection of future changes. However, the work presented here only includes an inventory of changes and thus doesn't provide a suitable foundation for constraining future mass loss.

References:

Anderson, L. S., Armstrong, W. H., Anderson, R. S., Scherler, D. & Petersen, E. The Causes of Debris-Covered Glacier Thinning: Evidence for the Importance of Ice Dynamics From Kennicott Glacier, Alaska. *Front. Earth Sci.* 9, 680995 (2021).

Anderson, B., A.N. Mackintosh, R. Dadić, J. Oerlemans, C. Zammit, A. Doughty, A. Sood, B. Mullan, Modelled response of debris-covered and lake-calving glaciers to climate change, Kā Tiritiri o te Moana/Southern Alps, New Zealand. *Global and Planetary Change*, 205, 103593 (2021) <https://doi.org/10.1016/j.gloplacha.2021.103593> (2021).

Sato, Y., Fujita, K., Inoue, H., Sakai, A. & Karma. Land- to lake-terminating transition triggers dynamic thinning of a Bhutanese glacier. *Cryosphere Discuss.* 2021, 1–21 (2021).

Terminology:

I suggest using accepted terminology:

Instead of film imagery (L78) and analog film (L62), use "film." Instead of historical contour maps (L58), historical topographic maps or historical elevation contour maps are used. A Digital Elevation Model is a 3D representation of a terrain's surface, so "spatially distributed record of digital elevation models (DEMs) (L53)" is a confusing terminology spatially distributed record of elevation, i.e., Digital Elevation Models (DEMs) would work better (L53).

In Extended Data Table 1:

Calling data optical is not informative as it simply means that they were collected using EM waves in the optical domain. Suggested categories include oblique stereo photographs from 1938 (terrestrial or aerial?), ASTER stereo satellite imagery, ice penetrating radar, etc.

Also, I suggest including dates (month, day) in Table 1.

Detailed comments:

L81: Does present-day refer to 2023?

L90: Elevation change is not observed but calculated

L148: Evidence of rising ELA? Perhaps showing ELA on maps?

L176-177: repeat statement

L241-244: Fig. 4 doesn't show a clear acceleration of the rate of mass loss by the end of the 21st century as stated The rate of mass loss is actually decreasing for most models (Fig. 4e-f)

L283-284: High-resolution declassified imagery in Alaska is already available through USGS Earth Explorer. Did the authors explore the coverage of these images? Is there any evidence that additional imagery was collected, which will provide further information about past changes once it becomes available?

L428: this should be average elevation change and not climatic mass balance (see also L485)

Fig. 2: Define the numbers in the middle of the disks.

Fig.4: e and f show annual mass balance (thinning rate), so the unit should be mm w. e. -1

Extended Data Fig. 3: Using positive images instead of negatives would make the interpretation easier.

Extended Data Fig. 4: The 2023 glacier outline shown in other figures (Fig. 1; extended Data Fig. 1) and distributed through Zenodo differs from the outline of the glacier shown here. Also, the 2023 image clearly indicates that the terminus region is now detached from the rest of the glacier. A discussion about how this change impacts glacier behavior and projections would be necessary. Finally, what is the blue "haze" of the 2023 image?

Extended Data Fig. 5: This figure is interesting but unclear about its relevance to the manuscript. It shows a much longer profile than the rest of the figures without a corresponding map and is not explained in the text.

(Remarks on code availability)

Reviewer #2

(Remarks to the Author)

This paper provides a detailed review of changes in the nature of Kennicott and Root Glacier utilizing data spanning the 1938-2023 period. The utilization of data over this span of time and the associated modelling provides a unique look at glacier response to climate over nearly a century. The methods used are appropriate, effective and detailed. The results are constrained by a lack of available data from the accumulation zone. A greater focus on flux divergence at the ELA and reporting the change in ELA elevation would provide greater insight into the accumulation zone changes. It is worth expanding on contrast/comparison to the long term data sets/locations at Taku Glacier and for Eclipse Icefield. Below are comments that are individually minor, but collectively should increase the value to the reader and clarity of the paper.

Specific Comments:

Introduction:

Overparameterization: Either here or later make sure to mention what variables are overparameterized.

Pg. 4 P1: You have a time gap make sure to discuss the 1978-2012 period or an interval within that.

Pg. 4 P3: It is noted that this is evidence of a glacier seeking a new point of equilibrium. This is true for Kennicott Glacier with much greater terminus thinning, However for Root Glacier with similar losses at all elevations this is typical of a

disequilibrium response., although in this case this does not include the accumulation zone, so it could be either.
Pg. 7 P1: More detail-For the ice shelf feature noted, what is the velocity across this feature in 1978 vs 2023? How much has the ELA risen? If there are not specific numbers to report here utilize other regional observations.
Pg. 7 P2: This illustrates importance of noting shifts in the elevation where the ablation area begins-ELA..
Pg. 8 P1: For Clarity-There is not an up valley migration of specific debris it is the upper limit of the surface debris cover that has migrated upglacier.
Pg. 10 Is it worth noting here the shift in equilibrium status of Taku Glacier which fit the pattern above from 1950-2006 as noted by flux near ELA (Pelto, 2008) and after 2010 does not Davies et al (2024)?
<https://tc.copernicus.org/articles/2/147/2008/tc-2-147-2008.html>

A new paper Kindstedt et al (2025) examines changes of the Eclipse Icefield melt events and firn character-“ 2016 to 2023 there has been a 1.67 °C warming of the firn at 14 m depth”. They indicate firn would become temperate allowing increased meltwater percolation without refreeze. Does this have relevance in your flux divergence for the future into the ablation zone?

Kindstedt, I., Winski, D., Stevens, C. M., Skelton, E., Copland, L., Kreutz, K., Mannello, M., Clavette, R., Holmes, J., Albert, M., and Williamson, S. N.: Ongoing firn warming at Eclipse Icefield, Yukon, indicates potential widespread meltwater percolation and retention in firn pack across the St. Elias Range, EGU sphere [preprint], <https://doi.org/10.5194/egusphere-2024-3807>, 2025.

(Remarks on code availability)

I reviewed this code to determine its availability and level of development explanation. Both are excellent. I am not capable of assessing the code effectiveness.

Reviewer #3

(Remarks to the Author)

See report as pdf.

(Remarks on code availability)

See comments in pdf.

Reviewer #4

(Remarks to the Author)

(Remarks on code availability)

Please see comments from co-reviewer.

Version 1:

Reviewer comments:

Reviewer #2

(Remarks to the Author)

The authors provide a detailed and compelling story of the mass loss and velocity changes of Kennicott and Root Glacier utilizing data that spans an 85 year time period. To develop this length of record required utilization of older aerial images with techniques that can be applied elsewhere. Below are several specific comments that I recommend the authors consider adopting. The paper is acceptable for publication whether the authors adopt these suggestions or not.

Specific Comments:

106: When was this notably thin ice observed, 2023? How thick was it in 1957?

164: Worth emphasizing that this reduction in velocity above the shelf is indicative of reduced volume flux which in turn leads to decoupling and stagnation below the shelf.

179: “..regardless of surface melt”. This is not true overall, but can be in a qualified sense. The reduced volume flux is to a large part derived from surface melt. It is true that below the shelf even if surface melt does not increase reduced volume flux will drive thinning.

294: Reword for accuracy “ In summary, observations over an ~85 period..”

372: Are there other Washburn (1941) areas mapped where the same approach could be used?

605: reword for clarity “utilizing the long-term geodetic mass balance record..”

(Remarks on code availability)

The authors provide a detailed record and analysis of mass balance and velocity changes of the Kennicott and Root Glacier's, Alaska spanning an 85-year period. To develop this long of a record required generating DEMs from older aerial photographs. The accelerating mass loss observed matches that of other Alaskan glaciers. I recommend answering or adopting several specific comments below, to clarify and add context. The paper is acceptable for publication regardless of whether the authors address the comments.

Specific Comments:

106: When was this notably thin ice observed, 2023? How thick was it in 1957?

164: Worth emphasizing that this reduction in velocity above the shelf is indicative or reduced volume flux which in turn leads to decoupling and stagnation below the shelf.

179: "...regardless of surface melt". This is not true overall, but can be in a qualified sense. The reduced volume flux is to a large part derived from surface melt. It is true that below the shelf even if surface melt does not increase reduced volume flux will drive thinning.

294: Reword for accuracy "In summary, observations over an ~85 period.."

605: reword for clarity "utilizing the long-term geodetic mass balance record.."

Reviewer #5

(Remarks to the Author)

The authors demonstrate a comprehensive study that integrates historical DEMs, modern satellite data, and ice-penetrating radar observations with climate-driven glacier model projections for Kennicott and Root Glaciers. The writing and figures in this manuscript are excellent. My main concern is that the DEMs carry a significant amount of uncertainty, which is not sufficiently accounted for and clarified in the manuscript. Does the lack of spatial coverage and level of uncertainty truly constrain your model projection?

Detailed comments:

L29 Long-term records do not directly have far reaching consequences. Suggest revising to something like: "This highlights the importance of long-term glacier mass-loss records that help us better project far-reaching consequences of climate change related to sea level rise, water resources, and natural hazards, climate, and culture."

L38 Add and consider citation for Arendt et al. 2002 "Rapid Wastage of Alaska Glaciers and Their Contribution to Rising Sea Level" DOI: 10.1126/science.1072497

L62 The statement "Only one study assesses changes in Alaska" using historical aerial photographs combined with structure-from-motion photogrammetry is not correct. Consider the work by the USGS in: O'Neel et al. 2019 "Reanalysis of the US Geological Survey Benchmark Glaciers: long-term insight into climate forcing of glacier mass balance" doi:10.1017/jog.2019.66

L63 Also consider the excellent early photogrammetric work by Robert Krimmel

Krimmel, R.M., and Rasmussen, L.A., 1986, Using sequential photography to estimate ice velocity at the terminus of Columbia Glacier, Alaska: *Annals of Glaciology*, v. 8, p. 117-123.

Krimmel, R.M., and Sikonia, W.G., 1986, Velocity and surface altitude of the lower Hubbard Glacier, Alaska, August 1978: U.S. Geological Survey Open-File Report 86-549, 20 p.

Krimmel, R.M., 1987, Columbia Glacier, Alaska, photogrammetry data set 1981-1982 and 1984-1985: <https://doi.org/10.3133/ofr87219>

Krimmel, R.M., 1992, Photogrammetric determination of surface altitude, velocity, and calving rate of Columbia Glacier, Alaska, 1983-91: U.S. Geological Survey Open-File Report 92-104, 72 p.

L128 Your study assumes that glacier mass loss captured by the DEMs accounts for 95% of the total mass change of both glaciers (Supplemental Text S1). However, the DEMs primarily cover the terminus and ablation area. The assumption that "thinning in the accumulation area from 1994–2013 converges to zero" at Kennicott and Root Glaciers is not directly substantiated by Larsen et al., 2015. Their study highlights that Alaskan glaciers are predominantly losing mass between 1994 and 2013 across all categories (land, lake-terminating, tidewater), but also points out significant spatial variability. The mean normalized hypsometric curve does converge to zero in their study (Figure 2 a), but also shows significant hypsometric variability for individual glaciers. I'm not sure this information fully supports the assumption for your entire 85 year long study period at these two specific glaciers. Additionally, your geodetic measurements used to constrain the mass loss have very large uncertainty (e.g. 79.8 (m) error for GCPs in 1938, as per Supplementary Table S1). Does the lack of spatial coverage and level of uncertainty truly help constrain your model projection?

L236-239 This is a long sentence that could be broken up, for example: "PyGEM estimates melt using a temperature-index model that accounts for sub-debris melt (Rounce et al. 2021), accumulation using temperature thresholds, and refreezing

using mean annual air temperature. Simultaneously, it updates the glacier geometry annually using a flowline model based on the shallow-ice approximation (Maussion et al. 2019).”

L337 Embedding EXIF information is not a requirement or objective of historical image preprocessing. I suggest stating something like: “Image preprocessing for historical imagery ensures that aerial photographs are standardized to calibrated or fixed dimensions, which is required to extract precise elevation information from stereo view.”

L342 Replace “Image EXIF metadata” with “Internal camera geometry data”.

L344 Replace “east-west and north-south” with “left-right and top-bottom”. Fiducial markers are referenced in the image plane, not cardinal directions.

L345 Fig. S1 Which years do a,b,c,d, belong to? How was a definitive center point for the markers a and d selected in the image pixel grid?

L346 What kind of correction was applied? Did you apply any sort of affine transformation to calibrated values? Was any form of image enhancement attempted to increase dense cloud matches over the snow covered accumulation areas? This warrants at least some form of discussion given the lack of data in the accumulation area and emphasis on the value of historical image-derived DEMs.

L352 What were the final reprojection errors and what level of error was deemed sufficient?

L354 Did you manually or programmatically identify tie points? Did you manually remove spurious tie points?

L369 Suggest naming this ‘1938 “Washburn” DEM’ or reorganizing the sections in to “DEMs from vertical images” and “DEMs from oblique images”.

L420 It would be useful to see a DEM difference map before and after the first-order polynomial fit to assess uncertainty and determine the nature of spatially correlated error that was modeled and removed.

L546 The appropriate citation here is Höhle and Höhle 2009 “Accuracy assessment of digital elevation models by means of robust statistical methods” <https://doi.org/10.1016/j.isprsjprs.2009.02.003>

L548 Extended Data Fig 6 shows random, systematic, and significant spatially correlated error. The Distance and Variance in the variograms should be shown with a log or jump scale on the x-axis to better show spatially correlated error at various distances.

While you calculate the distance and variance of the spatially correlated error, it does not appear these are considered when propagating your uncertainty to volume and mass change estimates. For example, how are the ~200 m variance over ~100 m distance in 1938 taken in to account?

The difference maps in Extended Data Fig 6 are barely visible which does not make them very useful. I suggest either a) giving Original-2004 its own color bar, b) using a log scale for the Elevation difference color map, or c) removing Original entirely. If you choose option c), you can focus this figure on highlighting random, systematic, and spatially correlated error in the final surfaces. Then have a dedicated figure for 1938 comparing the original before and after polynomial surface correction + co-registration, as well as a dedicated figure for 2004 before and after co-registration. Again, my biggest issue is that the uncertainties of these measurements are so large that I don’t see how they act as effective constraints on the glacier evolution model.

L552 and 554 It is odd to mention specific function names here, but not elsewhere. I recommend removing this detail and keeping it conceptual to be consistent with other sections.

If you are using a white background you need to choose a colormap that does not include white. This applies to:

Extended Data Fig. 2

Extended Data Fig. 3d

Extended Data Fig. 6

(Remarks on code availability)

The description of the methods, links to software, and provided code appear sufficient to reproduce the results presented in this study. I did not attempt to install and run the code, but it seems feasible.

Version 2:

Reviewer comments:

Reviewer #5

(Remarks to the Author)

The authors have addressed the concerns raised during initial review. I appreciate their clarification regarding the spread of

values in Figures 4a and 4b, which differs significantly from the model not calibrated with historical data. This strengthens my confidence that the inclusion of these historical data meaningfully constrains and improves the model projections, despite the high levels of DEM uncertainty.

The comprehensive work presented in this study - bridging long-term historical observations with modern data and model calibration - is excellent and I recommend it for publication.

Minor comments:

L31 Delete first "and"

L79 You state that "To our knowledge, we present the first study that uses geodetic mass balance from historical DEMs to calibrate a glacier evolution model." Are you aware of any glacier modeling studies that calibrate against long-term records of glaciological field mass-balance measurements, which in turn were calibrated with historical geodetic mass balance? If so, would this change your statement?

(Remarks on code availability)

The information presented in the study and accompanying code appear sufficient to reproduce the results. I have not tested the code.

We would like to thank both reviewers for their feedback and comments on the manuscript. We appreciate the time and thought that went into them, and have responded to each below. We feel that these changes have undoubtedly strengthened the manuscript overall.

Reviewer #1 (Remarks to the Author):

The manuscript presents long-term elevation change and velocity reconstructions of two Alaskan glaciers, Kennicott and Root glaciers. As of today, these are likely the longest complete reconstructions of mountain glacier changes, providing accurate elevation changes and velocities with high spatial resolution over the entire ablation zone. The authors identified a comprehensive list of resources (imagery, IFSAR) and calculated accurate elevations and velocities with reliable error estimates.

However, the manuscript falls short in analyzing the new record. The two sections on the observations (L86-L191) are limited to describing the changes (thinning, flux divergence, climatic mass balance, etc.) in different zones along the glaciers without providing insight into the processes driving these changes. For example, L154-L167 is simply a list of numbers that could be converted into a table.

Thank you for acknowledging the novelty and value of these records. In the revised version, we have sought to find more of a balance between sharing the detailed values of the diverse observations, while also providing more insight into the processes driving change as noted. We thus present salient details in the text, with the remainder of information in figures (Figs. 2 and 3) or tables (Extended Data Table 2). These passages have been amended to reduce some numbers (especially around the paragraph that was L154-167), thereby providing space to speak of the changes from a more process-oriented perspective.

We attempt to present the results in a logical manner: first we discuss thinning, then dynamic changes, and finally debris evolution. It is difficult to discuss complex processes without a full introduction of all of these topics. As such, we feel it is appropriate to discuss various elements regarding the processes driving changes at different parts of these sections. In addition to more minor line-edits, notable sentence additions/replacements have been made. These can be found in the track-changed manuscript as follows:

~L171: *“In turn, stagnation prompts further complexities that either reduce (e.g., debris thickening, reduction in ice cliffs) or enhance (e.g., supraglacial pond expansion, increased surface relief) melt (Anderson et al., 2021b).”*

~L195: *“The historical time-series thus shows drastic changes in flux divergence on the Kennicott Glacier terminus since 1938, displaying a decadal-scale feedback where up-glacier thinning leads to reduced ice influx down-glacier regardless of surface melt.”*

~L208: *“Near the terminus (below 2.5 km up-glacier; with full debris cover), Root Glacier displays a similar but subtler pattern as Kennicott Glacier, where decreasing fluxes drive elevation changes amid relatively unchanging climatic mass balance. Above this zone*

(2.5-5 km up-glacier; primarily clean-ice), uncertainties in flux divergence and climatic mass balance complicate the attribution of elevation changes, although the consistent velocity over time suggests the mechanism for thinning over the clean ice on Root Glacier is driven by surface melt processes rather than a change in glacier dynamics.”

The authors attribute the terminus waste to a decrease in upstream velocity caused by an upstream migration of a debris-free band (L193-L206). The idea of linking mass loss to changes in debris cover is not new. Anderson, L.S., et al., 2021 presented a detailed analysis to identify feedback mechanisms impacting the response of Kennicott Glacier to climate forcing, "Four additional feedbacks relating glacier thinning to melt changes are evident: the debris feedback (negative), the ice cliff feedback (negative), the pond feedback (positive), and the relief feedback (positive)." In this manuscript, very little (if any) new evidence is offered to confirm or elaborate on these feedback mechanisms. As shown here, by 2023, Kennicott Glacier calves into a large lake, and its terminus region became flat and likely floating (Extended Data Fig. 4, Fig. 3b). This result suggests complex interactions between frontal ablation, undercutting, ice cliff melt, and other factors. Several recent studies examined the dynamic response of lake-calving glaciers to climate warming (e.g., Sato et al., 2021). Moreover, Anderson, B. et al., 2021, examined the response of debris-covered and lake-calving glaciers to climate change, a scenario directly relevant to this study.

Noted. Thank you for referencing those studies, they are certainly interesting and relevant. The novelty in this study lies not in discovering new feedback mechanisms on Kennicott or Root Glacier, but (1) to extend the temporal observational record of changes, (2) present spatially-distributed changes over multi-decadal periods, and (3) to show how these data affect future projections from a glacier evolution model. We do not claim to be making a new discovery with regards to linking mass change and debris cover or glacier dynamics—in fact, we leave the more complex feedbacks untouched as they have been reported in previous work (Anderson et al. 2021a,b). We mention the feedbacks briefly, but focus instead on the novel elements and unique datasets produced/used in the study. Future work would focus on linking and evaluating these more complex feedbacks over longer time spans, and building them into glacier evolution models. Those tasks, unfortunately, are beyond the scope of this current study.

We have added a “Model Limitations” section to discuss some of these complexities in more detail [see comment below]. We’ve added mention to the Anderson, B. et al. (2021) reference in this section, which we feel is appropriate.

We have also added a sentence to the discussion at the end of the debris covered section, *“Ultimately, terminus stagnation and wastage leading to the formation of proglacial lakes (Extended Data Fig. 3) can increase thinning through a reduction in compressional flow at the terminus and frontal ablation (e.g., Sato et al., 2022), but we do not observe evidence of these mechanisms for enhanced thinning on Kennicott Glacier.”*

Regarding the specific references cited, Anderson, B. et al. (2021) is a relevant study that we have added as a reference to the manuscript (see comment below), however, they also assume static debris cover (i.e., unchanging over time) and find that the inclusion of lake-terminating processes does not increase mass loss. They find that this lack of mass loss is comparable to a 10% increase in precipitation, which is the equivalent of a change in the precipitation factor of 0.1, which is very minor. As such, we anticipate that the range of uncertainty in lake calving is likely captured by the overall parameter uncertainty and is unlikely to alter the mass change results in our study, although future studies should investigate this.

Sato et al. (2022) show interesting results for two very unique glaciers: both glaciers have accumulation zones that drop straight into a relatively small ablation area and have velocities ~ 50 m/yr at the terminus. This is very different from Kennicott and Root glaciers, which are both in long valleys that eventually reach the terminus. These glaciers exhibit large (relative to the glacier) calving events that can only take place where the terminus is at least partially floating, which is not the case for Kennicott Glacier (bathymetry data shows ~ 10 - 40 m lake depths at the Kennicott Glacier terminus and radar data shows ~ 50 - 100 m ice thickness. The lake would need to be at least ~ 50 m deep to support any flotation of this ice). As such, Thorthormi and Lugge glaciers in Sato et al. (2022) display characteristics much like a marine-terminating outlet glacier, where large ice fluxes are calving into proglacial lakes. Additionally, the process of evolving from a land- to lake-terminating glacier with significant frontal ablation takes time—even in the unique case of Thorthormi Glacier, it took ~ 30 years to form a full proglacial lake. The proglacial lake at Lugge Glacier was already fully formed in 1985. Similar studies (e.g., Watanabe et al., 2009) show the growth of Imja Lake from small ponds in the 1950s to a true proglacial lake by the 1980s took ~ 30 years to form a lake with calving/frontal ablation having considerable ice loss (and ~ 70 years to its current state). All of these examples are starkly different from Kennicott Glacier, where the proglacial lake is still relatively small: covering only a small fraction of the glacier width (this can be immediately seen if exploring Google Earth) and not causing much calving.

Watanabe et al., 2009: doi.org/10.1080/00291950903368367

Ultimately, these processes take time and are highly variable by glacier. On Kennicott Glacier, current indications show that the proglacial lake is expanding (glacier lake area estimates from Rickman and Rosenkrans, 1997 and National Park Service observations), but still in a state of development, and this expansion is unlikely to be a dominant source of mass loss given the overall size of the glacier and the relatively small size of the lake. Lake area at the terminus of Kennicott has risen steadily from 1995-2018, but the total lake area is still only ~ 1.6 km² and has only grown by ~ 1.2 km² in over 20 years. This only represents a small fraction of the ablation area of Kennicott Glacier, which is >108 km² (108 km² is the 2023 area covered by our DEMs on Kennicott Glacier, which misses some of the upper reaches of the ablation area). As

such, we expect this to play a small role in comparison to mass loss driven by surface melt.
Rickman and Rosenkrans, 1997: <https://doi.org/10.3133/wri964296>

Fig. R1: Kennicott Glacier lake area evolution from 1995-2018. Lake area is quantified from digitized maps from Rickman and Rosenkrans, 1997 and outline produced by the National Park Service.

The new reconstruction offers an opportunity to study these complex processes on the Kennicott/Rott glacier system and thus lay the groundwork for improving the projection of future changes. However, the work presented here only includes an inventory of changes and thus doesn't provide a suitable foundation for constraining future mass loss.

Thank you for your comment. We believe that it is unfair to assert that the study as a whole does not provide a suitable foundation for constraining future mass loss since our results show otherwise. Overall, we use our inventory of changes to calibrate a state-of-the-art large-scale glacier evolution model and assess the impacts on projections. The model uses the best available datasets of debris cover and the effects associated with it. The historical calibration implicitly accounts for processes such as surface relief, ice cliffs, and supraglacial ponds. However, explicit parameterizations of these data do not exist and thus cannot be incorporated into glacier evolution models—hence, we do our best with what currently exists in the modeling community and demonstrate an improvement to the current state of large-scale glacier evolution models. We have added a “Model Limitations” section at the end of the methods to discuss these limitations in more depth:

“On Kennicott and Root Glacier, PyGEM offers significant advancements over other large-scale glacier evolution models through a mass balance scheme that accounts for sub-debris melt using state-of-the-art debris thickness estimates and subsequent melt enhancement factors (Rounce et al., 2021). However, PyGEM does not explicitly account for the transient state of debris cover over time and the corresponding nonlinear feedback on sub-debris melt. Furthermore, the model does not account for proglacial lake evolution and frontal ablation on Kennicott Glacier, which could increase terminus retreat especially where the glacier bed is overdeepened (Fig. S1). While the impact of proglacial lake evolution and

frontal ablation can affect model results on decadal timescales (Anderson, B. et al., 2021), this ultimately is glacier specific and will have diminishing impact on projected glacier mass as the glacier retreats towards higher elevations (e.g., Trüssel et al., 2015). Other potentially impactful factors on modeling surface melt include ice cliff, supraglacial pond, and surface relief feedbacks (Anderson et al., 2021b) or model parameters changing over time (e.g., Huss et al., 2009; Compagno et al., 2022; Ismail et al., 2023), although incorporating these requires the development of new model parameterizations which is beyond the scope of this study.”

References:

Anderson, L. S., Armstrong, W. H., Anderson, R. S., Scherler, D. & Petersen, E. The Causes of Debris-Covered Glacier Thinning: Evidence for the Importance of Ice Dynamics From Kennicott Glacier, Alaska. *Front. Earth Sci.* 9, 680995 (2021).

Anderson, B., A.N. Mackintosh, R. Dadić, J. Oerlemans, C. Zammit, A. Doughty, A. Sood, B. Mullan, Modelled response of debris-covered and lake-calving glaciers to climate change, Kā Tiritiri ote Moana/Southern Alps, New Zealand. *Global and Planetary Change*, 205, 103593 (2021) <https://doi.org/10.1016/j.gloplacha.2021.103593> (2021).

Sato, Y., Fujita, K., Inoue, H., Sakai, A. & Karma. Land- to lake-terminating transition triggers dynamic thinning of a Bhutanese glacier. *Cryosphere Discuss.* 2021, 1–21 (2021).

Terminology:

I suggest using accepted terminology:

Instead of film imagery (L78) and analog film (L62), use "film." Instead of historical contour maps (L58), historical topographic maps or historical elevation contour maps are used. A Digital Elevation Model is a 3D representation of a terrain's surface, so "spatially distributed record of digital elevation models (DEMs) (L53)" is a confusing terminology spatially distributed record of elevation, i.e., Digital Elevation Models (DEMs) would work better (L53).

Noted. We have updated the manuscript to use: “DEMs” (removed ‘spatially-distributed’), “historical elevation contour maps”, and “film” (remove ‘analog’ and ‘imagery’)

In Extended Data Table 1:

Calling data optical is not informative as it simply means that they were collected using EM waves in the optical domain. Suggested categories include oblique stereo photographs from 1938 (terrestrial or aerial?), ASTER stereo satellite imagery, ice penetrating radar, etc.

The table has been updated to reflect this.

Also, I suggest including dates (month, day) in Table 1.
Done.

Detailed comments:

L81: Does present-day refer to 2023?

Yes, this has been changed to make this more clear.

L90: Elevation change is not observed but calculated

This has been fixed.

L148: Evidence of rising ELA? Perhaps showing ELA on maps?

We do not show an ELA on the maps because we do not derive it directly (since it requires end-of-summer imagery) and thus can only infer its location from long-term elevation changes. We make this inference with confidence based on glacier processes and observed glacier thinning over 85 years: a glacier in a state of mass loss will have a rising ELA. While we cannot directly calculate the ELA change, we can estimate it from the model outputs. We have added a reference to the modeled ELA in the methods, “...and show a long-term rise in equilibrium-line altitude of ~250 m and ~150 on Kennicott and Root Glaciers, respectively, from 1940-2022 (Fig. S8).”

“Fig. S8: Kennicott and Root Glacier equilibrium-line altitude from 1940-2022 for the historically-calibrated model. The points represent the equilibrium-line altitude for individual years with a line of best fit showing long-term trends.”

L176-177: repeat statement

Thanks for catching this. This has been removed.

L241-244: Fig. 4 doesn't show a clear acceleration of the rate of mass loss by the end of the 21st century as stated. The rate of mass loss is actually decreasing for most models (Fig. 4e-f)

Noted. We have adjusted the wording to clarify the point that mass balance is more negative in 2100, “Regardless of whether the historically-calibrated model predicts increased or decreased

mass by 2100 compared to the model calibrated with modern data, both Kennicott and Root Glaciers show a more negative rate of mass loss at the end of the 21st century, implying a more dramatic disequilibrium between the glacier and the climate than previously projected (Fig. 4e, f)."

L283-284: High-resolution declassified imagery in Alaska is already available through USGS Earth Explorer. Did the authors explore the coverage of these images? Is there any evidence that additional imagery was collected, which will provide further information about past changes once it becomes available?

Yes, the authors explored all possible imagery. There are USAF aerial photos from 1948, but they are located at the National Archives cold storage and not digitized. The authors also consulted Tim Smith from USGS EROS with respect to all digitized imagery within the USGS's collection, and from 1957 onward, all sets of existing photographs collected with sufficient overlap were processed in this study. With respect to the declassified spy satellite imagery available from the USGS, the only years with cloud-free, high-resolution imagery (thus excluding ARGON and CORONA imagery) are 1976 and 1979. Considering how close in time these acquisitions are to the 1978 set of air photos, we felt the inclusion of the Hexagon imagery would not influence the conclusions made in the study. We are aware that the NGA are in the middle of digitizing the entire spy satellite collection, but to date, that data is not currently available to the public.

L428: this should be average elevation change and not climatic mass balance (see also L485) The formulation on L485 follows from standard convention, it is essentially the continuity equation (as in Cogley et al., 2011). The equation has been updated to specify `b_dot_clim_zone` to align more precisely with the variables in this equation. L428 has been removed since this equation is not actually used in the results and the only climatic mass balance results (for the flux zones in Fig. 3) use the equation from L485.

Fig. 2: Define the numbers in the middle of the disks.

The numbers in the middle of the disks are explicitly stated in the figure caption, "*The value in the center of each disc represents the mean annual elevation change of the total measured area (Kennicott and Root Glaciers)*"

Fig.4: e and f show annual mass balance (thinning rate), so the unit should be mm w. e. -1 The axis labels have been fixed—mass balance is indeed a rate.

Extended Data Fig. 3: Using positive images instead of negatives would make the interpretation easier.

Noted. We have updated the figures showing positive images and adjusted the caption.

Extended Data Fig. 4: The 2023 glacier outline shown in other figures (Fig. 1; extended Data Fig. 1) and distributed through Zenodo differs from the outline of the glacier shown here. Also, the 2023 image clearly indicates that the terminus region is now detached from the rest of the glacier. A discussion about how this change impacts glacier behavior and projections would be necessary. Finally, what is the blue "haze" of the 2023 image?

We apologize for the confusion: both images of Extended Data Fig. 4 show the 1957 outline, which serves as a consistent frame of reference between panels a and b. The figure caption has been amended to make this more clear. The "haze" from the 2023 image appears to be a product of the photogrammetric stitching of many images and the slightly variable lighting conditions between those images, which are a combination of direct sunlight and shadows from clouds. The added discussions pertaining to glacier processes driving thinning throughout the manuscript, as well as the added section on model limitations should address the comments regarding discussion about the terminus detachment [see earlier comments].

Extended Data Fig. 5: This figure is interesting but unclear about its relevance to the manuscript. It shows a much longer profile than the rest of the figures without a corresponding map and is not explained in the text.

This figure has been removed due to its lack of relevance to the main manuscript.

Reviewer #2 (Remarks to the Author):

This paper provides a detailed review of changes in the nature of Kennicott and Root Glacier utilizing data spanning the 1938-2023 period. The utilization of data over this span of time and the associated modelling provides a unique look at glacier response to climate over nearly a century. The methods used are appropriate, effective and detailed. The results are constrained by a lack of available data from the accumulation zone. A greater focus on flux divergence at the ELA and reporting the change in ELA elevation would provide greater insight into the accumulation zone changes. It is worth expanding on contrast/comparison to the long term data sets/locations at Taku Glacier and for Eclipse Icefield. Below are comments that are individually minor, but collectively should increase the value to the reader and clarity of the paper.

Specific Comments:

Introduction:

Overparameterization: Either here or later make sure to mention what variables are overparameterized.

We feel that explaining exact variables of various glacier models in the introduction is too detailed, so we have simply included a general clarification of overparameterization, "...all glacier evolution models currently suffer from overparameterization issues (i.e., a lack of sufficient observations to constrain model parameters)..."

For PyGEM, there are only 3 model parameters: temperature bias, precipitation factor, and degree-day factor of snow (the degree-day factor of ice is assumed to be a proportion of that of snow, and is therefore not a parameter that is explicitly calibrated). These parameters are stated in the methods (see the “Historical glacier evolution reconstruction” section).

Pg. 4 P1: You have a time gap make sure to discuss the 1978-2012 period or an interval within that.

Noted. We have added thinning rates from 1978-2004 for both Kennicott and Root Glaciers.

“The greater ablation area thinned by 0.44 ± 0.02 m yr⁻¹ on Kennicott Glacier from 1957-1978, which increased to 0.74 ± 0.03 m yr⁻¹ from 1978-2004 and to 1.43 ± 0.06 m yr⁻¹ from 2012-2023. A similar trend exists on Root Glacier, which thinned by 0.25 ± 0.02 m yr⁻¹ from 1957-1978, 0.85 ± 0.05 m yr⁻¹ from 1978-2004, and 1.41 ± 0.09 m yr⁻¹ from 2012-2023.”

Pg. 4 P3: It is noted that this is evidence of a glacier seeking a new point of equilibrium. This is true for Kennicott Glacier with much greater terminus thinning, However for Root Glacier with similar losses at all elevations this is typical of a disequilibrium response., although in this case this does not include the accumulation zone, so it could be either.

This paragraph is referring to Kennicott Glacier and the Kennicott Glacier terminus. We have added “*on Kennicott Glacier*” to make this more clear.

Pg. 7 P1: More detail-For the ice shelf feature noted, what is the velocity across this feature in 1978 vs 2023? How much has the ELA risen? If there are not specific numbers to report here utilize other regional observations.

This is referring to a bed feature, not an ice feature. Velocity changes at a point above this shelf feature where the 1960 velocity has data shows a decrease from ~26 m/yr to <5 m/yr at present.

See the above response to reviewer 1 about the ELA changes. [we have added a supplemental figure to estimate and show this]

Pg. 7 P2: This illustrates importance of noting shifts in the elevation where the ablation area begins-ELA.

Noted.

Pg. 8 P1: For Clarity-There is not an up valley migration of specific debris it is the upper limit of the surface debris cover that has migrated upglacier.

Noted. We have specified that the “upper limit” has migrated up-valley.

Pg. 10 Is it worth noting here the shift in equilibrium status of Taku Glacier which fit the pattern above from 1950-2006 as noted by flux near ELA (Pelto, 2008) and after 2010 does not Davies et al (2024)? (<https://tc.copernicus.org/articles/2/147/2008/tc-2-147-2008.html>)

Yes, thank you for sharing this citation. Both studies show a shift in ELA over decades in the 20th and/or 21st centuries. These are certainly worth noting (Davies et al. 2024 was already cited), and we have added the Pelto et al. (2008) citation on page 10, stating it in reference to long-term patterns of rising ELAs in Alaska. While Pelto et al. (2008) observe a shift to thinning just above the ELA, they do not observe a notable decrease in ice flux. However, they suspect this to begin to play an impact on flux into the ablation area over longer timescales (20+ years), which is likely part of the changes we observe on Kennicott Glacier in recent years (e.g., Fig. 3f the 7.5-10 km gate shows a decrease in influx. It would be interesting to see fluxes above this, with more data).

A new paper Kindstedt et al (2025) examines changes of the Eclipse Icefield melt events and firn character-“ 2016 to 2023 there has been a 1.67 °C warming of the firn at 14 m depth”. They indicate firn would become temperate allowing increased meltwater percolation without refreeze. Does this have relevance in your flux divergence for the future into the ablation zone?

This is a very interesting study, thank you for bringing it to our attention. There are definitely potential implications of these findings to our study site; these findings imply that similar processes are or will likely occur in the firn on Kennicott and Root Glaciers (some of which is below 3000 m a.s.l.), altering the hydrologic systems of these glaciers. A warming firn reduces the ability of the system to buffer runoff, particularly during extreme melt events. On a short term, meltwater runoff would percolate to the base of the glacier through crevasses and englacial streams, ultimately increasing velocity through sliding. Due to the large and heavily crevassed icefalls below nearly all the accumulation areas on Kennicott and Root Glaciers, and the relative size of the glaciers, this effect would likely not play a large role in flux divergence estimates in the lower portions of the glaciers. Over long periods of time, warming firn would lead to a reduction in glacier mass due to less water being stored or refrozen in the firn and more water leaving the system as runoff. This would impact the volume of ice delivered down-glacier, but the timescale and magnitude of these changes are likely minor compared to relatively large thinning signals in the ablation area due to surface mass balance.

There are a lot of complexities when it comes to glacier processes and feedbacks, many of which are difficult to disentangle and currently beyond the scope of existing glacier evolution models. We do believe that firn warming is an important aspect in understanding glacier processes (especially in the accumulation area), but do not believe warming firn would have a large effect on flux divergence compared to the effects due to thinning in the ablation zone. As such, discussion and implications regarding firn are beyond the scope of this study, although they would be interesting to investigate in the future (e.g., mapping firn aquifers and potential changes over time).

Kindstedt, I., Winski, D., Stevens, C. M., Skelton, E., Copland, L., Kreutz, K., Mannello, M., Clavette, R., Holmes, J., Albert, M., and Williamson, S. N.: Ongoing firn warming at Eclipse Icefield, Yukon, indicates potential widespread meltwater percolation and retention in firn pack across the St. Elias Range, EGUsphere [preprint], <https://doi.org/10.5194/egusphere-2024-3807>, 2025.

Reviewer #2 (Remarks on code availability):

I reviewed this code to determine its availability and level of development explanation. Both are excellent. I am not capable of assessing the code effectiveness.

Thank you!

Reviewer #3 and #4:

(see attached)

review NCOMMS-24-80605

Albin Wells and co-authors: An 85-year record of glacier change and impacts on future projections for Kennicott and Root Glaciers, Alaska

General comments:

This is a solid and rich work on changes of two Alaskan glaciers including new data sets and modelling. They present noteworthy results on glacier changes and reconstruct glacier changes and velocity records using historical aerial photographs. We find that references to previous literature could be added.

Thank you for your sincere, thoughtful, and thorough feedback, which we believe has substantially strengthened the overall writing, clarity, and work. We have responded to your comments line-by-line, and implemented changes to the manuscript as appropriate. The amount of comments shows a clear interest in the study and a large investment of time, which we greatly appreciate. Many of the changes are stylistic, so at some points we have opted to keep our personal style. We have clearly explained the motivation behind our choices to each of your comments in hopes that this helps to clarify things from your end as well.

Assessment of “glacier-wide” changes throughout the manuscript. This requires further justification and explanation. It is not a glacier wide average when not all of a glacier is covered. If changes are small in upper parts, but largely negative in other parts, inclusion of upper parts would influence the average value per glacier. We question the use of glacier wide averages as their glacier mapping does not cover all of the glaciers, quite a big part of the upper regions is missing.

We agree with you completely. We have removed the term “glacier-wide” when referring to geodetic mass balance over the area covered by our DEMs.

You are correct that this discrepancy in area coverage would bias “glacier-wide” elevation change results if the missing areas were not considered. This is the exact reason that we tried to explicitly refer to elevation change only for specific regions of the glacier throughout the manuscript. This is particularly prevalent in the *Historical record reveals periods of equilibrium and accelerated mass loss* section, where we present elevation change results and always explicitly state the specific area of the glacier (e.g., “lowest ~5km of Kennicott”, “greater ablation area”, and “terminus”). In reviewing each occurrence, we did find places where this could be more clear. We thus made the following changes, from:

“The rate of mass loss at the terminus of Kennicott Glacier increased nearly six-fold (-0.33 to -1.84 m/yr) from 1957-1978 to 1978-2004 compared to a modest 16% increase in glacier-wide mass loss (-0.46 to -0.53 m/yr) during the same timeline.”

to:

“The rate of thinning at the terminus of Kennicott Glacier increased nearly six-fold (0.33 to 1.84 m/yr) from 1957-1978 to 1978-2004, compared to a relatively modest 16% increase across the rest of the ablation zone during the same period (0.46 to

0.53 m/yr).”

We have also removed the term “glacier-wide” from the sentence, “*On Root Glacier—which has notably less debris cover than Kennicott Glacier (Fig. 1a)—the glacier-wide spatially-distributed thinning pattern is relatively consistent even as the magnitude of thinning increases over time (Fig. 2a-e)*” as it is not relevant to the overall statement, and not correctly stated, as we only observe thinning patterns in the ablation zone of Root Glacier.

We have, however, noted that the area covered by historical DEMs represents over 95% of the mass loss. This assumption is stated in the caption of figure 2, “*The area covered by historical DEMs post-1957 represents over 95% of glacier-wide mass loss.*” This is based on the calculations from 2000-2019 using glacier-wide Hugonnet et al. (2021) elevation change rates with both shapefiles for the historical DEM extent and the complete glacier outline. We have added a Supplemental Text S1 explaining this:

“To assess the representation of glacier-wide mass changes, we used glacier-wide elevation change maps from 2000-2019 (Hugonnet et al., 2021). The total mass change was calculated in the ablation areas and glacier-wide for Kennicott and Root Glaciers (Fig. 2g) using a density of 900 kg m⁻³ and 850 kg m⁻³ for the ablation area and the full glacier, respectively. The areal coverage presented in the study represents 95.1% of total mass loss on Kennicott Glacier (41.4% of area) and 95.2% of total mass on Root Glacier (38.9% of area), indicating that the area covered by the generated DEMs from this study capture nearly all of the present-day mass change. The minimal mass change in the accumulation areas align with previous work, which shows thinning in the accumulation area from 1994-2013 converges to zero across much of Alaska (Larsen et al., 2015).”

The authors claim that they have a 85 year record, we expected annual series with such wording, not some subperiods of surface elevation maps. We encourage to revisit the title and how the use of record and glacier-wide averages are presented throughout the document. See also detailed comments below. The figures are of varying quality and could be harmonised more.

We appreciate you sharing your interpretation. We do not claim to have 85 years of continuous, annual measurements nor want to give off that impression. We will work to address your comments down below, but we have decided to retain the current title of “*An 85-year record of glacier change...*” because we feel that this more accurately portrays the nature of our data in comparison to alternative like “*85 years of glacier change...*” which we feel would potentially be more misleading with regards to having annual measurements. Additionally, as stated by other reviewers, the length of the record is considered a major strength of the study. Hence, we feel it is important to highlight the length of the record within the title.

The study provides several interesting and valuable datasets related to different time periods and spatial extents, and contains different components and analysis. This requires a high level of precision in formulations when presenting and discussion of results, and the

manuscript would benefit from more specific formulations several places (please see Detailed comments). For example, please be specific about which data, locations and periods is referred to and avoid using “historical” and “modern” instead of specific periods (the use of these terms is ok in relation to model calibration data since it is clearly defined here).

Please find our responses to specific comments below.

Brief method explanations are needed in several places in the main text to give the reader an overview of how the different results/data are produced. The main text before the methods section needs to be written such that the reader can read this text in its entirety and understand what is done (at a high level) and then choose to dive into the methods for details. A single sentence or amendment of a sentence is often sufficient, such as the first sentence in the section “Historical record reveals periods of equilibrium and accelerated mass loss”. Please see specific suggestions in the detailed comments.

Please find our responses to specific comments below. We agree that this is a delicate balance coming from the high-impact journal format and will seek to improve this.

Regarding the mentioned example, the first sentence of this section is a broad continuation of the final paragraph of the introduction (which mentions more specifically the types of data/changes we look into), but is also a general topic sentence for the first half of the paper. As such, we find it difficult to appropriately refer to specific methods/changes in this sentence, and opt to keep it broad. We considered flipping the structure of the sentence to incorporate concepts/phrases like “DEM differencing” or “...from surface elevation observations...”, but ultimately opted to keep this sentence as is, referring primarily to spatial and temporal elements rather than methods or data types.

Reference to findings from other studies is (partly) missing, e.g. are the temporal patterns of mass loss in line with those on other glaciers in Alaska, are there examples of other glaciers where the same mechanisms behind wastage of the terminus is found? Are there relevant previous findings on Kennicott and Root?

It is a little difficult to tell exactly, since we show the oldest record in Alaska and the only estimates that have consistent multi-decadal estimates. That said, a previous study looking at long-term changes of the Wrangell Mountains showed comparable overall mass change since 1957 using the USGS topographic maps compared to our study (Das et al., 2014). This is discussed in Supplementary Text S2 with a comparison between our 1957 DEM and the USGS topographic map from 1957 in Fig. S5. For temporal patterns, the most useful dataset to compare is the USGS glacier benchmark program (O’Neel et al., 2019). Lemon Creek Glacier (near Juneau) has the longest mass balance record, dating back to the mid-1950s. Wolverine and Gulkana Glaciers each have records dating to the late-1960s. Of particular interest is the Lemon Creek record, which shows near equilibrium in the 1950s. Cumulative mass balance throughout the 1960s and 1970s is slightly negative. Since ~1980, the mass balance has become more negative with each decade, and shows drastic mass loss. This trend aligns with our findings on Kennicott and Root Glaciers. For all three Alaskan benchmark glaciers, negative mass balance more than doubled from 1967-1990 to

post-1990. Davies et al. (2024) similarly show a doubling of volume loss from 1979-2010 to 2010-2020 on the Juneau Icefield. Pelto et al. (2008) show a shift in thickening to thinning just above the ELA on Taku Glacier, AK before and after 1988. We observe similar relative changes in the mass balance on Kennicott and Root Glaciers over time.

We have added the O'Neel et al. (2019) and Pelto et al. (2008) references to the text, and included them in the discussion in the penultimate paragraph of the main text. The other two references are already cited there:

Das et al. 2014: doi.org/10.3189/2014JoG13J119

O'Neel et al. 2019: doi.org/10.1017/jog.2019.66

Davies et al. 2024: doi.org/10.1038/s41467-024-49269-y

Pelto et al. 2008: doi.org/10.5194/tc-2-147-2008

We have also added a few references to other parts of the text, which report similar mechanisms behind wastage and change. Sato et al. (2022) discuss the dynamic response of lake-terminating glaciers: they showed it can increase flow, reduce compression at the terminus, and ultimately lead to dynamic thinning. We added this reference to the section on debris cover and terminus stagnation. Anderson, B. et al. (2021) show the impact of debris and calving on modeling lake-terminating glaciers: they find that calving can be important, with its relative importance diminishing if the glacier retreats out of its proglacial lake. We've added this reference to the model limitations section.

Sato et al., 2022: doi.org/10.5194/tc-16-2643-2022

Anderson, B. et al., 2021: doi.org/10.1016/j.gloplacha.2021.103593

There are certainly relevant previous studies on Kennicott and Root Glaciers, which are outlined in Supplemental Test S2 in slightly more detail than the main text. In particular, Anderson et al. (2021a,b) show in detail various feedback processes on Kennicott Glacier. This study builds off of and supports many of the findings in this study by expanding the temporal series and obtaining spatially-distributed surface and bed elevations.

Anderson et al. 2021a: doi.org/10.5194/tc-15-265-2021

Anderson et al. 2021b: doi.org/10.3389/feart.2021.680995

All data links in data availability are working, but it would be helpful if the authors had provided more direct links to the datasets where possible. It is not straightforward to find them.

All products produced from the study can be downloaded directly from Zenodo. We linked source data (e.g., digitized aerial photographs) to the website/portal it can be found on, but unfortunately cannot provide links to every image that is used due to practical constraints.

The PyGEM model code and documentation is available on GitHub (github.com/drounce/PyGEM). OK Code is available at https://github.com/albinwwells/past_and_future_mb. Seems fine. Model code - no comments.

Great, thank you.

Other comments:

As the manuscript did not provide line numbers when we first looked at it, we copied the text in the manuscript for reference. It is also challenging when as in this case two persons (senior +ECR) are reviewing the manuscript. We received line numbers after partly finished the review, so the feedback will be a mixture of copied text and linenumber. Please next time provide line numbers. Some minor comments are therefore not included as it was too cumbersome to refer to them.

We apologize for this inconvenience and understand the mixture of lines and paragraphs. We had not initially realized that line numbers were not maintained when converting the document to the pdf, but will make sure to avoid this in the future.

We are not used to the format of Nature Com and find it inconvenient in the reading to have it so divided. First results then methods then more on methods in supplementary. e.g. both in methods and in suppl the density conversion factor is mentioned, but only in the methods it is explained why this factor is used. The uncertainty section does not give the uncertainties, could be given where they are explained.

We acknowledge that the structure of a Nature Communications paper deviates from traditional academic writing. This poses a challenge in both presenting results and understanding them, which we have sought to address in response to your comments below. Specifically, we sought to introduce enough methods to understand the paper at first read, while excluding redundancies in methods later on. We find it difficult to include values for uncertainty in the methods without presenting results, since these values reflect many different spatial areas, time periods, and datasets. As such, we opt to present only the methodology and equations used to solve for the uncertainties, and include particular error values in figures/tables/results throughout the paper.

Detailed comments

Title: is the title representative? An 85-year record gives the impression of annual measurements, could you rather state the period in the title?

After much thought and discussion, we feel the title is representative. We do not believe that it is misleading, and do not feel like “85 years of change...” or “Glacier change from 1938-2023...” is a more effective title, but will ultimately defer to the editor on this.

Abstract:

p1, Abstract: “results prior to 1957 show two decades of near equilibrium followed” -> hard to follow which time period this refers to, perhaps change to “results show two decades of near equilibrium prior to 1957, followed by”

The sentence has been changed as suggested.

p1, Abstract: “These results highlight the unique insights that historical aerial photographs provide into past and future changes” -> Be more specific about which changes.

The word “glacier” has been added to emphasize that these are “...past and future glacier changes”. We look at elevation/thickness, mass, debris, velocity/dynamics, and mass balance, which we feel is too much detail/too difficult to effectively include in the abstract, and thus opt to refer to this more broadly as “glacier changes” here.

L19. Not sure that it is unique that one uses historical aerial photographs to assess changes, I would remove unique here. The authors could in introduction refer to studies using aerial photographs to assess historical change, there are several from Norway, the European Alps etc. and other regions.

You are certainly correct—and we hope to elevate those studies by highlighting them in the introduction and references. The usage of the word “unique” here is not claiming we are unique in using these data, rather, that historical imagery provides unique insights into understanding a glacier’s past and future. These insights cannot be gleaned just from present-day data or information; we believe that this makes them “unique”.

The abstract is rather general, we would prefer more results and facts.

Thank you for this suggestion, it’s certainly noted. Due to the broad audience of Nature Communications and strict abstract length requirements, we have opted not to include more technical details in the abstract. We agree that the abstract would benefit from these details if afforded slightly more space (and actually had such details in previous versions before tightening it down to align with journal requirements), but we ultimately can only emphasize a few key results in the abstract beyond motivating the work to a broad audience.

Introduction:

p2. first paragraph. however, there are longer time series available from glaciers that have more details and are more accurate, such as glaciological mass balance. This information could be added.

Noted—this is a good point. However, within the frame of large-scale modeling and glacier change, this point does not quite fit in in this opening. In the second paragraph, we give a nod to “...or older records that are only available for a small subset of monitored glaciers” and believe this captures the sentiment of this point.

We also note that while glaciological measurements (particularly the longer time series of some key benchmark glaciers) are invaluable datasets, estimating glacier-wide changes from point measurements is non-trivial as well. Hence, we feel this mention of older records and their availability is more appropriate than including specific details.

p2, Introduction, paragraph 1: “the recent ubiquity” -> the word “ubiquity” may be difficult for an international readership, consider replacing it with a more accessible word.

We considered synonyms, but feel the use of the word ubiquity captures our meaning perfectly and succinctly. We will defer to the editor if they feel the text has become inaccessible.

p2, Introduction, paragraph 1: “has enabled two decades” -> specify the time period after this or in relation to the citation, e.g. (2000-2019; Hugonnet et al., 2021).

This has been changed as suggested.

p2, Introduction, paragraph 1: “our ability to model future glacier change is shrouded with uncertainty due to the relatively short time periods of these continuous observations”. This is only one of many factors contributing to the uncertainty in model projections, what about e.g. uncertainties in future projections, accuracy and resolution of climate data, or

parameterization of models? Consider rephrasing such that it is clear that this is not the only factor. I would also suggest adding a reference to this statement.

Yes, there are many sources of uncertainty, and GCM and RCP uncertainty are large. The reference Marzeion et al., 2020 has been added, which directly compares sources of uncertainty in glacier projections. The sentence has been changed to emphasize this is a key limitation in model uncertainty, not overall projection uncertainty:

“Still, the models for estimating future glacier change are shrouded with uncertainty due to the relatively short time periods of these continuous observations (two decades) in comparison with glacier response times (decades to centuries) (Marzeion et al., 2020).”

p2, Introduction, paragraph 1 or 2: Consider specifying in paragraph 1 or 2 how the mass loss estimates are derived, i.e. that they represent geodetic mass balance derived from differencing DEMs. This will make it more clear in paragraph 3 how historical DEMs can be used to extend mass change records.

Noted, this has been added to the final sentence in paragraph 2:

“Fortunately, long-term mass balance data—which include glaciological (in situ) observations and geodetic mass balance from digital elevation model (DEM) differencing—provide valuable constraints to overcome these challenges and improve glacier projections (Eis et al., 2019; Eis et al., 2021).”

p2, Introduction, paragraph 2: Redundant semicolon “;” after “Marzeion et al., 2012”. Could add a more recent reference here also, e.g. Zekollari, H., Huss, M., and Farinotti, D.: Modelling the future evolution of glaciers in the European Alps under the EURO-CORDEX RCM ensemble, *The Cryosphere*, 13, 1125–1146, <https://doi.org/10.5194/tc-13-1125-2019>, 2019.

Thank you for catching that, and for the suggested citation. It certainly fits, and has been added.

p2, Introduction, paragraph 3. The first sentence could be shortened to: Historical aerial photographs provide unique opportunities to derive digital elevation models (DEMs) and orthophotos to assess glacier changes and go well back prior to the satellite era. And here you could cite some of these studies, e.g. ...

The suggestion removes the part of the sentence “...combined with structure-from-motion photogrammetry...”, which we believe is a key component of this point. There are tons of historical aerial photographs, but only the ones of high enough quality and with enough overlap (i.e., they can be used for SfM photogrammetry) are useful for extending records of glacier change. As such, we have maintained this component of the original sentence.

Aerial photographs have been used for a long time in glaciology. There are many earlier references than this. I don’t understand why you mention three regions, there are also other regions monitored, I would drop it and rather join the references. thus, quantify glacier changes (e.g.....; ...; ...). You could also add they are used to validate

glaciological mass balance records.

We want to highlight widespread studies, of which but none have been performed in most of the glaciated regions of Alaska. We've opted to keep the primary formulation of the sentence, but have stated this more explicitly:

“...and to quantify glacier mass change in Europe (e.g., Geyman et al., 2022; Mannerfelt et al., 2022), western North America (e.g., Tennant and Menounos, 2013; Knuth et al., 2023; Mukherjee et al., 2023), and Antarctica (Child et al., 2021; Dømggaard et al., 2024). Only one study assesses changes in Alaska (Juneau Icefield; Davies et al., 2024).”

p2, Introduction, paragraph 3: “before Landsat and» Add a reference year to indicate which period “before Landsat” refers to.

We've added “(1972)”.

p2, Introduction, paragraph 3: “although these data often have large uncertainties (~15 to 45 m)” It is not clear what these uncertainties refer to, is it the elevation differences derived from these maps? Consider being more specific than “data”.

The “data” refer to the “historical contour maps” mentioned at the start of the sentence. The wording has been modified to perhaps connect the sentence in a more cohesive way, *“Historical elevation contour maps from the 1950s and 1960s have also been used to quantify long-term glacier changes over larger regions in Alaska, although these contain large uncertainties (~15 to 45 m) such that uncertainty in 40+ years mass balance exceed 20% of the elevation change signal (e.g., Berthier et al., 2010; Das et al., 2014).”*

p3, Introduction, paragraph 4: , we use a suite of datasets ... this sentence is very long and hard to read.

Noted. We've maintained the current sentence structure as we want to keep the description of the glaciers as *“...located adjacent to the Kennecott Mines National Historic Landmark within Wrangell-St. Elias National Park and Preserve”*. This project was largely made possible by the support from the National Park Service and we would thus like to keep this description of these glaciers within the parks. As hallmarks of the park, these glaciers impact thousands of people annually, and are crucial both for tourism and as sentinels of climate change.

avoid and/or use just or

Done

L67 rather write ..quantify changes since 1938 for two...

Done

p2, Introduction, paragraph 4: “We produce and/or use” -> This study makes significant contributions in processing and providing historical datasets. The contributions of the study would be better highlighted if this statement was split in two, first stating which new datasets are produced in this study and then a second statement on which sources are “only” used, e.g. “We produce several new datasets, including [...]. We complement this data with [...] to estimate spatially-distributed changes in elevation and glacier dynamics over time.”

Noted. We have split the sentence such that it now reads, “*We produce several new datasets, including high-resolution DEMs from film in 1938, 1957, and 1978, as well as new surface velocity data from 1957-1962 from orthophotos. We complement these data with DEMs from 2004, 2012, and 2023 to obtain spatially-distributed changes in elevation and glacier dynamics over time.*”

p2, Introduction, paragraph 4: Not clear what is meant by “modern sources”, please be more specific.

“Modern sources” basically refers to not historical film images—these ASTER, IFSAR, and a NPS campaign producing a DEM from optical imagery. These are all clearly shown in Extended Data Table 1, which is referenced in the previous sentence. The rewriting of the sentence precludes any necessary changes, and this phrase was removed.

p3, Introduction, paragraph 4: “measure spatially-distributed changes” -> “estimate spatially-distributed changes”?

We have changed “measure” to “obtain”.

L75 continuous elevation change. do you mean since you have digital DEM? I am not sure this adds anything.

“Continuous” is meant in contrast to numerous studies that solely measure/calculate mass change from historical to present-day. We have data from every ~20 years since 1938, giving us clear patterns and changes in mass over time, which is unique to previous work in other regions (and crucial for model calibration). We have replaced “continuous” with “multi-temporal.”

p3, Introduction, paragraph 4: “revealing considerable changes in future estimates” -> Not clear what is compared here. Suggest to be more specific, e.g. “revealing considerable differences between future estimates compared to model calibration using present day mass changes”.

Thank you, the sentence has been amended per the suggestion.

Introduction, Fig. 1. We suggest some modification of the figure. Use smaller font on a-g. It is a bit confusing that the inset (g) with zoom-in on bed topography is marked 1938 on the overview map (a), and that the image in inset (b) shows a historical photograph of not only the terminus. Why not show the Washburn 1938 DEM (Methods, p. 13) of the terminus in Fig. 1b? This would make the figure more consistent with respect to how the other DEMs and DEM extents are shown. Also, please add the year of the glacier outline in the caption in relation to “present-day glacier outlines”, be specific.

Noted. The outline years have been added to the figure caption. Text size was slightly reduced. None of the panels actually show DEMs, rather, they show orthophotos (as described in the figure caption). We opt to show orthophotos because DEMs are difficult to look at and interpret by themselves (all of the DEMs are similar in appearance with the exception of extent) and the orthophotos give a clear, easily understandable representation of the data we use to produce the DEMs. DEM differences are shown in Figure 2, which reveal much more information. We use some of the orthophotos for feature-tracking to

derive velocities, so we also feel it is beneficial to show what these orthophotos look like. We would also like to note that all DEMs are available for download, so it is not necessary to include graphic examples of them in the main text.

The key difference between the Washburn images and the other images we use is that the Washburn images are the only ones that are oblique. As such, they look fundamentally different from panels c-e.

The figure would be easier to read with less panels.

Noted. We do not see an alternative way to share a comprehensive overview of the study site and data, and feel it would be misleading to include/exclude only some of the datasets.

Historical record reveals periods of equilibrium and accelerated mass loss:

p.4, first paragraph: “assess changes across” -> being specific about what types of changes are assessed helps the reader to know what to expect in this section, e.g. “assess changes in glacier geometry and associated mass losses across”

Noted. The sentence now reads, “*We assess changes in glacier geometry, thinning, and associated mass loss across...*”

p.4, first paragraph: “including a specific focus on the terminus of Kennicott Glacier—delineated to be below Root Glacier—to extend the record back to 1938 (Fig. 2g).” The part “-delineated to be” can be removed. “-to extend the record back to 1938” is a bit confusing because it is not clear which “record” is referred to, suggest to reformulate: “including a specific focus on the terminus of Kennicott Glacier (below Root Glacier), where DEM coverage extends back to 1938 (Fig. 2g).”

Noted. This has been implemented.

p4. first paragraph after headline, 2g is it meant 1g?

No—this is pointing to the Fig. 2g, which shows the various extents of the elevation and mass change calculations, including the terminus of Kennicott Glacier (dark brown).

A bit confusing that results are here prior to methods - but this is perhaps the journal style. Yes, this is the journal formatting.

L96 66-year period of mass loss. you mean mass loss recorded over this 66 year period, you don't have results for all years.

Yes, you are correct. We do not have (nor claim to have) annual measurements of mass loss over the 66-year time period. The sentence now reads, “*Mass loss over this 66-year period corresponds to...*”

p4, first paragraph: Statement “Elevation change observations reveal Kennicott and Root Glaciers were roughly in a state of equilibrium from 1938 to 1957, since which mass loss has accelerated (Fig. 2).” I don't agree with this statement. If you consider 1938-1957 (Fig. 2a) to show near equilibrium due to limited mass change/slightly positive then I would also say that 1957-1978 (Fig. 2b) also shows near equilibrium (limited mass change/slightly negative). Also, 1938-1957 shows mass gains on the lower part of the tongue, if the 1938 DEM covered larger parts of the ablation area, wouldn't you expect to see even more

positive rates here such that this may have been a period of mass gains (larger than the losses 1957-1978)?

Noted—We agree that there is definitely a lot to consider here. First and foremost, due to the nature of the 1938 photographs (oblique, and few in number), the 1938 DEM has substantially larger uncertainties than the other DEMs. This is due to both limited ground control as a result of the oblique imagery and the distance of the point of acquisition to the ground features. As such, the study area appears to have a slightly positive elevation change (as you state), but this signal is within the range of uncertainty (as in Fig. 2f; Extended Data Fig. 7). We thus qualitatively characterize changes during this time period as “balanced to slightly positive” and “roughly in equilibrium.” We present the actual quantities of elevation change in the text.

We agree that mass change at the Kennicott Glacier terminus from 1957-1978 is near equilibrium, but it is distinctly negative (the signal here exceeds the error). We have chosen to keep the sentiment of the statement, “*Elevation change calculations reveal Kennicott and Root Glaciers were roughly in a state of equilibrium from 1938 to 1957, with mass loss accelerating thereafter (Fig. 2)*” because the emphasis we are making is that post-1957, mass loss has continuously accelerated. This is shown distinctly in Fig. 2f, and can also be seen across Fig. 2b-e.

If the 1938 DEM showed more expansive coverage, we would not necessarily expect more positive rates. The 1938 DEM uncertainties are greatest farthest away from where the photographs were taken, which is towards the top of the DEM coverage. Again, this is due to the ground-control points being spaced where available, which was generally towards the terminus. In turn, the 1938 DEM is most accurate at the terminus and may suffer from larger biases farther up-glacier (which we capture in the uncertainty), despite detrending the 1938 DEM to help reduce these artifacts.

p4, first paragraph: “30% decrease in Kennicott Glacier’s terminus area” -> Refer to Fig. 2g again here at end of sentence.

Noted, thanks. We have added a reference to Fig. 1g, which shows this difference.

p4, second paragraph and the use of “glacier-wide”: “The historical elevation change record reveals roughly two decades of glacier-wide thinning from 1957-1978 followed by anomalous terminus thinning on Kennicott Glacier relative to the rest of the glacier (Fig. 2).” It is not fair to say glacier-wide thinning because you only have coverage over the lower parts of the glaciers. Please add the time period related to the terminus thinning -> post 1978? Please also see comment on Fig. 2 in relation to the use of “glacier-wide”.

Noted. The term “glacier-wide” has been removed here.

p4, second paragraph: “the western side of the terminus has particularly thin ice (<30 m) and—given current thinning rates—will likely be ice-free in the next decade” -> would add here the current thinning rates to support the statement that this part will be ice-free within a decade. The thinning rate for 2012-2023 of -1.41 myr^{-1} was just mentioned and does not

align with this statement. Suggestion: “and given current thinning rates in this area (around X-Y myr-1; Fig. 2e)”.

Noted. There is a lot of local variability in thinning rates in this part of the glacier, but values range from roughly -2 to -4 m/yr. We have added as suggested, stating “...*given current thinning rates in this area (~3 m/yr; Fig. 2f)*...”

p4, second paragraph: “the glacier-wide spatially-distributed thinning pattern [...] magnitude of thinning increases over time (Fig. 2a-e)” -> Using “glacier-wide spatially-distributed pattern” is confusing, please reformulate. Also, please be specific instead of “over time”. Magnitude of thinning for Root increased until 2004-2012, it did not increase in the last period (Fig. 2f).

The term “glacier-wide” was removed per the earlier comment. You are correct that it did not increase from 2012-2023 compared to 2004-2012. However, we have maintained the current formulation as the thinning rates have “increased over time” despite the relative consistency in thinning from 2004-2012 and 2012-2023; when looking at the three ~20-year periods since 1957, thinning rates have increased over time.

p4, second paragraph: “In turn, on Root Glacier, thinning rates at the terminus are similar to those up-glacier.” -> Merge with previous sentence to describe the pattern you are referring to, or with the next sentence: “In contrast to Kennicott Glacier, thinning rates on Root Glaciers are [...], suggesting that [...]” Also, it is not entirely clear what you refer to as the Root terminus. Last sentence: “glacier dynamics and debris-cover are critical drivers in understanding thinning patterns” -> “glacier dynamics and debris-cover are critical drivers behind thinning patterns” (glacier dynamics and debris-cover are drivers of patterns, not of the understanding?)

The sentences have been merged following the suggestion. And noted, they are drivers of thinning patterns and critical in understanding those patterns. The sentence now reads, “*In contrast to Kennicott Glacier, thinning rates at the terminus of Root Glacier are similar to those up-glacier, suggesting that glacier dynamics and debris-cover are critical drivers of spatial thinning patterns on Kennicott Glacier.*”

The exact terminus of Root Glacier is not exactly defined because it is not all that important in this context. Regardless how you delineate it, there is no change in the spatial pattern of thinning on Root Glacier across the ablation zone (down to the terminus)--thinning only increases in magnitude over time, but the pattern remains constant (unlike Kennicott).

p4, last paragraph: “Since the turn of the century,” -> Please specify 21st century.
Done.

p4, last paragraph: “terminus thinning has remained steady while thinning in the ablation area above the terminus has roughly doubled” Is this referring to the Kennicott Glacier Terminus vs Kennicott Glacier (dark brown and light brown in Fig. 2f)? And roughly doubled compared to the previous two decades (1978-2004)? Please be more specific, suggest: “Since the turn of the 21 century (2004-2023), the thinning of the Kennicott Glacier terminus has remained steady while thinning in the Kennicott Glacier ablation area has roughly doubled, compared to the previous two decades (1978-2004; Fig. 2f).”

Yes, it is referring to the dark brown and light brown in Fig. 2f. The sentence has been updated following the suggested edits.

p4, last paragraph: “Although thinning is slightly greater for the years” -> “Although thinning in both areas is slightly greater for the period”

Done.

p4, last paragraph: “since 1957 and 1978 to the 21st century” -> this is a bit hard to read. Maybe just write “the multi-decadal trend of accelerated thinning from 1957 to 2023 (Fig.2a-e) is evidence..”

Noted. This has been updated following the suggestion.

p4-5, last paragraph: “Relatively modest mass loss throughout the second half of the 20th century has been exacerbated by increasing temperatures that have caused Kennicott and Root Glaciers to lose a combined 180 ± 10 Mt yr⁻¹ from 2012-2023.” And even more in the period 2004-2012. Maybe state the mass loss for the combined period 2004-2023 here? “exacerbated by increasing temperatures” -> probably true, but could be other climatic variability also, e.g. related to precipitation amounts, so maybe be more general: “exacerbated by climate changes”?

Yes, other climatic variables also play a role in this. The sentence now uses “exacerbated by climate changes.” We could easily cite 2004-2023 (which, as you stated, would show even greater losses), but opt to keep the mass loss for 2012-2023 to cite values for the most recent losses.

p5. Anthropocene is not an accepted term, use time specific period.

Noted. This sentence was mostly a repetition of the previous two sentences, so it has been removed entirely.

IFSAR - define

IFSAR is already defined earlier in the manuscript (Fig. 1 caption)

According to figure 2g. A large part of the upper glacier is missing, I am not sure if the sentence ‘..a reasonable estimate of glacier-wide changes.’ is justified.

Please see our response to the comment below, which addresses this in more detail.

p5, Fig. 2, legend/colors: Nice figure. The hatches on the terminus of Kennicott in Fig. 1g does not match the hatches in the legend of 1f. The hatches in the legend matches the annual mass change in 1f from 1938 to 1957, but not the other time periods. This is confusing. The hatches/background colors used for the Area bars in 1g are also slightly different from the rest, please update. Also, consider updating the thickness of the lines for the Hugonnet data such that they match better with the lines shown in the figure.

Thank you. The panel g should show up better now with the hatches of both plots. Per later comments, we’ve added uncertainty to the Hugonnet data.

p5, Fig. 2, caption and use of “glacier-wide”: “The area covered by historical DEMs

post-1957 represents over 95% of mass loss and is thus a reasonable estimate of glacier-wide changes” -> The reasoning here is not clear and requires more explanation. Is it so that you compare your mass loss estimates over the entire hatched area in Fig. 2g with glacier-wide mass losses estimated by Hugonnet et al. (2020) for 2000-2019 and find that your estimates correspond to 95% of this, and thereby assume that most of the mass loss occurs in your surveyed area, while the rest of the glaciers (accumulation areas) have limited mass changes. From Fig. 1f it looks like this may be the case for 2010-2020, where the mass losses 2012-2023 are relatively well aligned, but not for 2004-2012? Do you think this assumption holds over time? The assumption requires more explanation (from the caption it is not clear where the 95% mass loss estimate comes from), and would suggest adding it in the main text since it is very important for the statements in paragraph 2. Currently it is very confusing that glacier-wide changes are mentioned in this paragraph in relation to Fig. 2. Fig. 2 does not show glacier-wide changes and in paragraph 1 it is stated that changes are assessed across the various extents in Fig 2g. Please consider this comment in relation to comment on p. 14, Glacier outline digitization, paragraph 1, below.

We've added a section, which is now the Supplemental Text S1 to clarify this statement:

“To assess the representation of glacier-wide mass changes, we used glacier-wide elevation change maps from 2000-2019 (Hugonnet et al., 2021). The total mass change was calculated in the ablation areas and glacier-wide for Kennicott and Root Glaciers (Fig. 2g) using a density of 900 kg m⁻³ and 850 kg m⁻³ for the ablation area and the full glacier, respectively. The areal coverage presented in the study represents 95.1% of total mass loss on Kennicott Glacier (41.4% of area) and 95.2% of total mass on Root Glacier (38.9% of area), indicating that the area covered by the generated DEMs from this study capture nearly all of the present-day mass change. The minimal mass change in the accumulation areas align with previous work, which shows thinning in the accumulation area from 1994-2013 converges to zero across much of Alaska (Larsen et al., 2015).”

We've made it clear what area mass loss is referring to and how it compares to glacier-wide estimates (e.g., L127). In the outline digitization section, we state that, *“Nonetheless, our digitized outlines (Fig. 2g) represent over 95% of all mass loss (~40% of total glacier area) on Kennicott and Root Glaciers from 2000-2019 (Hugonnet et al., 2021).”*

p5, Fig. 2, caption: Caption header, add “mass changes”: “Multi-decadal mass changes on...”. The description “Kennicott Glacier ablation area” is not entirely clear because the terminus can also be considered a part of the ablation area. Consider updating the caption to a-e: “Kennicott Glacier ablation area (excluding the terminus, labeled as “Kennicott Glacier”)”. Add an explanation in the caption of what σ_{tot} in the disc refers to.

We have added the following statement to the caption: *“The value in the center of each disc represents the mean annual elevation change of the total area (Kennicott and Root Glaciers)”*

The Kennicott Glacier ablation area does indeed include the terminus—this is the primary motivation for needing the hatching and slightly transparent boxes in Fig. 2f-g: it shows that the terminus area overlaps the ablation area.

We've opted to keep the caption as "Annual mass changes..." as it more accurately reflects the data being plotted: mass change per year. We do not feel that this is misleading readers into believing that we have annual data, as the plots clearly show the temporal spans of the mass changes and the caption further states that we are plotting "multi-decadal periods."

p5, Fig.2f, Hugonnet: This dataset is associated with relatively large uncertainties. Would add error bars or perhaps some sort of shaded area around the lines in Fig. 2f to indicate the uncertainty in the Hugonnet estimates. It might actually help you with the "glacier-wide"-argument. Note that studies have shown that Hugonnet estimates differ from repeat airborne LIDAR (e.g. Andreassen et al, 2023. DOI: <https://doi.org/10.1017/aog.2023.70>)

Thank you, noted. The figure has been updated to show errors in the Hugonnet data.

Glacier slowdown drives present-day retreat on debris-covered terminus:

p 5, first paragraph: "The availability and integration of glacier bed observations and long-term velocity changes provide unique observations of nearly six decades of dynamic changes on Kennicott and Root Glaciers." Not clear how these datasets provide dynamic changes. Do you mean here that the availability of bed topography to derive ice thickness (that you use in the flux and stress calculations)? Perhaps (although a bit long): "Ice thickness distributions (1938-2023) derived from differencing DEMs and glacier bed topography, in combination with long-term velocity changes (1960-2017), provide a unique opportunity to assess ice flux divergence and gravitational driving stresses, revealing nearly six decades of dynamic changes on Kennicott and Root glacier."

Yes, this refers to two things: (1) the velocity observations show "dynamic changes" directly, and (2) the bed observations yield ice thickness which enables flux/stress calculations, as you stated. The sentence has been modified following the suggestion.

p. 5-6, first paragraph: "To our knowledge, we present the oldest spatially-distributed record of surface velocities on a mountain glacier." Repetition from introduction, remove.

This has been removed.

p6, paragraph 1: "Both velocity and driving stress have decreased with thinning, with the largest changes occurring in the latter half of the observational period." Please be specific about the period and insert references to Fig 3 in this first sentence, and maybe also about the area considered. "the latter half of the observational period" is very unspecific as the "observational period" for velocities and driving stresses are different. Also, velocities are assessed from 1957/62 to 2017, but in the introduction where you specify the different datasets used there is no mention of this (reader has to go to Methods). Would suggest adding the time periods somewhere, including them in suggestion for p5, first paragraph, maybe also add in the sentence here: "Both velocity (1960-2017; Fig. 3a) and driving stress (1938-2023; Fig. 3e) have decreased with thinning of the terminus and lower ablation areas, with the largest changes occurring in the latter halves of the respective observational periods."

The sentence has been amended following the suggestion.

p6, paragraph 1: Sentence starting with “The near-zero driving” is long and somewhat repetitive, consider shortening or splitting in two. Also, it seems from the sentence that the glacier is not experiencing sufficient gravitational force to drive ice flow in general?

Noted. For now, we have maintained the current sentence structure. Yes—the driving stress in 2023 near the glacier terminus is very low, and undoubtedly a cause for ongoing (and likely continued) decreased surface velocities/stagnation at this part of the glacier.

p6, paragraph 1: “Above this region, a reduction in driving stress of ~30% (~30 kPa) is observed throughout the lowest 4 km of Kennicott Glacier” -> “above this region” in combination with “lowest 4km” makes it difficult to understand where you refer to, please remove “above this region” if the sentence refers to the lowest 4 km.

Done.

p6, paragraph 1: Explanation of the role of the shelf feature starting with “This reduction in driving stress” is difficult to follow. Also, there are three sentences after each other that contain some version of the formulation “the reduction in driving stress over the lowest 4 km”. Please consider if these explanations can be reformulated to be more clear and avoid repetition.

Noted. One instance of “lowest 4km of the glacier” has been removed as it seems very clear which portion of the glacier is being discussed without this stated explicitly. The topographic shelf feature is important because it disrupts the ice flow path: ice must flow over or around this unexpected bed feature. This serves as a sort of pinch-point, cutting off the glacier below this point. We had previously used the term “pinch-point” but later changed it to “shelf” during revision as it more accurately describes this bed feature, which does not extend across the entire glacier width. We have added “*which ice must flow over or around*” to this sentence to help make the importance/consequence of this shelf feature more clear.

p6, paragraph 2: State in the beginning of the paragraph briefly how these quantities were derived so that the reader does not have to refer to the methods, e.g. by amending the second sentence to: “We partition thinning signals into ice flux divergence and climatic mass balance to assess the relative contributions of flux divergence and enhanced melt on thinning rates in different zones in the lower ablation areas on Kennicott and Root glaciers.”

Great, thank you for the suggestion. The sentence has been amended per the suggestion.

p6, paragraph 2: Please provide some reasoning behind why you consider this to be the climatic mass balance and not the climatic-basal mass balance. I am confused by the notation $b_{\text{clim_dot}}$ in the Fig. 3 and the use of the same notation in both Eq 4 and 6. To my understanding Eq 6 is $b_{\text{clim_zone_dot}}$. Combining Eq 2-4: $b_{\text{clim_dot}} = H_{\text{dot}} * \rho_{\text{ice}}$, which is not in line with Eq 6. To our understanding, Eq 4 actually assumes ice flux divergence into the surveyed areas is zero, since you consider only part of the glacier and not a whole glacier. This should be mentioned in relation to Eq 4 and the notation should be cleaned up. Please see in relation to comments in Methods.

Equation 6 has been updated to include “zone”. Further comments will be addressed as appropriate in the Methods.

We've added a statement, "*Assuming negligible basal melt, ...*"

methods

p6, paragraph 2: "slightly more positive" -> "slightly less negative"

Done.

p7, paragraph 3: "Root Glacier displays a similar but subtler pattern as Kennicott Glacier, where decreasing fluxes drive elevation changes despite relatively unchanging climatic mass balance." Not clear in Fig. 3f which pattern you are referring to here, where on Kennicott?

In the lowest 4km and 5km of Root and Kennicott Glaciers, respectively, there is a decrease in flux divergence after 2004 compared to before 1978 or 2004 (exact timing varies a bit). This is referring to the green bars in Fig. 2f on Kennicott Glacier for the <2.5 and 2.5-5 km boxes decreasing after 2004 compared to 1938-1978.

p6 and 7, paragraphs 2 and 3: These paragraphs contain many numbers, please summarize the changes briefly at the end of each paragraph - what are the main drivers and how has this changed over time? Also, the comments on increased thinning on Kennicott glacier are somewhat redundant since this is already discussed in relation to Fig. 1, suggest to skip sentence starting with "Above the terminus.." and rather focus on the drivers of thinning.

Noted. We have made some of these changes: reducing the number of values presented and including a sentence on the drivers of thinning here, "*The historical time-series thus shows drastic changes in flux divergence on the Kennicott Glacier terminus since 1938, displaying a decadal-scale feedback where up-glacier thinning leads to reduced ice influx down-glacier regardless of surface melt.*"

In the previous paragraph, we have also highlighted the complexities regarding consequences of glacier stagnation, "*In turn, stagnation prompts further complexities that either reduce (e.g., debris thickening, reduction in ice cliffs) or enhance (e.g., supraglacial pond expansion, increased surface relief) melt (Anderson et al., 2021b).*"

In paragraph 3, we've also removed unnecessary values/results and focused on the processes driving thinning, "*Near the terminus (below 2.5 km up-glacier; with full debris cover), Root Glacier displays a similar but subtler pattern as Kennicott Glacier, where decreasing fluxes drive elevation changes amid relatively unchanging climatic mass balance. Above this zone (2.5-5 km up-glacier; primarily clean-ice), uncertainties in flux divergence and climatic mass balance complicate the attribution of elevation changes, although the consistent velocity over time suggests the mechanism for thinning over the clean ice on Root Glacier is driven by surface melt processes rather than a change in glacier dynamics.*"

p6 and 7, paragraphs 2 and 3, Fig 3f: In general it is very difficult to see the magnitudes of the changes in flux divergence vs mass balance in Fig. f., top row. A large part of the text is concerned with these differences, consider giving more space to Fig. 3f top row to make them more visible to the reader. The panels can be more rectangular, extending the y-axis, adding more ticks and perhaps some thin grid lines.

Noted. We shifted the focus of the paragraphs on drivers of thinning rather than exact values (see comments below; the comment with quotations of specific lines show some notable additions). We have added some gridlines to make the figure a bit more clear without

introducing too much clutter.

Fig. 3: Please specify which outline (year) is shown in Fig. 3a. Please specify for which extent the lowest point on the centerline refers to (i.e. what does the 0 point on the x-axis in b-e refer to?) The red color on the font is hard to read. Also, consider if switching the color of V_D and V_E gives better contrast on the blue? Could the flux gate cross sections in f be shown on the map in a? Fig. 3f -> consider balancing/harmonize the position of axis text Elevation (m) and Elevation change

We have updated the caption to make these more clear. We now include “...the 1957 glacier outline...” and “The distance up-glacier corresponds to the centerline in (a), which begins at the 2023 glacier terminus.”

Flux gates could be shown in panel a, however, we removed them to reduce the clutter and help improve overall figure readability. These were shown in previous draft iteration of this figure, but removed before the final version. We felt that, in addition to clutter, the delineation of zones as distances were clearly aligned with (b-e) and shown by the centerline in (a). Since we calculate “robust” flux zones (i.e., we calculate flux zones +/-400 m from the specific flux gate placement), we deemed the exact delineation of these zones to be non-critical.

We tested switching the colors of V_D and V_E, but found that the contrast was not improved. We have thus kept the current point colors.

We considered harmonizing the position of the y-axis labels of Fig. 3f, but chose to maintain the current positioning. “Elevation (m)” is best fit on one line since the y-axis values contain 3 digits, and thus take up more space than those for the “Elevation change (m/yr)” plots. We did not want to shift “Elevation (m)” left to align with “Elevation change (m/yr)” because the labels begin to get crowded and jumbled together.

Fig. 3, caption: “The product” -> please be specific: the 1960 velocity map. “Velocity magnitudes from 1960 and since 1990 for ITS_LIVE and Anderson et al. (2021) products are shown at five points on Kennicott and Root Glaciers.” -> difficult sentence, suggest: “Velocity magnitudes from this study (1960), ITS_LIVE (1990 and 2004) and Anderson et al. (2021) (2017) are shown at five points on Kennicott and Root glaciers.” Please explain in the caption what the whiskers on the elevation change plots in f are and note the difference between the time periods covered for Kennicott and Root.

Noted. The caption has been updated following the suggestions. Velocities from ITS LIVE and Anderson et al. are shown for multiple dates post 1990, not just 1990 and 2004 (not every point has an x-label, but each has a tick mark). As such, we’ve specified “(post-1990)” instead of the suggested “(1990 and 2004)” for those data.

We have also added the following to the caption, “Box whiskers represent uncertainty for each value (see Methods)” and specified “...cross-sections on Kennicott (1938-2023) and Root (1957-2023) Glaciers...”

Debris cover evolution leads to terminus wastage:

Could 'used' be used instead of 'leveraging'? Easier to understand for non native English speakers. P8. here you define modern data but as mentioned this term already used earlier, define if first time used.

In general, the language around this has been tightened so we feel this to be the appropriate place to define "modern" data. We've used the words "leverage" and "use" interchangeably throughout the abstract and main text, and even in this paragraph. We will heed the advice of the journal regarding this language, but feel that keeping the term "leveraging" in this sentence improves the overall writing quality and reduces repetition as "using" is used two sentences later.

p. 8: Avoid use "retreat of ice", can be confused with glacier retreat. Remove "further". "over time, showing an up-valley migration" -> "over time, with the up-valley migration". Both of these statements are related to what the orthophotos show.

Noted. The sentence has been amended slightly, "*Orthophotos display the reduction in clean ice extent over time, with the migration of debris...*" but we feel that "upper limit of debris cover" is not necessary to explicitly state. Referring to and "up-valley migration of debris" is consistent with previous literature, e.g. B. Anderson et al. 2021: "Debris cover of the Haupapa/Tasman Glacier has migrated upstream to cover more of the glacier between 1889 and present"

p. 8: Sentence starting with "While the stagnant" is long and difficult to follow, please reformulate.

The sentence has been split into two sentences.

p. 8: "the symptoms of these changes are similarly evident over time." This is unclear, please reformulate and be specific. Is the last sentence "Around 7.5-10 km .. " an example of changes that are "similarly evident over time"? This is not clear.

The following sentence describes this, as you infer. The sentences have been connected with a colon to make this more clear.

p. 8: At the end of the section, could you briefly sum up the effect of the debris cover on thinning rates and the mechanisms behind, and perhaps how you expect this to affect the evolution of Kennicott in the near future?

The glacier processes behind thinning patterns should be more evident throughout the first sections of the manuscript, including here. Some key additions are (approximate line numbers in the track-changed manuscript):

~L168: "*Below this feature, decreased ice velocity (from ~26 m yr⁻¹ in 1960 to <5 m yr⁻¹ in 2017) and in-flux due to a rising equilibrium-line altitude leads to drastic thinning, which has led to substantial reductions in driving stress into the lowest 4 km of Kennicott Glacier. In turn, stagnation prompts further complexities that either reduce (e.g., debris thickening, reduction in ice cliffs) or enhance (e.g., supraglacial pond expansion, increased surface relief) melt (Anderson et al., 2021b).*"

~L195: "*The historical time-series thus shows drastic changes in flux divergence on the Kennicott Glacier terminus since 1938, displaying a decadal-scale feedback where*

up-glacier thinning leads to reduced ice influx down-glacier regardless of surface melt.”

~L203: *“Near the terminus (below 2.5 km up-glacier; with full debris cover), Root Glacier displays a similar but subtler pattern as Kennicott Glacier; where decreasing fluxes drive elevation changes amid relatively unchanging climatic mass balance. Above this zone (2.5-5 km up-glacier; primarily clean-ice), uncertainties in flux divergence and climatic mass balance complicate the attribution of elevation changes, although the consistent velocity over time suggests the mechanism for thinning over the clean ice on Root Glacier is driven by surface melt processes rather than a change in glacier dynamics.”*

~L244: *“Ultimately, terminus stagnation and wastage leading to the formation of proglacial lakes (Extended Data Fig. 3) can increase thinning through a reduction in compressional flow at the terminus and frontal ablation (e.g., Sato et al., 2022), but we do not observe evidence of these mechanisms for enhanced thinning on Kennicott Glacier.”*

Past glacier observations constrain future mass loss:

p. 8, first paragraph: “Leveraging historical data for calibrating glacier evolution models is a considerable advance compared to prior studies that only use more recent (2000-2019) glacier-wide mass balance data” -> in general, this is not a novelty but is commonly done for glaciers where such data are available. But perhaps the only time it is done for Kennicott and Root? This statement needs to be reformulated. Combine Hugonnet and 2000-2019 into a single parentheses as this refers to the same data: “(e.g. 2000-2019; Hugonnet et al., 2021).

Noted; the parenthesis have been merged.

We believe that this is a novelty. We have rigorously searched for studies that use mass balance from historical DEMs to calibrate an evolution model, but have not found studies that do this. We searched all studies citing the historical DEM literature (e.g., Geyman et al 2022, Knuth et al 2023, Mannerfelt et al 2022, Wytiahlowsky et al 2023, Vargo et al 2017, Mukherjee et al 2023, Maurer et al 2019, Dehecq et al 2020, Girod et al 2018, Davies et al 2024, Tennant and Menounos 2013, Domgaard et al. 2024).

The most relevant studies we found were Kinnard et al. (2022), who use historical DEMs from Tennant and Menounos (2013) to initialize an energy balance model. However, they (1) do not project future changes and (2) do not incorporate glacier dynamics in their model. Schuster et al. (2023) project future changes by calibrating OGGM with a combination of geodetic data (i.e., Hugonnet et al., 2021) and glaciological measurements, but do not incorporate historical geodetic mass balance into calibration. Other similar modeling efforts use glaciological mass balance for calibration as well (see introduction for a few of these), however, we do believe that using historical DEMs to constrain an evolution model and evaluating its impact on projections is novel.

Kinnard et al., 2022: doi.org/10.5194/tc-16-3071-2022

Schuster et al., 2023: doi.org/10.1017/aog.2023.57

It seems also to be a mix up global vs local studies. Hugonnet is a newer study giving global estimates, however, many studies in the past have been done for local glaciers and mountain ranges.

Noted. We cite some of these studies in the introduction, as appropriate. Our goal, broadly, is to present an exciting new dataset and show how it impacts our understanding of past changes and modeled projections. A large motivation of this work is that these types of historical imagery exist more broadly but remain underutilized—we are demonstrating a case study on how this can be used to calibrate large-scale glacier evolution models and alter projections. Studies on local glaciers are fascinating but cannot provide this broader scope. Still, to the best of our knowledge, we are not aware of any studies that leverage historical DEMs (spatially-distributed elevation changes over time) into glacier evolution model calibration to project future changes.

p. 8, first paragraph: “Leveraging historical data”/“Calibration with historical data ensures” -> Please be more specific, 2000-2019 is also a historical period, “long-term historical data”?

Noted. We have specified “*Calibration with historical data from 1940-2004 (the data from DEMs produced with historical film imagery) ensures...*”

p. 8, first paragraph: Please specify what period historical data/calibration refers to (which datasets) in the same manner as is done with “modern”.

Done [see comment above]

p. 8, first paragraph: “the model calibrated with” -> please explain briefly what has been done and specify which model is being used (PyGEM), this is currently not done until the methods. For example, this is nicely done in the first sentence of the second paragraph - provide a similar overview in the first paragraph. Suggest to split the sentence starting with “Using monthly air temperature ..” which is currently a bit hard to read, merge with short introduction to PyGEM and continue with the calibration results.

Noted. We have added, “*we simulate glacier mass balance and dynamics with the Python Glacier Evolution Model (PyGEM; Rounce et al., 2023). PyGEM estimates melt using a temperature-index model that includes accounting for sub-debris melt (Rounce et al. 2021), accumulation using temperature thresholds, and refreezing using mean annual air temperature, and updates the glacier geometry annually using a flowline model based on the shallow-ice approximation (Maussion et al. 2019).*”

p. 8, first paragraph: Please briefly explain the experiment with the “historical calibration” vs “modern calibration”. Consider in relation to the above comment.

This should be more clear following the above additions.

p. 8, first paragraph: “fits the known glacier mass in the 21st century,” -> These are also glacier mass estimates, but from other sources, please reformulate.

Noted. This has been reworded as “consensus glacier mass” following from the ‘consensus’ ice thickness products (Farinotti et al., 2019).

p. 8/9, second paragraph: “a notable reduction in mass loss in the first half of the 21st century compared to the model calibrated with modern data, followed by accelerated mass loss after 2050 (Fig. 4c, d).” -> notable reduction, better to use “shows lower mass loss” as reduction in mass loss sounds like the mass losses decelerate. Also, it is a bit confusing that the last part of the sentence is related to the “historical” model when the “modern” model is just mentioned: make sure that it is clear what is comparison and what is description.

We have changed this to “...*notably less mass loss*...”

We have decided to keep the remainder of the sentence the same, as repeating “model calibrated with historical data” would be redundant and make the sentence more difficult to read. We have maintained the “compared to the model calibrated with modern data” because we need a reference for the “notably less mass loss”

p. 9, second paragraph: “While the decrease in the rate of mass loss” -> this sentence is very unclear, what is comparison and what is description? Difficult to follow this in the figure. Please reformulate. And Fig. e and f does not show rate of mass loss, but climatic mass balance in mm w.e (volumetric).

The sentence has been amended to specify “...*between the models with historical and modern calibrations*...” The unit of mm w.e. is essentially $[\text{kg}][\text{m}^{-2}][\text{yr}^{-1}][10^{-3}]$, which is thus a *specific rate of mass change* (Cogley et al. 2011).

p. 9, second paragraph: “By 2100, GCM spread plays a large role in model projections as the remaining glacier mass falls within GCM spread”. This is unclear, what does remaining glacier mass falls within GCM spread mean? Please reformulate. It is not clear from this sentence if you are comparing the differences between models in relation to the uncertainty associated with GCMs.

We have added a reference to the Fig. 4c-d boxplots, which show the GCM spread for each SSP scenario. This is essentially what we are referring to. Within each SSP, there are various climate model forcings (i.e., GCMs). In Fig. 4c-d, the lines represent the mean of the GCMs for a given SSP, and the shaded area shows a few examples of the GCM spread (the boxplots show the GCM spread at 2100 for each SSP and the two models). While the overall spread of these GCMs exceeds the differences between the model calibrated from historical and modern data, for individual GCMs (the minimum and maximum of GCM spread is shown by the shaded region of Fig. 4c,d) we do find significant differences in mass loss between the models calibrated with historical and modern data.

p. 9, second paragraph: “on average, 22% less mass loss” -> not clear, referring to cumulative mass loss for mean of projections?

Yes, this is a mean difference in projected mass loss from 2000-2100 for the four SSP scenarios. We have added “..., *depending on the SSP scenario*, ...” to clarify this.

p. 9 second paragraph: “Regardless of whether the historically-calibrated model predicts increased or decreased mass by 2100, both Kennicott and Root Glaciers show a clear acceleration in the rate of mass loss at the end of the 21st century,” Where is the increased mass? Seems like mass balance is negative throughout the 21st century? Please reformulate

this sentence, the statement is not clear.

The sentence has been amended to make this more clear that this is a comparison with the model calibrated with modern data, *“Regardless of whether the historically-calibrated model predicts increased or decreased mass by 2100 compared to the model calibrated with modern data, both Kennicott and Root Glaciers...”*

p. 9 second paragraph: “implying a more dramatic disequilibrium between the glacier and the climate than previously forecasted”. Cannot follow this. Assuming that previously forecasted refers to the “modern calibration”. Is the reduction in mass loss (fig 3e, f) with the modern calibration in the 21st century implying that the modern model shows a stronger disequilibrium earlier in the century, with earlier glacier retreat compensating for mass losses?

Yes, you are correct that previously forecasted refers to the “modern calibration”. Fig 4c-d show the modern calibration overestimated mass in 2100 on Kennicott Glacier and underestimated mass in 2100 on Root Glacier. However, for both glaciers, in 2100 the modern calibration underestimated mass balance (Fig. 4e-f)--so in both cases, we are showing a more negative mass balance in 2100.

p. 9, second paragraph: Regarding the discussion of parameter values, could you report these in the methods? Would be helpful to see what the differences are. Could the different calibration procedures used in the two experiments affect the results, not only the different calibration data? Also, the introductory paragraph 1 focuses on capturing glacier dynamics and does not mention that climate sensitivity is captured through parameter values, which may be more representative when using long-term records for calibration. It would be beneficial to the explanation if this is mentioned before and then relate it to the explanation of the findings.

Unfortunately, reporting the actual parameter values is not as practical as it may seem. While the historical data helps to constrain the model, the model is still overparameterized (primarily due to the fact that we calibrate all 3 model parameters simultaneously on glacier mass balance observations with large uncertainties). As such, we obtain a combination of “best” model parameter sets for the temperature bias, precipitation factor, and degree-day factor of snow. These parameter sets all estimate modeled mass change to be consistent with the multi-decadal mass balance observations, but vary drastically in their particular parameter values, as the parameters can compensate for each other to match overall mass loss and thinning trends. There are 89 parameter combinations for Kennicott Glacier and 22 for Root Glacier. We ultimately run the model to 2100 for each of these parameter sets (for each SSP and GCM combination) and ultimately take the mean as the model mass balance output. This methodology is specified in the methods.

We have added a figure to the supplemental material showing the difference between model parameters of both calibration methods:

“Fig. S6: PyGEM optimal precipitation factor (left), temperature bias (center), and degree-day factor of snow (right) distributions for historical and modern calibrations. Modern calibration follows a Bayesian framework to obtain probabilistic distributions of each parameter (Rounce et al., 2023). Historical calibration takes parameter sets that align with long-term mass balance records ($n=89$ for Kennicott Glacier, $n=22$ for Root Glacier), where histograms of the individual parameter distributions are shown.”

The difference in calibration procedures is precisely the difference between the models. The model itself is exactly the same, the only difference is how we calibrate and select the parameters. Our goal is to compare results from our calibration with the most recent, state-of-the-art projections (Rounce et al. 2023). From this standpoint, we feel that changing the calibration of the modern data would actually reduce the overall value of our comparison and thus the contribution of this study.

Specifically, the use of historical data constrains feasible parameter combinations (i.e., those that would match the 2000-2019 mass balance data) by more than 99% (i.e., less than 1% agree with the historical changes). The fact that they would otherwise agree with the 2000-2019 mass balance data explicitly highlights the overparameterization issues that plague all existing models. The study by Rounce et al. (2023), which builds on the calibration routine developed by Rounce et al. (2020), explicitly addresses this overparameterization issue by using an empirical Bayesian framework. This is well described in both publications and thus beyond the scope of this study to describe it in detail again; however, it is noteworthy here to highlight that we’re using the state-of-the-art Bayesian calibration framework for large-scale glacier evolution models and want to compare our results against those data.

Regarding the climate sensitivity being captured by the historical calibration, we have added a sentence in the historical calibration methods section, *“By utilizing the long-term mass balance record for calibration, these parameter combinations are an inherent improvement over the model calibration with only data from the 21st century as they capture climate sensitivity, which may impact calibration over shorter timescales (e.g., 2000-2019).”*

Additional Fig. of surface profiles: Please add a figure in the methods showing the modelled glacier profile with the two calibration experiments from 1940-2020, together

with the known surface height in 1957, 1978, 2004 and 2012 in areas where this is available. It would help to validate the glacier geometry evolution and show that the dynamic state of the glacier is better captured using the historical calibration, which you hypothesise in the first paragraph but do not comment on later.

Representing the modeled glacier profile with a DEM profile is difficult due to elevation-band flowlines used in the model (<https://docs.oggm.org/en/latest/flowlines.html>). These flowlines do not conserve length, and thus are only approximate, so that they cannot be plotted on a map anymore.

While we cannot map elevation profiles, we have included a figure that compares the mean elevation change rate for elevation bins on Kennicott and Root Glaciers of the historically- and modern-calibrated model with the surface elevation change observations from DEMs. This is referenced in the methods, “*The difference in elevation change rates between observations and the models calibrated with historical and modern data shows reduced bias when using historical data for calibration (Fig. S7).*” The supplemental figure is as follows:

“Fig. S7: Elevation-binned surface elevation change from DEM observations (left) and elevation change difference between geodetic and modeled estimates (right) for Kennicott (top) and Root (bottom) Glaciers. Results are shown from 1957-1978, 1978-2004, and 2000-2019. Model calibration with historical data shows less bias on elevation change rates compared to observations, including from 2000-2019.”

Limitations: Please comment on the limitations of the modelling here or in the methods. For example, both models use constant parameter values over the historical and future time

periods, while melt-factors have been shown by several studies to not be constant in time and not necessarily representative of future conditions, e.g. Huss et al. (2009) and Ismail et al. (2023). Do you think the parameters calibrated with historical or modern data are more representative for the future climate? The historical model clearly captures climate sensitivity better than the modern model in the historical period, but could the historical model underestimate future melt since parameter values are more representative of the historical climate rather than the future one? Please comment on this in the main text or in methods.

We have included a discussion on these limitations of the modeling in the methods. As you state, certain parameters that are held constant in the model actually do change over time, which is a limitation. However, we have no way to address this without comprehensive transient parameterizations of the model parameters. This requires specific datasets (over appropriate timescales) for calibrating each model parameter individually, which do not exist, and is well beyond the scope of this study. The previous studies that show this (e.g., Huss et al. 2009) ultimately assume that their climate forcing is accurate (which is known to be highly uncertain especially further back in time) thereby allowing them to change one parameter to fit observations over time. Hence, this remains a large overparameterization issue and we can not accurately estimate how model parameters have and will change over time.

We also do not feel it is particularly useful to discuss all of these intricacies in this paper—which focused more on model calibration and application than development—considering that the particular studies mentioned in this comment highlight limitations associated with projections from any temperature-index model. Nonetheless, we have highlighted key model limitations, including model parameters changing over time in a “Model limitations” section.

The added “Model limitations” section reads:

“On Kennicott and Root Glacier, PyGEM offers significant advancements over other large-scale glacier evolution models through a mass balance scheme that accounts for sub-debris melt using state-of-the-art debris thickness estimates and subsequent melt enhancement factors (Rounce et al., 2021). However, PyGEM does not explicitly account for the transient state of debris cover over time and the corresponding nonlinear feedback on sub-debris melt. Furthermore, the model does not account for proglacial lake evolution and frontal ablation on Kennicott Glacier, which could increase terminus retreat especially where the glacier bed is overdeepened (Fig. S1). While the impact of proglacial lake evolution and frontal ablation can affect model results on decadal timescales (Anderson, B. et al., 2021), this ultimately is glacier specific and will have diminishing impact on projected glacier mass as the glacier retreats towards higher elevations (e.g., Trüssel et al., 2015). Other potentially impactful factors on modeling surface melt include ice cliff, supraglacial pond, and surface relief feedbacks (Anderson et al., 2021b) or model parameters changing over time (e.g., Huss et al., 2009; Compagno et al., 2022; Ismail et al., 2023), although incorporating these requires the development of new

model parameterizations which is beyond the scope of this study.”

Debris cover: Since debris cover is seemingly important for the ablation area of Kennicott glacier, an explanation of how PyGEM handles debris cover needs to be included in the methods, and possibly commented on as part of limitations or discussion. Orthophotos show an increasingly large area of debris cover that has significant impacts on melt and glacier dynamics, do you expect these mechanisms to be represented in the model? If not, how does that affect past and future projections?

We have added this to the model limitations and note that a constant debris cover area is a common assumption in most models (e.g., Anderson et al. 2021; Rounce et al. 2023), although some models do consider this assuming or calibrating expansion rates (e.g., Rowan et al. 2015; Compagno et al. 2022). The model used here accounts for debris cover using sub-debris melt enhancement factors based on debris thickness estimates. However, PyGEM does not account for transient debris cover or thickness. We note that accounting for this would primarily modify the melt rates at the debris-clean ice interface. Thicker debris (> 5 cm) will already suppress the sub-debris melt and thus we'd expect it to have very little influence on the projected mass change. Hence, given the simplicity of the modeled mass balance (i.e., a temperature index model with elevation bins), we feel this is a reasonable assumption to include. [see above]

Rowan et al., 2015: doi.org/10.1002/2013JF003009

Compagno et al., 2022: doi.org/10.5194/tc-16-1697-2022

Fig. 4: The consensus mass estimates shown in Fig. 4a and b. Which densities were used to convert Farinotti et al 2019 volume to mass? Same as used in Hugonnet 2021? It would be useful to add error bars to these estimates based on uncertainty in the volume-to-mass conversion. I would specify in the caption how the consensus estimate for 2020 is derived: “adjusted to 2020 using mass change rates from Hugonnet et al., 2021”. Add gridlines to the plots in Fig. 4 so it is easier to follow the discussion on different time periods.

The Farinotti volume is converted to mass with a density of 900 kg/m³ and the Hugonnet volume change is converted to mass with a density of 850 kg/m³. We chose this following convention for glacier-wide surface elevation changes. We have not included uncertainty bars as density is only one of many sources of uncertainty in both the ice thickness and elevation change data, and would thus not accurately represent the true uncertainty. Also, the conversion of ice thickness to mass aligns with the density used for model initialization (i.e., the mass in 1940), and changing the density would affect both the consensus mass in 2000 and the initialized mass in 1940. We feel it makes the most sense to keep these consistent.

We've amended the caption to specify “...using mass change rates from Hugonnet et al. (2021).” We have not added gridlines as this is a stylistic choice.

Fig. 4 caption: “using ERA5 reanalysis data”, the model is equally relevant here: “modelled using the Python Global Glacier Model with ERA5 climate reanalysis data”. “Results show all individual and the mean of optimal parameter sets that align with multi-decadal mass balance” -> this is not clear, please see in relation to comment on this in Methods (modelled using mean of parameter values from n=.. parameter sets, or mean of modelled mass balance

using n=.. parameter sets). Please specify what “Historical” and “Modern” calibration refers to in the caption. Specify (Hugonnet et al. 2021) after modern data from 2000-2019. Also, if understood correctly from the methods, the historical calibration also uses the modern data, please specify this.

Thank you, we’ve added “...using PyGEM with ERA5...” to the caption. We have added years “(1940-2023)” for the historical data to help clarify the difference in model calibrations. The Hugonnet et al. 2021 citation has been added. We defer to the comment in the Methods regarding confusion about optimal parameter sets.

Summary, p.10-11:

p.10: “In summary, ~85 years of observations show a clear shift from a likely near-equilibrium state prior to 1957, to ongoing and accelerating mass loss throughout the late 20th and into the 21st century.” -> The latter part is shown by modelling, not by observations. Please rewrite. And be more specific than observations: geodetic mass balance? Since this is a summary, mention again the names of the glaciers.

We feel that this is clearly implied (nobody can have observations of the future). Still, the key to our understanding of future changes *are* the historical observations. We’ve decided to keep “observations” broad here and elaborate in the following sentences (thinning, velocity, etc). We have added “on Kennicott and Root Glaciers” has been added.

p. 11: “Projections leveraging” -> Add “projections from a glacier evolution model”
Done.

p. 11: “and project this acceleration to continue through much of the 21st century—decades later than previous studies suggest” -> This is a bit unclear: “show a later onset of mass loss acceleration than previous studies suggest”?

Noted. We are pointing here to the longer phase of increasingly negative mass balance (Fig. 4e-f). Mass loss is still accelerating at present, it just accelerates a bit slower initially and peaks later.

p. 11: “other large glaciers (>50 km²) in the Wrangell Mountains” -> Reference?
Added Hugonnet et al. 2021 reference.

p. 11, last paragraph: State you contribution in this paragraph, e.g. “We have shown that such imagery can provide unprecedented insights into long-term glacier mass losses and their mechanisms and provide valuable data that improves projections of glacier change in both the near and distant future.”

Thank you for the suggestion, we have amended the statement so it reads, “*Once digitally scanned, we have shown that such images provide unprecedented insights into mechanisms of long-term glacier mass change and provide valuable data that improve projections of glacier change in both the near and distant future.*”

Methods:

p. 11, Methods: Consider adding a section header before sections “Historical imagery” and “Historical DEM generation”, in the same manner as the “Additional DEMs” header.

Here, it could also be useful for the reader to navigate if the DEM years are specified in the headers, e.g. “Additional DEMs (1938, 2004, 2012, 2023)”, or perhaps in subsections, e.g. “Washburn DEM (1938)”.

Noted. A “Historical DEMs” section header has been added. We have left out the years of DEMs in the section headers because they are all stated in the first sentence of each section and the stylistic choice not to include parenthesis in section headers.

p. 12, Methods, Historical DEM generation, paragraph 4: Reference to Fig. 1, add subplot numbers c, d, e.

Done.

p. 12, Methods, Historical DEM generation, paragraph 4: The statement “which constrained the final DEMs to the ablation areas of Kennicott and Root Glaciers” is somewhat confusing since the DEM extents in Fig. 1 c and e also partly cover the accumulation area. Do you mean that reliable elevation change estimates could only be determined from the ablation areas (Fig. 2 a-e)?

We have added a reference to Fig. 2a-e to make the DEM extent more clear.

L379. You need a larger buffer in the terminus area probably?

The buffer is primarily meant to account for past glacier extent along the sides of the glacier, not necessarily the terminus. At the terminus, our restriction based on the NLCD classes omit past glaciated terrain. In general, we don’t want to make the buffer too large and limit the stable terrain across the broader area and reduce the accuracy of co-registration.

p. 13, Methods, Aster DEM: “adjusted long-term (1978-2012) thinning rates” -> Consider specifying that these thinning rates are derived from differencing 1978 and 2012 DEMs since the 2012 DEM is not presented until the next paragraph.

Noted. At the moment, we have kept the current sentence.

p 13, Methods, National Park Service DEM: “from 1,250 images collected” -> specify how these images were collected.

We have added “aerial photographs” to the sentence, to add this information without getting into all the details pertaining to the exact setup, which can be found in the citation and in the data for the paper.

p. 14, Glacier outline digitization, paragraph 1: “Outlines were produced for [...] two time periods (historical and modern).” What does historical and modern refer to? In the rest of the manuscript, “modern” and “historical” refer to datasets used for model calibration. Please provide the years of the outlines here instead. Same for paragraph 2 of the same section where it is referred to “modern extent”. be specific on years with outlines and which outlines used.

Noted. These have been replaced with 1957 and 2023.

p. 14, Glacier outline digitization, paragraph 1: “Accumulation areas were excluded during outline digitization due to poor image contrast on snow and a lack of suitable ground-control points which encumbered DEM generation in these regions: above the

uppermost icefalls on Root Glacier, the Gates and LaChapelle tributaries, and the edge of the aerial photograph extent on the main branch of Kennicott Glacier (Fig. 1a).” Consider moving the description of the specific areas to the first time this is mentioned, under “Historical DEM generation”, last paragraph.

Noted. We feel this information is more relevant where it currently stands, and feel that the broader statement about not having DEM coverage of accumulation areas in the “Historical DEM generation” section is the adequate amount of information needed in that section.

p. 14, Glacier outline digitization, paragraph 1: “Nonetheless, our digitized outlines (Fig. 2g) represents over 95% of all mass loss on Kennicott and Root Glaciers from 2000-2019 (Hugonnet et al., 2021); we thus assume that mass loss calculations are representative of the entire area of Kennicott and Root Glaciers.” This statement/reasoning is not clear (please see comment on Fig. 2).

Added reference to Supplemental Text S1, which explains this.

p. 14 Glacier outline digitization, paragraph 1: “The terminus boundary outline includes Kennicott Glacier below its junction with Root Glacier, aligning in extent with the 1938 DEM coverage (Figs. 1a, 2g).” Which outline is referred to here? Do you mean that all outlines cover this part of the terminus? In general the descriptions on outlines are unclear. We have removed the sentence regarding the terminus here to avoid such confusion. This is already stated in the main text where appropriate, and doesn’t actually have anything to do with glacier outline digitization. The terminus boundary is primarily a choice in processing data and characterizing glacier changes in certain areas of the glacier.

p. 14, Glacier outline digitization, paragraph 2: “Calculations requiring glacier area depend on the considered elevation change time period to determine the boundary used to delineate valid pixels for the volume change (Eqn 1) and the area (e.g., Eqns 3 and 4).” This sentence is difficult to understand, consider reformulating.

This has been reformulated. “*Calculations that use glacier area require contemporaneous glacier boundaries with regards to the relevant dataset (e.g., Eqns 1 and 3).*”

p. 15, Volume, elevation and mass change: “Volume change rate ..”, “Mass change rate ..” and “Mean elevation change ..” Suggest to use “The” in front, e.g. “The volume change rate ..” to make this easier to read.

Done.

L401-405: it will impact results, show a comparison.

We unfortunately cannot show a comparison of what the results would look like with full area coverage. However, we appropriately account for not having “glacier-wide” mass balance for model calibration in the uncertainty.

L408-412. Be specific, why not use outlines for each year and period? Do you not have them? How does it impact the results?

We do not have outlines for each year because:

- 1938: oblique imagery is not suitable for outline digitization

- 1978: we use the 1957 outline. The 1978 imagery shows no substantial differences in glacier outline compared to 1957; differences are within the method uncertainty and thus should not affect results
- 2012: we use the 2023 DEM; differences between 2012 and 2023 outlines are negligible (only differences are choices between stagnant ice at the terminus that represent small overall areas) and are within the method uncertainty

For the 1978-2004 and 2004-2012 time periods, we use an average area from the 1978 and 2012 outlines and use the minimum and maximum as the uncertainty, so any impact on results should be captured by the uncertainty.

p. 15, Volume, elevation, and mass change: “A density of 900 kg m⁻³ was preferred to the commonly-used geodetic volume-to-mass conversion 850 kg m⁻³ (Huss 2013) because our study is constricted to ablation areas over long temporal spans, and thus not affected by snow or firn” -> Can be more clear. The term density conversion factor is better to use. This statement should be related to the choice of using p_{ice} in Eqn 2, rather than the choice of value for p_{ice} . Consider “The density of ice in Eqn 2 was used for volume-to-mass conversion since we consider mass changes over ablation areas where mass loss is limited to the melt of ice (rather than the combination of ice, firn and snow)”, or something similar.

We use the term “density” because, by definition, density is required to convert volume to mass. We’ve decided to keep the term “density” due to its physical meaning. The rest of the sentence is motivated by the choice of p_{ice} in Eqn 2, as we state “...because our study is constricted to ablation areas over long temporal spans, and thus is not affected by snow or firn.” We prefer the current sentence and have maintained it, as a stylistic choice.

p. 15, Volume, elevation, and mass change: “where the area A aligns with V the temporal span.” -> Suggest to write out the description of V_{dot} here to increase readability: “where the area A aligns with the time span of the volume change rate V_{dot} ”. Also, consider some additional explanation of which area you are referring to here before Eqn 3, e.g. “The mean elevation change rate within a given glacier boundary is the volume change rate divided by the area within the glacier boundary?”

Noted. We have added “*volume change rate V_{dot}* ”. We have not added “within the glacier boundary” to the prior sentence because it is strongly implied with the area and referenced in the “Glacier outline digitization” section as well. We feel adding this is somewhat redundant and lengthens the sentence without adding much.

p. 15, Volume, elevation, and mass change: “Mass balance is simply the specific mass change rate: b_{clim_dot} ” -> Why do you refer to the climatic mass balance rate here? Here, the considered mass balance rate is perhaps the climatic-basal mass balance rate because mass changes include both surface, internal and basal processes, unless you assume basal processes to be negligible (in which case that should be stated). A short explanation of the mass balance components that are accounted for should be added here,

in relation to the choice of notation.

Noted. We have simply removed this equation since it is not actually used in the main results and the climatic mass balance for flux zones is calculated via Eqn 5.

p. 15, Volume, elevation and mass change: The notation $M_{\dot{}}$ is used for the mass change rate. This is strictly the geodetic mass balance in mass units and should be denoted by $B_{\dot{}}$. It is not clear why the geodetic mass balance rate becomes the climatic mass balance in Eqn 4, see above comment.

Noted. We follow notation from previous literature regarding the mass change rate: Mannerfelt et al. 2022: doi.org/10.5194/tc-16-3249-2022

p. 15-16, Ice thickness and bed topography: This section describes ice thickness measurements from 2024 and derivation of the bedrock topography. Sections “Flux divergence and climatic mass balance” and “Driving stress” describe the use of ice thickness to compute flux divergence and driving stress and from the first sentences on p. 18 under “Driving stress”. It is so that ice thickness is computed for each of the DEM years by subtracting the bedrock topography described in section “Ice thickness and bed topography” from each surface DEM, providing several ice thickness datasets from which ice thickness is retrieved in sections “Flux divergence and climatic mass balance” and “Driving stress”? Suggest adding a short explanation at the end of the “Ice thickness and bed topography” section describing this procedure and reiterating which years (DEMs) this is done for. This explanation could be before the “Flux divergence and climatic mass balance” section since ice thickness is used here.

Yes, your understanding is correct. The bed topography is calculated for 2024 and then subtracted from each DEM to get ice thickness. We have added the sentence, “*Ice thickness was calculated for each time period by subtracting the bed elevation from the DEM.*”

Is the velocity representative of the period 1957-1962?

We expect it to be relatively representative, considering it is a 5-year mean and Kennicott and Root glaciers are not surging.

p. 17, Flux divergence and climatic mass balance: Again, it is not clear why the elevation changes are assumed to represent the climatic mass balance (and not the climatic-basal mass balance). In relation to Eqn 6, shouldn't this be the climatic mass balance of the flux zone, $b_{\text{clim,zone}_{\dot{}}}$?

Noted. We added “zone” to the notation. Regarding assumptions around climatic-basal mass balance, see comment earlier.

p. 17, Flux divergence and climatic mass balance: “This flux-gate method emulates previous literature (Anderson et al. 2021b) but is robust to specific flux gate placement, thus reducing bias and accurately capturing uncertainty in climatic mass balance results as it ensures that localized uncertainties in velocity, ice thickness, and surface elevation change are consistently accounted for regardless of particular flux gate positions.” Consider splitting this sentence in two, it is difficult to follow.

This has been split into two sentences.

P18. Miss values in uncertainty part.

We present uncertainty values with the results as they are reported in the main text and figures. We cannot effectively report uncertainty values in the methods given the journal length constraints, and feel that the uncertainty is most effectively conveyed with the results. This section simply serves to explain how we derive and calculate uncertainty for each parameter/value.

p. 18, Driving stress: Equation 7 is referenced differently from previous equations (Eqn 7 vs. 7).

Noted. This is fixed now.

p. 18, Driving stress and Extended Data Fig. 2: In Extended Data Fig. 2 it would be helpful to add the glacier outline from 1957/1978 in the subplots or at least specify which extents the change in driving stress is computed for in subplots f-k. I assume that the driving stress/change in driving stress is not plotted for areas with no ice? To my understanding the driving stress is computed over each glacier extent, but I assume that the change in driving stress is plotted for the minimum extent of the two years in f-k?

Noted. The caption of Extended Data Fig. 2 was amended per other comments (see response to a later comment about the figure specifically), which should clarify this. The figure uses the maximum outline (i.e., 1957) for all plots to show the full extent of changes over time. There is no real consequence to choosing the maximum outline, as driving stress simply drops to 0 on deglaciated terrain.

p. 19, Uncertainty assessments, Volume, elevation and mass change uncertainty: Eqn 10: Here $b_{\dot{}}$ is used for specific mass change rate instead of $b_{\text{clim}_{\dot{}}}$. Should be consistent with Eqn. 4 and 6.

Noted. This equation is simply removed per previous comments.

p. 19, Glacier evolution modelling: Please provide a short introduction to how PyGEM models mass balance and dynamics.

Noted. This brief introduction is in the main text, per previous comments.

p. 19, Glacier evolution modelling, Historical glacier evolution reconstruction: “The historical reconstruction requires an estimate of the glacier mass and area from 1940 to initialize the simulation.” Shouldn’t this rather be “estimate of the glacier geometry (ice thickness and extent)” rather than “mass and area”?

Noted. We have updated this following the recommendation. We have maintained “mass” instead of “ice thickness” because this is what the model actually requires.

p. 19, Glacier evolution modelling, Historical glacier evolution reconstruction: The explanations in this first paragraph are not entirely clear. Why was the consensus ice thickness estimate from Farinotti used? Due to the complete coverage over the glaciers? This should be clarified. Consider using “consensus ice thickness” rather than “ice volume”, since volume suggests a quantity integrated over the glacier. Please be more specific in relation to elevation change rates missing in accumulation areas and remind the

reader: “Since elevation change rates from 1957-2012 could not be reliably determined in the accumulation areas of Kennicott and Root Glaciers, we estimated elevation-changes from the 1957 U.S. ...” How are elevation changes in these areas determined from the 1957 map, could you explain? It is also not clear how the statement “Glacier-wide mass change from 1940-1957 was assumed to be negligible given no surface elevation change at the terminus” relates to the rest of the paragraph, was the 2000 ice thickness adjusted to 1957 and then no changes were assumed between 1940 and 1957? In that case it is a bit confusing that it is stated previously in the paragraph that the elevation change rates from 1957-2012 are used to adjust back to 1940. Also, is this last statement justified? Surface elevation changes at the terminus were positive 1938-1957 (same magnitudes, but opposite of 1957-1978), wouldn't you expect positive glacier-wide mass balance? Needs more justification and explanation of how this fits in with the reconstruction of the 1940s glacier.

Yes, it was used due to the complete coverage. We have specified, “*The multi-model product (Farinotti et al., 2019) was chosen due to its complete coverage and to facilitate comparisons with previous studies (i.e., Rounce et al., 2023).*”

The “consensus ice thickness” and the volume from it (“consensus ice volume”) are essentially the same, but what we are actually using is the volume (aggregated ice thickness * area) instead of the ice thickness (spatially-distributed). As such, we have kept “consensus ice volume”

The 1957 topographic map was a USGS contour map, which is spatially complete but has large uncertainties (these maps are alluded to in the introduction, discussed briefly in Supplemental Text S2, and used in the study Das et al., 2014, for example). A comparison of this map and our 1957 DEM is shown in Fig. S5, which we've now referenced here.

Yes, your understanding is correct. Ice volume was corrected to 1957 based on the surface elevation change, and then we assumed no change to 1940. Note, we only need to initialize with glacier volume, not spatially-distributed ice thickness. As such, we believe assuming negligible glacier-wide changes is justified. Due to the large uncertainties in the 1938 DEM, we cannot definitely say the terminus experienced mass gain during this period. Any mass gain/loss from 1940-1957 we would expect to be small in comparison to the total ice volume, and thus not affect model results much. Furthermore, we anticipate that small differences would be accounted for via the glacier retreat feedback (i.e., slightly more glacier mass in 1940 would ultimately lead to slightly more melt during the simulation period and end up with the same mass).

During calibration, we incorporate large uncertainties in the mass balance from 1940-1957 due to the limited coverage/uncertainty in the DEM: we only constrain the mass balance from 1940-1957 to be less negative than that from 1957-1978.

p. 20, Glacier evolution modelling, Historical glacier evolution reconstruction: Regarding the calibration data used (ice thickness change from DEM differencing 1957-1978, 1978-2004, 2000-2019). Why were these periods chosen? For example, why not use 2004-2012 and 2012-2023? I assume the 2000-2019 data is Hugonnet et al. (2019), please specify.

Yes, the 2000-2019 data used is the Hugonnet et al. data. We made this decision to emphasize the added value of historical observations. We wanted data from the 21st century to be controlled, and not a possible source of discrepancy between the results. As such, our “historical calibration” includes the same present-day data in addition to the historical data (1940-1957, 1957-1978, and 1978-2004).

p. 20, Glacier evolution modelling, Historical glacier evolution reconstruction: Last sentence in paragraph 2: not clear what mean refers to here. Is it the mean of each parameter value from the n=89 and n=22 combinations, or are you referring to the mean of the modelled mass change as the best estimate of historical glacier evolution?

Yes, the mean mass change (output) of the 89 simulations on Kennicott and 22 simulations on Root. We have added “...from which the mean modeled mass change is used...” to clarify this.

p. 20, Glacier evolution modelling, Future glacier evolution: “each model run had 48 individual realizations” -> it is not clear which parameter values are used here. It seems from “using the calibrated sets of model parameters” that the model is run 89x48 and 22x48 times per glacier, or are future simulations run with mean parameter values (ref previous comment)?

Yes, your intuition is correct. The model was run 22x48 and 89x48 times for Root and Kennicott glaciers, respectively.

Extended data:

Extended Fig. 1: The caption to Extended Data Fig. 1 states: “The ice thickness from this study is derived using the 2012 IFSAR DEM and corrected with elevation change from Hugonnet et al. (2021) to adjust (c) to the year 2000 to match (d) and (e).” This is confusing. From section “Flux divergence and climatic mass balance” and “Driving stress” it seems that ice thickness is computed using the bed topography and each surface DEM. From the caption it seems that only ice thickness in 2012 is estimated? Assuming this is only shown as an example/for comparison, this should be stated. The figure caption should state clearly why it is adjusted to 2000 (not clear that this is the representative year for the other estimates). Reference to a comparison between these products in the main or method text seems to be missing. Would suggest adding a sentence about the comparison in this figure somewhere in the methods where you state the purpose and interpretation of the comparison.

Noted; we apologize for the confusion. Your interpretation is correct: ice thickness is calculated using bed topography and each surface DEM, and that this correction described in the figure caption here is for the purposes of comparison to existing products.

Both the Millan et al. (2022) and Farinotti et al. (2019) state a reference year of 2000 for the ice thickness products. This choice is largely motivated by global-scale processing of ice thickness using Randolph Glacier Inventory (RGI) glacier outlines, which have a reference year ~2000. We agree that these reference years may not truly reflect the data itself (firstly, because Farinotti et al. (2019) is a ‘consensus’ estimate that merges a variety of products, and secondly, because Millan et al. (2022) is inverted from surface velocity observations

~2018), but given the jumble of datasets and dates associated with them that are used to derive these ice thickness products, it makes the most sense to simply use the year 2000. Ultimately, the relatively minor surface changes from 2000-2012 compared to full ice column thickness do not alter the takeaways or conclusions of this comparison.

Extended Data Fig. 2: Please specify in the caption the outlines plotted in c and d - 1957? Noted. The following has been added to the caption, “*The 1957 glacier outline is used (a-k) to show changes over all ice-covered parts of the terrain since 1957.*”

Extended Data Table 1: Would suggest changing the order of the columns to “Product, Year, Source, Data” to increase readability (helps reader locate product). Also, suggest adding a new column with a letter code to differentiate between the products derived or used in the study, instead of using different fonts. Noted—these changes have been made.

Citations

Check the order of citations. These are not in alphabetical order or ordered by year, e.g. two places in p2, Introduction, paragraph 3. Noted. This has been fixed.

A reference seems also to be missing from the suppl material. We are not sure exactly what or where you are referring to. We feel Supplemental Text S1 (now S2) references all relevant studies related to Kennicott and Root glaciers. Supplemental Text S2 (now S3) appropriately references software, and describes pre-processing steps using these software. Supplemental Text S3 (now S4) describes processing methods and additional steps we took to calibrate PyGEM.

Perhaps you are referring to the McNabb et al. (2019) citation in Fig. S4 caption which is not in the reference list of this section? We excluded McNabb et al. (2019) from this reference list as the study is already cited in the reference list for the main text.

References

Huss, M., M. Funk, and A. Ohmura (2009), Strong Alpine glacier melt in the 1940s due to enhanced solar radiation, *Geophys. Res. Lett.*, 36, L23501, doi:10.1029/2009GL040789.

Ismail, M. F., Bogacki, W., Disse, M., Schäfer, M., and Kirschbauer, L.(2023) Estimating degree-day factors of snow based on energy flux components, *The Cryosphere*, 17, 211–231, <https://doi.org/10.5194/tc-17-211-2023.dem2023>

We would like to thank both reviewers for their valuable insights and detailed comments. Their input has helped refine and improve the manuscript. Below, we provide responses to each point raised, and believe the resulting revisions have meaningfully enhanced the strength of the work.

Reviewer #2 (Remarks to the Author):

The authors provide a detailed and compelling story of the mass loss and velocity changes of Kennicott and Root Glacier utilizing data that spans an 85 year time period. To develop this length of record required utilization of older aerial images with techniques that can be applied elsewhere. Below are several specific comments that I recommend the authors consider adopting. The paper is acceptable for publication whether the authors adopt these suggests or not.

We appreciate the continued time you've put into helping improve this manuscript. We've responded to the specific comments below.

Specific Comments:

106: When was this notably thin ice observed, 2023? How thick was it in 1957?

This comment refers to ice thickness measurements from 2024. We've added "...in 2024..." to the sentence to make this more clear. This part of the glacier has lost ~115 m from 1957-2023.

164: Worth emphasizing that this reduction in velocity above the shelf is indicative of reduced volume flux which in turn leads to decoupling and stagnation below the shelf.

Thank you for pointing this out: we agree that these are important dots to connect. A sentence in this paragraph reads, "Below this feature, decreased ice velocity (from ~26 m yr⁻¹ in 1960 to <5 m yr⁻¹ in 2017) and corresponding decrease in ice flux due to a rising equilibrium-line altitude leads to drastic thinning, which has led to substantial reductions in driving stress into the lowest 4 km of Kennicott Glacier." We have not added further emphasis on the reduction in volume flux to avoid repetition.

179: "...regardless of surface melt". This is not true overall, but can be in a qualified sense. The reduced volume flux is to a large part derived from surface melt. It is true that below the shelf even if surface melt does not increase reduced volume flux will drive thinning.

Yes, you are correct. Surface melt drives the up-glacier thinning that subsequently leads to reduced ice flux at the terminus. We've specified this to make it more clear, such that the sentence now reads, "...regardless of surface melt at the terminus."

294: Reword for accuracy " In summary, observations over an ~85 period.."

The sentence has been reworded following this suggestion.

372: Are there other Washburn (1941) areas mapped where the same approach could be use?

Yes, there are a lot of other regions with images from the Washburn collection. Similar approaches to generate DEMs could likely be used, however, further investigation of these images on a site-by-site basis is required to comprehensively answer this question. The Washburn photos pose somewhat of a challenge due to their oblique nature, the limited number of photos taken, and the distance to the glacier/size of the glacier in the photos. Generating a DEM of the terminus area of Kennicott Glacier from the Washburn images is largely possible due to the highly heterogeneous, rough surface, and the large size of the glacier, which is not the case at many sites.

605: reword for clarity “utilizing the long-term geodetic mass balance record..”

The word “utilizing” has been replaced with “using” to improve clarity per the suggestion.

Reviewer #5 (Remarks to the Author):

The authors demonstrate a comprehensive study that integrates historical DEMs, modern satellite data, and ice-penetrating radar observations with climate-driven glacier model projections for Kennicott and Root Glaciers. The writing and figures in this manuscript are excellent. My main concern is that the DEMs carry a significant amount of uncertainty, which is not sufficiently accounted for and clarified in the manuscript. Does the lack of spatial coverage and level of uncertainty truly constrain your model projection?

We appreciate your feedback for this review, and are glad to hear that the manuscript reads well. DEM uncertainty is critical towards many of the analyses in this study, and we believe we have accurately quantified and carried these errors through in the data. A large source of uncertainty in mass change to constrain the modeled output are the unmapped accumulation areas, as you’ve noted. While this does introduce considerable uncertainty, the enhanced temporal record still vastly constrains the set of viable model parameters and subsequently the future projections, despite the uncertainty on total glacier volume changes over time. Furthermore, we use binned elevation change observations of the ablation area during model calibration, which does not rely on any assumptions of the accumulation area. We’ve responded in more detail to all of the comments below.

Detailed comments:

L29 Long-term records do not directly have far reaching consequences. Suggest revising to something like: "This highlights the importance of long-term glacier mass-loss records that help us better project far-reaching consequences of climate change related to sea level rise, water resources, and natural hazards, climate, and culture."

The sentence has been updated as suggested.

L38 Add and consider citation for Arendt et al. 2002 “Rapid Wastage of Alaska Glaciers and Their Contribution to Rising Sea Level” DOI: 10.1126/science.1072497

Thank you for the suggested reference. While we do not feel this citation directly fits the primary focus of this initial topic sentence, we have added it in the conclusion when referencing previous studies that have shown records of glacier mass change in Alaska.

L62 The statement “Only one study assesses changes in Alaska” using historical aerial photographs combined with structure-from-motion photogrammetry is not correct. Consider the work by the USGS in: O’Neel et al. 2019 “Reanalysis of the US Geological Survey Benchmark Glaciers: long-term insight into climate forcing of glacier mass balance”
doi:10.1017/jog.2019.66

Thank you for noting this distinction. We’ve included this study here and changed the sentence to read, “Few studies assess changes in Alaska...”

L63 Also consider the excellent early photogrammetric work by Robert Krimmel

Thank you for pointing out these works by Krimmel and others. It is truly impressive photogrammetric work, especially considering these were ~40 years ago. While we do not feel this work directly exemplifies the statements made ~L63, we’ve acknowledged the Krimmel and Rasmussen (1986) study ~L60.

- Krimmel, R.M., and Rasmussen, L.A., 1986, Using sequential photography to estimate ice velocity at the terminus of Columbia Glacier, Alaska: *Annals of Glaciology*, v. 8, p. 117-123.
- Krimmel, R.M., and Sikonia, W.G., 1986, Velocity and surface altitude of the lower Hubbard Glacier, Alaska, August 1978: U.S. Geological Survey Open-File Report 86-549, 20 p.
- Krimmel, R.M., 1987, Columbia Glacier, Alaska, photogrammetry data set 1981-1982 and 1984-1985: <https://doi.org/10.3133/ofr87219>
- Krimmel, R.M., 1992, Photogrammetric determination of surface altitude, velocity, and calving rate of Columbia Glacier, Alaska, 1983-91: U.S. Geological Survey Open-File Report 92-104, 72 p.

L128 Your study assumes that glacier mass loss captured by the DEMs accounts for 95% of the total mass change of both glaciers (Supplemental Text S1). However, the DEMs primarily cover the terminus and ablation area. The assumption that “thinning in the accumulation area from 1994–2013 converges to zero” at Kennicott and Root Glaciers is not directly substantiated by Larsen et al., 2015. Their study highlights that Alaskan glaciers are predominantly losing mass between 1994 and 2013 across all categories (land, lake-terminating, tidewater), but also points out significant spatial variability. The mean normalized hypsometric curve does converge to zero in their study (Figure 2 a), but also shows significant hypsometric variability for individual glaciers. I’m not sure this information fully supports the assumption for your entire 85 year long study period at these two specific glaciers. Additionally, your geodetic measurements used to constrain the mass loss have very large uncertainty (e.g. 79.8 (m) error for GCPs in 1938, as per

Supplementary Table S1). Does the lack of spatial coverage and level of uncertainty truly help constrain your model projection?

The claim that the mass loss captured by the DEMs produced in this study account for 95% of total mass change is based on surface elevation change maps of the entire glacier (Hugonnet et al., 2021). These maps show that the spatial area covered by the DEMs in this study (since 1957) represent 95% of the total mass loss from 2000-2019 (see Supplemental Text S1).

When estimating the entire glacier volume in 1957, we use the U.S. Geological Survey 1957 topographic map (bias-adjusted with our 1957 DEM) to account accumulation area changes. A comparison between our 1957 DEM and the USGS 1957 topographic map is shown in Fig S5. In addition to glacier-wide mass change estimates (which have large uncertainties due to unmeasured accumulation areas), the model is calibrated with binned elevation change data of the ablation areas, which does not rely on any assumption of the accumulation area. Such data greatly reduces the number of feasible model parameter combinations, as some model parameter combinations yield reasonable glacier-wide mass change but unrealistic binned elevation change profiles. Altogether, the glacier-wide and elevation-binned data substantially constrain and thus improve model projections, despite large uncertainties (as indicated by the spread of values in Figs 4a,b being significantly different from the model not calibrated with historical data).

The 1938 DEM has substantially larger uncertainties than the other DEMs due to the nature of the photographs (oblique) and limited stable terrain for ground control. While the reported error in ground-control points is large, DEM co-registration, in theory, minimizes these errors. We assess errors in the stable terrain after DEM co-registration (i.e., Extended Data Fig. 6) and comprehensively account for these errors in all calculations (hence large uncertainties for 1938-1957 data in Fig. 2, for example). We discuss your comment about the 1938 DEM spatially-autocorrelated error in a later comment. Overall, we account for large uncertainties in model runs (e.g., Fig. 4 a,b) and see a large spread of viable model outputs pre-1957, however, these still help to constrain the model over time.

L236-239 This is a long sentence that could be broken up, for example: “PyGEM estimates melt using a temperature-index model that accounts for sub-debris melt (Rounce et al. 2021), accumulation using temperature thresholds, and refreezing using mean annual air temperature. Simultaneously, it updates the glacier geometry annually using a flowline model based on the shallow-ice approximation (Maussion et al. 2019).”

Thank you for the suggestion, we agree that the sentence reads more smoothly now.

L337 Embedding EXIF information is not a requirement or objective of historical image preprocessing. I suggest stating something like: “Image preprocessing for historical imagery

ensures that aerial photographs are standardized to calibrated or fixed dimensions, which is required to extract precise elevation information from stereo view.”

Noted. we’ve removed the sentence on EXIF information and added the suggested correction, explicitly stating that these images are being standardized to fixed dimensions.

L342 Replace “Image EXIF metadata” with “Internal camera geometry data”.

Done.

L344 Replace “east-west and north-south” with “left-right and top-bottom”. Fiducial markers are referenced in the image plane, not cardinal directions.

Good point—done.

L345 Fig. S1 Which years do a,b,c,d, belong to? How was a definitive center point for the markers a and d selected in the image pixel grid?

The dates of these images are as follows: (a) 1938, (b) 1957, (c) 1978, and (d) 1948. We included a sample fiducial marker from an image in 1948 (which we did not end up using) to demonstrate another example of the types of fiducial markers, which we thought could be helpful to see for readers who may be interested in conducting similar studies. The 1962 imagery has the same fiducial markers as the 1957 imagery, so an example was not shown.

The points were selected as consistently as possible between images. While the base of the protrusion is technically the ‘official’ fiducial mark location for these images, we identified the tip of the protrusion for all images with fiducial marks shown in (a). This location was easiest to consistently identify between images, and the slightly reduced image extent by selecting the tip is not an important factor in this particular case.

L346 What kind of correction was applied? Did you apply any sort of affine transformation to calibrated values? Was any form of image enhancement attempted to increase dense cloud matches over the snow covered accumulation areas? This warrants at least some form of discussion given the lack of data in the accumulation area and emphasis on the value of historical image-derived DEMs.

The Scale Invariant Feature Transform (SIFT) algorithm used to identify tie points is able to detect ground information with great accuracy. Enhancing images helps us see ground features more clearly, but ultimately does not make any new information appear that was not already there. In this case, there simply are not enough features in the accumulation area to successfully identify tie points and subsequently obtain ground measurements for. We still attempted to enhance images to make any features in the accumulation area more visible, but the surfaces were too featureless or saturated to see or resolve any tie points.

All scanned air photos from film have different dimensions and/or different alignment. The correction applied around L346 is simply a geometric rectification to account for these inconsistencies resulting from digitally scanning film images. In essence, this correction aligns all scanned historical aerial photographs to a common internal coordinate system. The camera calibration was done with the Agisoft Metashape default Brown-Conrady model, as described in L349-354. While we do not explicitly apply transformations, the processing workflow implicitly performs affine transformations. Normalizing scanned images with fiducial marks creates a standard image geometry, which applies affine-like corrections to do so. Furthermore, the camera calibration and bundle adjustment in Metashape performs a non-linear optimization, which includes effects similar to affine transformation, as well as higher-order geometric corrections.

L352 What were the final reprojection errors and what level of error was deemed sufficient?

The reprojection error was 4.3 pixels for 1957, 0.63 pixels for 1962, and 1.2 pixels for 1978. We deemed this to be acceptable based on the fact that we are working with historical imagery (unknown internal camera parameters, film and scanning artifacts) and much of the poorly-contrasting parts of the images are shaded off-glacier terrain. Furthermore, since we expect large surface changes, we do not need extremely high (e.g., sub-meter) accuracy to be useful. Hence, these levels of errors were deemed sufficient, which we also retrospectively validate after performing DEM co-registration and uncertainty analysis (i.e., Extended Data Fig. 6).

L354 Did you manually or programmatically identify tie points? Did you manually remove spurious tie points?

Yes, tie points were determined programmatically with Agisoft Metashape and spurious points were removed manually. We've added this by stating, "...and spurious points were manually removed."

L369 Suggest naming this '1938 "Washburn" DEM' or reorganizing the sections in to "DEMs from vertical images" and "DEMs from oblique images".

The section header has been renamed following the suggestion: 1938 "Washburn" DEM.

L420 It would be useful to see a DEM difference map before and after the first-order polynomial fit to assess uncertainty and determine the nature of spatially correlated error that was modeled and removed.

The following plot is a difference between the original and detrended 1938 DEMs. Note that the plot shows off-glacier terrain, as well as terrain farther up-glacier than is used in the analysis. The detrending removes a terrain bias that is largely a function of the distance from the camera, which was expected from oblique imagery. We also assessed changes with a 2nd

order polynomial, but did not observe any substantial differences with the results using the 1st order polynomial.

L546 The appropriate citation here is Höhle and Höhle 2009 “Accuracy assessment of digital elevation models by means of robust statistical methods”
<https://doi.org/10.1016/j.isprsjprs.2009.02.003>

Thank you for providing this—the reference has been updated.

L548 Extended Data Fig 6 shows random, systematic, and significant spatially correlated error. The Distance and Variance in the variograms should be shown with a log or jump scale on the x-axis to better show spatially correlated error at various distances.

Extended Data Fig. 6 has been adjusted to show distance of spatially autocorrelated error on a logarithmic scale.

While you calculate the distance and variance of the spatially correlated error, it does not appear these are considered when propagating your uncertainty to volume and mass change estimates. For example, how are the ~200 m variance over ~100 m distance in 1938 taken in to account?

We use the NMAD to estimate uncertainty in elevation change from the 1938 DEM. The spatially autocorrelated error ~200-300 m at distances in the range of 100-1000 m is still much smaller than the domain area (Kennicott ‘terminus’ area of ~17.8 km²) of the 1938 DEM. As such, this variance would not increase error in mean elevation difference over the terminus area, which is ultimately used to derive mass change estimates.

If we use equations for mean elevation difference uncertainty that account for such variance (e.g., following from Rolstad et al., 2009 – equations (13) and (14) in this case), the mean

elevation difference uncertainty would be smaller than simply using the NMAD of stable terrain. We opt to maintain the current formulation, noting that it is a more conservative estimate of DEM difference uncertainty, especially considering the limited amount of stable terrain available in the 1938 DEM.

This is similarly the case for the other DEMs: incorporating the spatially-correlated error into mass change calculations following similar methods as previous literature would reduce the uncertainty. We show these to demonstrate the distance of spatial correlation between errors and that these calculations were considered, but do not use them in error calculations following the results that show error is not correlated over large distances.

- Rolstad, C., T. Haug, and B. Denby. “Spatially Integrated Geodetic Glacier Mass Balance and Its Uncertainty Based on Geostatistical Analysis: Application to the Western Svartisen Ice Cap, Norway.” *Journal of Glaciology* 55, no. 192 (2009): 666–80. <https://doi.org/10.3189/002214309789470950>.

The difference maps in Extended Data Fig 6 are barely visible which does not make them very useful. I suggest either a) giving Original-2004 its own color bar, b) using a log scale for the Elevation difference color map, or c) removing Original entirely. If you choose option c), you can focus this figure on highlighting random, systematic, and spatially correlated error in the final surfaces. Then have a dedicated figure for 1938 comparing the original before and after polynomial surface correction + co-registration, as well as a dedicated figure for 2004 before and after co-registration. Again, my biggest issue is that the uncertainties of these measurements are so large that I don't see how they act as effective constraints on the glacier evolution model.

We have decided to keep the original images and statistics in addition to the co-registered images and statistics in the figure because we feel it is important to demonstrate the before and after of co-registration. We do want to highlight that the DEMs produced from structure-from-motion photogrammetry with manually-selected ground control points show very limited bias to the reference DEM. The 2004 DEM, derived as an ASTER product, has large errors that co-registration reduces. We feel that this is useful to highlight, and have thus kept these data in the figure. The colorbar in this figure has been updated following the later comment, but we feel that using a logarithmic scale for the colorbar and elevation change warps the data in a way that makes the figure more difficult to interpret.

Generally speaking, we do not believe that the uncertainties are particularly large compared to the observed glacier thinning signal, especially over relatively large domain sizes. The 1938 DEM indeed has large uncertainties, however, this DEM shows at most, accounting for uncertainty, very minor thinning at the terminus (likely potential thickening), which is a useful constraint. The 2004 DEM has NMAD values over stable terrain of 7.25 m and very minor bias (median value of 0.48 m) over stable terrain. Calculating volume and mass

changes over large areas (10s to 100s km²), spatially-uncorrelated or short-range correlated noise will offset (which we demonstrate with the variograms) and only bias plays a significant role: hence, this is actually very small compared to average thinning ~10-20 m per 10-20 year time span.

L552 and 554 It is odd to mention specific function names here, but not elsewhere. I recommend removing this detail and keeping it conceptual to be consistent with other sections.

Noted. These have both been removed such that it now reads, “We sampled five unique empirical variograms with a sample size of 10,000 using the xDEM Python package (xDEM contributors, 2021; Hugonnet et al., 2022). We subsequently fit a short-range (maximum pixel distance of 500 m) and long-range (no limit) spherical variogram model to the mean empirical variogram.”

If you are using a white background you need to choose a colormap that does not include white. This applies to Extended Data Fig. 2, Extended Data Fig. 3d, and Extended Data Fig. 6.

- Extended Data Fig. 6: updated colorbar to exclude white
- Extended Data Fig. 3d: background changed to off-white color
- Extended Data Fig. 2: updated colorbar to exclude white

We would like to thank the reviewer for their valuable insights and detailed comments. Below, we provide responses to each point raised.

Reviewer #5 (Remarks to the Author):

The authors have addressed the concerns raised during initial review. I appreciate their clarification regarding the spread of values in Figures 4a and 4b, which differs significantly from the model not calibrated with historical data. This strengthens my confidence that the inclusion of these historical data meaningfully constrains and improves the model projections, despite the high levels of DEM uncertainty.

The comprehensive work presented in this study - bridging long-term historical observations with modern data and model calibration - is excellent and I recommend it for publication. Great, we are glad that our response to the initial reviews was clear and the changes were effective.

Minor comments:

L31 Delete first “and”

Done.

L79 You state that “To our knowledge, we present the first study that uses geodetic mass balance from historical DEMs to calibrate a glacier evolution model.” Are you aware of any glacier modeling studies that calibrate against long-term records of glaciological field mass-balance measurements, which in turn were calibrated with historical geodetic mass balance? If so, would this change your statement?

The statement has changed due to editorial policies in *Nature Communications* regarding using this type of language with reference to previous work. It now states, “We demonstrate the value of leveraging geodetic mass balance from historical DEMs to calibrate a glacier evolution model.”

Still, this is a very interesting point being raised. There are a few studies that use long-term glaciological measurements for calibration (e.g., Marzeion et al., 2012; Giesen and Oerlemans, 2013; Zekollari et al., 2019; Schuster et al., 2023). While we are not sure of exactly what calibrations are applied to all glacier inventories part of the World Glacier Monitoring Service (which these studies rely on for calibration), the long-term glaciological mass balance sites in Alaska are indeed calibrated with geodetic data (O’Neel et al., 2019). Still, these measurements differ from the direct use of geodetic mass balance measurements for model calibration. The nature of these datasets and associated uncertainties are categorically different from that of geodetic mass balance observations. As such, we believe this statement to still be true. Furthermore, the original statement emphasizes and brings attention to such historical datasets and demonstrates their use for model calibration; we hope that such historical datasets are

processed and eventually integrated into large-scale model calibration schemes to improve projections.

- Marzeion, B., Jarosch, A. H. & Hofer, M. Past and future sea-level change from the surface mass balance of glaciers. *The Cryosphere* **6**, 1295–1322 (2012).
- Zekollari, H., Huss, M. & Farinotti, D. Modelling the future evolution of glaciers in the European Alps under the EURO-CORDEX RCM ensemble. *The Cryosphere* **13**, 1125–1146 (2019).
- Schuster, L., Rounce, D. R. & Maussion, F. Glacier projections sensitivity to temperature-index model choices and calibration strategies. *Ann. Glaciol.* **64**, 293–308 (2023).
- O’Neel, S. et al. Reanalysis of the US Geological Survey Benchmark Glaciers: long-term insight into climate forcing of glacier mass balance. *Journal of Glaciology* **65**, 850–866 (2019).

review NCOMMS-24-80605

Albin Wells and co-authors: An 85-year record of glacier change and impacts on future projections for Kennicott and Root Glaciers, Alaska

General comments:

This is a solid and rich work on changes of two Alaskan glaciers including new data sets and modelling. They present noteworthy results on glacier changes and reconstruct glacier changes and velocity records using historical aerial photographs. We find that references to previous literature could be added.

Assessment of “glacier-wide” changes throughout the manuscript. This requires further justification and explanation. It is not a glacier wide average when not all of a glacier is covered. If changes are small in upper parts, but largely negative in other parts, inclusion of upper parts would influence the average value per glacier. We question the use of glacier wide averages as their glacier mapping does not cover all of the glaciers, quite a big part of the upper regions is missing.

The authors claim that they have a 85 year record, we expected annual series with such wording, not some subperiods of surface elevation maps. We encourage to revisit the title and how the use of record and glacier-wide averages are presented throughout the document. See also detailed comments below. The figures are of varying quality and could be harmonised more.

The study provides several interesting and valuable datasets related to different time periods and spatial extents, and contains different components and analysis. This requires a high level of precision in formulations when presenting and discussion of results, and the manuscript would benefit from more specific formulations several places (please see Detailed comments). For example, please be specific about which data, locations and periods is referred to and avoid using “historical” and “modern” instead of specific periods (the use of these terms is ok in relation to model calibration data since it is clearly defined here).

Brief method explanations are needed in several places in the main text to give the reader an overview of how the different results/data are produced. The main text before the methods section needs to be written such that the reader can read this text in its entirety and understand what is done (at a high level) and then choose to dive into the methods for details. A single sentence or amendment of a sentence is often sufficient, such as the first sentence in the section “Historical record reveals periods of equilibrium and accelerated mass loss”. Please see specific suggestions in the detailed comments.

Reference to findings from other studies is (partly) missing, e.g. are the temporal patterns of mass loss in line with those on other glaciers in Alaska, are there examples of other glaciers where the same mechanisms behind wastage of the terminus is found? Are there relevant previous findings on Kennicott and Root?

All data links in data availability are working, but it would be helpful if the authors had provided more direct links to the datasets where possible. It is not straightforward to find them.

The PyGEM model code and documentation is available on GitHub (github.com/drounce/PyGEM). OK Code is available at https://github.com/albinwwells/past_and_future_mb. Seems fine. Model code - no comments.

Other comments:

As the manuscript did not provide line numbers when we first looked at it, we copied the text in the manuscript for reference. It is also challenging when as in this case two persons (senior +ECR) are reviewing the manuscript. We received line numbers after partly finished the review, so the feedback will be a mixture of copied text and linenumber. Please next time provide line numbers. Some minor comments are therefore not included as it was too cumbersome to refer to them.

We are not used to the format of Nature Com and find it inconvenient in the reading to have it so divided. First results then methods then more on methods in supplementary. e.g. both in methods and in suppl the density conversion factor is mentioned, but only in the methods it is explained why this factor is used. The uncertainty section does not give the uncertainties, could be given where they are explained.

Detailed comments

Title: is the title representative? An 85-year record gives the impression of annual measurements, could you rather state the period in the title?

Abstract:

p1, Abstract: "results prior to 1957 show two decades of near equilibrium followed" -> hard to follow which time period this refers to, perhaps change to "results show two decades of near equilibrium prior to 1957, followed by"

p1, Abstract: "These results highlight the unique insights that historical aerial photographs provide into past and future changes" -> Be more specific about which changes.

L19. Not sure that it is unique that one uses historical aerial photographs to assess changes, I would remove unique here. The authors could in introduction refer to studies using aerial photographs to assess historical change, there are several from Norway, the European Alps etc. and other regions.

The abstract is rather general, we would prefer more results and facts.

Introduction:

p2. first paragraph. however, there are longer time series available from glaciers that have more details and are more accurate, such as glaciological mass balance. This information could be added.

p2, Introduction, paragraph 1: "the recent ubiquity" -> the word "ubiquity" may be difficult for an international readership, consider replacing it with a more accessible word.

p2, Introduction, paragraph 1: "has enabled two decades" -> specify the time period after this or in relation to the citation, e.g. (2000-2019; Hugonnet et al., 2021).

p2, Introduction, paragraph 1: "our ability to model future glacier change is shrouded with uncertainty due to the relatively short time periods of these continuous observations". This is only one of many factors contributing to the uncertainty in model projections, what about e.g. uncertainties in future projections, accuracy and resolution of climate data, or parameterization of models? Consider rephrasing such that it is clear that this is not the only factor. I would also suggest adding a reference to this statement.

p2, Introduction, paragraph 1 or 2: Consider specifying in paragraph 1 or 2 how the mass loss estimates are derived, i.e. that they represent geodetic mass balance derived from differencing DEMs. This will make it more clear in paragraph 3 how historical DEMs can be used to extend mass change records.

p2, Introduction, paragraph 2: Redundant semicolon “;” after “Marzeion et al., 2012”. Could add a more recent reference here also, e.g. Zekollari, H., Huss, M., and Farinotti, D.: Modelling the future evolution of glaciers in the European Alps under the EURO-CORDEX RCM ensemble, The Cryosphere, 13, 1125–1146, <https://doi.org/10.5194/tc-13-1125-2019>, 2019.

p2, Introduction, paragraph 3. The first sentence could be shortened to: Historical aerial photographs provide unique opportunities to derive digital elevation models (DEMs) and orthophotos to assess glacier changes and go well back prior to the satellite era. And here you could cite some of these studies, e.g. ...

Aerial photographs have been used for a long time in glaciology. There are many earlier references than this. I don't understand why you mention three regions, there are also other regions monitored, I would drop it and rather join the references. thus, quantify glacier changes (e.g.....;.....; ...). You could also add they are used to validate glaciological mass balance records.

p2, Introduction, paragraph 3: “before Landsat and» Add a reference year to indicate which period “before Landsat” refers to.

p2, Introduction, paragraph 3: “although these data often have large uncertainties (~15 to 45 m)” It is not clear what these uncertainties refer to, is it the elevation differences derived from these maps? Consider being more specific than “data”.

p3, Introduction, paragraph 4: , we use a suite of datasets ... this sentence is very long and hard to read.

avoid and/or use just or

L67 rather write ..quantify changes since 1938 for two...

p2, Introduction, paragraph 4: “We produce and/or use” -> This study makes significant contributions in processing and providing historical datasets. The contributions of the study would be better highlighted if this statement was split in two, first stating which new datasets are produced in this study and then a second statement on which sources are “only” used, e.g. “We produce several new datasets, including [...]. We complement this data with [...] to estimate spatially-distributed changes in elevation and glacier dynamics over time.”

p2, Introduction, paragraph 4: Not clear what is meant by “modern sources”, please be more specific.

p3, Introduction, paragraph 4: “measure spatially-distributed changes” -> “estimate spatially-distributed changes”?

L75 continuous elevation change. do you mean since you have digital DEM? I am not sure this adds anything.

p3, Introduction, paragraph 4: “revealing considerable changes in future estimates” -> Not clear what is compared here. Suggest to be more specific, e.g. “revealing considerable differences between future estimates compared to model calibration using present day mass changes”.

Introduction, Fig. 1. We suggest some modification of the figure. Use smaller font on a-g. It is a bit confusing that the inset (g) with zoom-in on bed topography is marked 1938 on the overview map (a), and that the image in inset (b) shows a historical photograph of not only the terminus. Why not show the Washburn 1938 DEM (Methods, p. 13) of the terminus in Fig. 1b? This would make the figure more consistent with respect to how the other DEMs and DEM extents are shown. Also, please add the year of the glacier outline in the caption in relation to “present-day glacier outlines”, be specific.

The figure would be easier to read with less panels.

Historical record reveals periods of equilibrium and accelerated mass loss:

p.4, first paragraph: “assess changes across” -> being specific about what types of changes are assessed helps the reader to know what to expect in this section, e.g. “assess changes in glacier geometry and associated mass losses across”

p.4, first paragraph: “including a specific focus on the terminus of Kennicott Glacier—delineated to be below Root Glacier—to extend the record back to 1938 (Fig. 2g).” The part “-delineated to be” can be removed. “-to extend the record back to 1938” is a bit confusing because it is not clear which “record” is referred to, suggest to reformulate: “including a specific focus on the terminus of Kennicott Glacier (below Root Glacier), where DEM coverage extends back to 1938 (Fig. 2g).”

p4. first paragraph after headline, 2g is it meant 1g?

A bit confusing that results are here prior to methods - but this is perhaps the journal style.

L96 66-year period of mass loss. you mean mass loss recorded over this 66 year period, you don't have results for all years.

p4, first paragraph: Statement “Elevation change observations reveal Kennicott and Root Glaciers were roughly in a state of equilibrium from 1938 to 1957, since which mass loss has accelerated (Fig. 2).” I don't agree with this statement. If you consider 1938-1957 (Fig. 2a) to show near equilibrium due to limited mass change/slightly positive then I would also say that 1957-1978 (Fig. 2b) also shows near equilibrium (limited mass change/slightly negative). Also, 1938-1957 shows mass gains on the lower part of the tongue, if the 1938 DEM covered larger parts of the ablation area, wouldn't you expect to see even more positive rates here such that this may have been a period of mass gains (larger than the losses 1957-1978)?

p4, first paragraph: “30% decrease in Kennicott Glacier's terminus area” -> Refer to Fig. 2g again here at end of sentence.

p4, second paragraph and the use of “glacier-wide”: “The historical elevation change record reveals roughly two decades of glacier-wide thinning from 1957-1978 followed by anomalous terminus thinning on Kennicott Glacier relative to the rest of the glacier (Fig. 2).” It is not fair to say glacier-wide thinning because you only have coverage over the lower parts of the glaciers. Please add the time period related to the terminus thinning -> post 1978? Please also see comment on Fig. 2 in relation to the use of “glacier-wide”.

p4, second paragraph: “the western side of the terminus has particularly thin ice (<30 m) and—given current thinning rates—will likely be ice-free in the next decade” -> would add here the current thinning rates to support the statement that this part will be ice-free within a decade. The thinning rate for 2012-2023 of -1.41 myr^{-1} was just mentioned and does not align with this statement. Suggestion: “and given current thinning rates in this area (around X-Y myr^{-1} ; Fig. 2e)”.

p4, second paragraph: “the glacier-wide spatially-distributed thinning pattern [...] magnitude of thinning increases over time (Fig. 2a-e)” -> Using “glacier-wide spatially-distributed pattern” is confusing, please reformulate. Also, please be specific instead of “over time”. Magnitude of thinning for Root increased until 2004-2012, it did not increase in the last period (Fig. 2f).

p4, second paragraph: “In turn, on Root Glacier, thinning rates at the terminus are similar to those up-glacier.” -> Merge with previous sentence to describe the pattern you are referring to, or with the next sentence: “In contrast to Kennicott Glacier, thinning rates on Root Glaciers are [...], suggesting that [...]” Also, it is not entirely clear what you refer to as the Root terminus. Last sentence: “glacier dynamics and debris-cover are critical drivers in understanding thinning patterns” -> “glacier dynamics and debris-cover are critical drivers behind thinning patterns” (glacier dynamics and debris-cover are drivers of patterns, not of the understanding?)

p4, last paragraph: “Since the turn of the century,” -> Please specify 21st century.

p4, last paragraph: “terminus thinning has remained steady while thinning in the ablation area above the terminus has roughly doubled” Is this referring to the Kennicott Glacier Terminus vs Kennicott Glacier (dark brown and light brown in Fig. 2f)? And roughly doubled compared to the previous two decades (1978-2004)? Please be more specific, suggest: “Since the turn of the 21 century (2004-2023), the thinning of the Kennicott Glacier terminus has remained steady while thinning in the Kennicott Glacier ablation area has roughly doubled, compared to the previous two decades (1978-2004; Fig. 2f).”

p4, last paragraph: “Although thinning is slightly greater for the years” -> “Although thinning in both areas is slightly greater for the period”

p4, last paragraph: “since 1957 and 1978 to the 21st century” -> this is a bit hard to read. Maybe just write “the multi-decadal trend of accelerated thinning from 1957 to 2023 (Fig.2a-e) is evidence..”

p4-5, last paragraph: “Relatively modest mass loss throughout the second half of the 20th century has been exacerbated by increasing temperatures that have caused Kennicott and Root Glaciers to lose a combined 180 ± 10 Mt yr⁻¹ from 2012-2023.” And even more in the period 2004-2012. Maybe state the mass loss for the combined period 2004-2023 here? “exacerbated by increasing temperatures” -> probably true, but could be other climatic variability also, e.g. related to precipitation amounts, so maybe be more general: “exacerbated by climate changes”?

p5. Anthropocene is not an accepted term, use time specific period.

IFSAR - define

According to figure 2g. A large part of the upper glacier is missing, I am not sure if the sentence ‘...a reasonable estimate of glacier-wide changes.’ is justified.

p5, Fig. 2, legend/colors: Nice figure. The hatches on the terminus of Kennicott in Fig. 1g does not match the hatches in the legend of 1f. The hatches in the legend matches the annual mass change in 1f from 1938 to 1957, but not the other time periods. This is confusing. The hatches/background colors used for the Area bars in 1g are also slightly different from the rest, please update. Also, consider updating the thickness of the lines for the Hugonnet data such that they match better with the lines shown in the figure.

p5, Fig. 2, caption and use of “glacier-wide”: “The area covered by historical DEMs post-1957 represents over 95% of mass loss and is thus a reasonable estimate of glacier-wide changes” -> The reasoning here is not clear and requires more explanation. Is it so that you compare your mass loss

estimates over the entire hatched area in Fig. 2g with glacier-wide mass losses estimated by Hugonnet et al. (2020) for 2000-2019 and find that your estimates correspond to 95% of this, and thereby assume that most of the mass loss occurs in your surveyed area, while the rest of the glaciers (accumulation areas) have limited mass changes. From Fig. 1f it looks like this may be the case for 2010-2020, where the mass losses 2012-2023 are relatively well aligned, but not for 2004-2012? Do you think this assumption holds over time? The assumption requires more explanation (from the caption it is not clear where the 95% mass loss estimate comes from), and would suggest adding it in the main text since it is very important for the statements in paragraph 2. Currently it is very confusing that glacier-wide changes are mentioned in this paragraph in relation to Fig. 2. Fig. 2 does not show glacier-wide changes and in paragraph 1 it is stated that changes are assessed across the various extents in Fig 2g. Please consider this comment in relation to comment on p. 14, Glacier outline digitization, paragraph 1, below.

p5, Fig. 2, caption: Caption header, add “mass changes”: “Multi-decadal mass changes on..”. The description “Kennicott Glacier ablation area” is not entirely clear because the terminus can also be considered a part of the ablation area. Consider updating the caption to a-e: “Kennicott Glacier ablation area (excluding the terminus, labeled as “Kennicott Glacier”)”. Add an explanation in the caption of what σ_{tot} in the disc refers to.

p5, Fig.2f, Hugonnet: This dataset is associated with relatively large uncertainties. Would add error bars or perhaps some sort of shaded area around the lines in Fig. 2f to indicate the uncertainty in the Hugonnet estimates. It might actually help you with the “glacier-wide”-argument. Note that studies have shown that Hugonnet estimates differ from repeat airborne LIDAR (e.g. Andreassen et al, 2023. DOI: <https://doi.org/10.1017/aog.2023.70>)

Glacier slowdown drives present-day retreat on debris-covered terminus:

p 5, first paragraph: “The availability and integration of glacier bed observations and long-term velocity changes provide unique observations of nearly six decades of dynamic changes on Kennicott and Root Glaciers.” Not clear how these datasets provide dynamic changes. Do you mean here that the availability of bed topography to derive ice thickness (that you use in the flux and stress calculations)? Perhaps (although a bit long): “Ice thickness distributions (1938-2023) derived from differencing DEMs and glacier bed topography, in combination with long-term velocity changes (1960-2017), provide a unique opportunity to assess ice flux divergence and gravitational driving stresses, revealing nearly six decades of dynamic changes on Kennicott and Root glacier.”

p. 5-6, first paragraph: “To our knowledge, we present the oldest spatially-distributed record of surface velocities on a mountain glacier.” Repetition from introduction, remove.

p6, paragraph 1: “Both velocity and driving stress have decreased with thinning, with the largest changes occurring in the latter half of the observational period.” Please be specific about the period and insert references to Fig 3 in this first sentence, and maybe also about the area considered. “the latter half of the observational period” is very unspecific as the “observational period” for velocities and driving stresses are different. Also, velocities are assessed from 1957/62 to 2017, but in the introduction where you specify the different datasets used there is no mention of this (reader has to go to Methods). Would suggest adding the time periods somewhere, including them in suggestion for p5, first paragraph, maybe also add in the sentence here: “Both velocity (1960-2017; Fig. 3a) and driving stress (1938-2023; Fig. 3e) have decreased with thinning of the terminus and lower ablation areas, with the largest changes occurring in the latter halves of the respective observational periods.”

p6, paragraph 1: Sentence starting with “The near-zero driving” is long and somewhat repetitive, consider shortening or splitting in two. Also, it seems from the sentence that the glacier is not experiencing sufficient gravitational force to drive ice flow in general?

p6, paragraph 1: “Above this region, a reduction in driving stress of ~30% (~30 kPa) is observed throughout the lowest 4 km of Kennicott Glacier” -> “above this region” in combination with “lowest 4km” makes it difficult to understand where you refer to, please remove “above this region” if the sentence refers to the lowest 4 km.

p6, paragraph 1: Explanation of the role of the shelf feature starting with “This reduction in driving stress” is difficult to follow. Also, there are three sentences after each other that contain some version of the formulation “the reduction in driving stress over the lowest 4 km”. Please consider if these explanations can be reformulated to be more clear and avoid repetition.

p6, paragraph 2: State in the beginning of the paragraph briefly how these quantities were derived so that the reader does not have to refer to the methods, e.g. by amending the second sentence to: “We partition thinning signals into ice flux divergence and climatic mass balance to assess the relative contributions of flux divergence and enhanced melt on thinning rates in different zones in the lower ablation areas on Kennicott and Root glaciers.”

p6, paragraph 2: Please provide some reasoning behind why you consider this to be the climatic mass balance and not the climatic-basal mass balance. I am confused by the notation $b_{\text{clim_dot}}$ in the Fig. 3 and the use of the same notation in both Eq 4 and 6. To my understanding Eq 6 is $b_{\text{clim_zone_dot}}$. Combining Eq 2-4: $b_{\text{clim_dot}} = H_{\text{dot}} * \rho_{\text{ice}}$, which is not in line with Eq 6. To our understanding, Eq 4 actually assumes ice flux divergence into the surveyed areas is zero, since you consider only part of the glacier and not a whole glacier. This should be mentioned in relation to Eq 4 and the notation should be cleaned up. Please see in relation to comments in Methods.

p6, paragraph 2: “slightly more positive” -> “slightly less negative”

p7, paragraph 3: “Root Glacier displays a similar but subtler pattern as Kennicott Glacier, where decreasing fluxes drive elevation changes despite relatively unchanging climatic mass balance.” Not clear in Fig. 3f which pattern you are referring to here, where on Kennicott?

p6 and 7, paragraphs 2 and 3: These paragraphs contain many numbers, please summarize the changes briefly at the end of each paragraph - what are the main drivers and how has this changed over time? Also, the comments on increased thinning on Kennicott glacier are somewhat redundant since this is already discussed in relation to Fig. 1, suggest to skip sentence starting with “Above the terminus..” and rather focus on the drivers of thinning.

p6 and 7, paragraphs 2 and 3, Fig 3f: In general it is very difficult to see the magnitudes of the changes in flux divergence vs mass balance in Fig. f., top row. A large part of the text is concerned with these differences, consider giving more space to Fig. 3f top row to make them more visible to the reader. The panels can be more rectangular, extending the y-axis, adding more ticks and perhaps some thin grid lines.

Fig. 3: Please specify which outline (year) is shown in Fig. 3a. Please specify for which extent the lowest point on the centerline refers to (i.e. what does the 0 point on the x-axis in b-e refer to?) The red color on the font is hard to read. Also, consider if switching the color of V_D and V_E gives better contrast on the blue? Could the flux gate cross sections in f be shown on the map in a? Fig. 3f -> consider balancing/harmonize the position of axis text Elevation (m) and Elevation change

Fig. 3, caption: "The product" -> please be specific: the 1960 velocity map. "Velocity magnitudes from 1960 and since 1990 for ITS_LIVE and Anderson et al. (2021) products are shown at five points on Kennicott and Root Glaciers." -> difficult sentence, suggest: "Velocity magnitudes from this study (1960), ITS_LIVE (1990 and 2004) and Anderson et al. (2021) (2017) are shown at five points on Kennicott and Root glaciers." Please explain in the caption what the whiskers on the elevation change plots in f are and note the difference between the time periods covered for Kennicott and Root.

Debris cover evolution leads to terminus wastage:

Could 'used' be used instead of 'leveraging'? Easier to understand for non native English speakers. P8. here you define modern data but as mentioned this term already used earlier, define if first time used.

p. 8: Avoid use "retreat of ice", can be confused with glacier retreat. Remove "further". "over time, showing an up-valley migration" -> "over time, with the up-valley migration". Both of these statements are related to what the orthophotos show.

p. 8: Sentence starting with "While the stagnant" is long and difficult to follow, please reformulate.

p. 8: "the symptoms of these changes are similarly evident over time." This is unclear, please reformulate and be specific. Is the last sentence "Around 7.5-10 km .. " an example of changes that are "similarly evident over time"? This is not clear.

p. 8: At the end of the section, could you briefly sum up the effect of the debris cover on thinning rates and the mechanisms behind, and perhaps how you expect this to affect the evolution of Kennicott in the near future?

Past glacier observations constrain future mass loss:

p. 8, first paragraph: "Leveraging historical data for calibrating glacier evolution models is a considerable advance compared to prior studies that only use more recent (2000-2019) glacier-wide mass balance data" -> in general, this is not a novelty but is commonly done for glaciers where such data are available. But perhaps the only time it is done for Kennicott and Root? This statement needs to be reformulated. Combine Hugonnet and 2000-2019 into a single parentheses as this refers to the same data: "(e.g. 2000-2019; Hugonnet et al., 2021).

It seems also to be a mix up global vs local studies. Hugonnet is a newer studie giving global estimates, however, many studies in the past have been done for local glaciers and mountain ranges.

p. 8, first paragraph: "Leveraging historical data"/"Calibration with historical data ensures" -> Please be more specific, 2000-2019 is also a historical period, "long-term historical data"?

p. 8, first paragraph: Please specify what period historical data/calibration refers to (which datasets) in the same manner as is done with "modern".

p. 8, first paragraph: "the model calibrated with" -> please explain briefly what has been done and specify which model is being used (PyGEM), this is currently not done until the methods. For example, this is nicely done in the first sentence of the second paragraph - provide a similar overview in the first paragraph. Suggest to split the sentence starting with "Using monthly air temperature .."

which is currently a bit hard to read, merge with short introduction to PyGEM and continue with the calibration results.

p. 8, first paragraph: Please briefly explain the experiment with the “historical calibration” vs “modern calibration”. Consider in relation to the above comment.

p. 8, first paragraph: “fits the known glacier mass in the 21st century,” -> These are also glacier mass estimates, but from other sources, please reformulate.

p. 8/9, second paragraph: “a notable reduction in mass loss in the first half of the 21st century compared to the model calibrated with modern data, followed by accelerated mass loss after 2050 (Fig. 4c, d).” -> notable reduction, better to use “shows lower mass loss” as reduction in mass loss sounds like the mass losses decelerate. Also, it is a bit confusing that the last part of the sentence is related to the “historical” model when the “modern” model is just mentioned: make sure that it is clear what is comparison and what is description.

p. 9, second paragraph: “While the decrease in the rate of mass loss” -> this sentence is very unclear, what is comparison and what is description? Difficult to follow this in the figure. Please reformulate. And Fig. e and f does not show rate of mass loss, but climatic mass balance in mm w.e (volumetric).

p. 9, second paragraph: “By 2100, GCM spread plays a large role in model projections as the remaining glacier mass falls within GCM spread”. This is unclear, what does remaining glacier mass falls within GCM spread mean? Please reformulate. It is not clear from this sentence if you are comparing the differences between models in relation to the uncertainty associated with GCMs.

p. 9, second paragraph: “on average, 22% less mass loss” -> not clear, referring to cumulative mass loss for mean of projections?

p. 9 second paragraph: “Regardless of whether the historically-calibrated model predicts increased or decreased mass by 2100, both Kennicott and Root Glaciers show a clear acceleration in the rate of mass loss at the end of the 21st century,” Where is the increased mass? Seems like mass balance is negative throughout the 21st century? Please reformulate this sentence, the statement is not clear.

p. 9 second paragraph: “implying a more dramatic disequilibrium between the glacier and the climate than previously forecasted”. Cannot follow this. Assuming that previously forecasted refers to the “modern calibration”. Is the reduction in mass loss (fig 3e, f) with the modern calibration in the 21st century implying that the modern model shows a stronger disequilibrium earlier in the century, with earlier glacier retreat compensating for mass losses?

p. 9, second paragraph: Regarding the discussion of parameter values, could you report these in the methods? Would be helpful to see what the differences are. Could the different calibration procedures used in the two experiments affect the results, not only the different calibration data? Also, the introductory paragraph 1 focuses on capturing glacier dynamics and does not mention that climate sensitivity is captured through parameter values, which may be more representative when using long-term records for calibration. It would be beneficial to the explanation if this is mentioned before and then relate it to the explanation of the findings.

Additional Fig. of surface profiles: Please add a figure in the methods showing the modelled glacier profile with the two calibration experiments from 1940-2020, together with the known surface height in 1957, 1978, 2004 and 2012 in areas where this is available. It would help to validate the glacier geometry evolution and show that the dynamic state of the glacier is better captured using the historical calibration, which you hypothesise in the first paragraph but do not comment on later.

Limitations: Please comment on the limitations of the modelling here or in the methods. For example, both models use constant parameter values over the historical and future time periods, while melt-factors have been shown by several studies to not be constant in time and not necessarily representative of future conditions, e.g. Huss et al. (2009) and Ismail et al. (2023). Do you think the parameters calibrated with historical or modern data are more representative for the future climate? The historical model clearly captures climate sensitivity better than the modern model in the historical period, but could the historical model underestimate future melt since parameter values are more representative of the historical climate rather than the future one? Please comment on this in the main text or in methods.

Debris cover: Since debris cover is seemingly important for the ablation area of Kennicott glacier, an explanation of how PyGEM handles debris cover needs to be included in the methods, and possibly commented on as part of limitations or discussion. Orthophotos show an increasingly large area of debris cover that has significant impacts on melt and glacier dynamics, do you expect these mechanisms to be represented in the model? If not, how does that affect past and future projections?

Fig. 4: The consensus mass estimates shown in Fig. 4a and b. Which densities were used to convert Farinotti et al 2019 volume to mass? Same as used in Hugonnet 2021? It would be useful to add error bars to these estimates based on uncertainty in the volume-to-mass conversion. I would specify in the caption how the consensus estimate for 2020 is derived: “adjusted to 2020 using mass change rates from Hugonnet et al., 2021”. Add gridlines to the plots in Fig. 4 so it is easier to follow the discussion on different time periods.

Fig. 4 caption: “using ERA5 reanalysis data”, the model is equally relevant here: “modelled using the Python Global Glacier Model with ERA5 climate reanalysis data”. “Results show all individual and the mean of optimal parameter sets that align with multi-decadal mass balance” -> this is not clear, please see in relation to comment on this in Methods (modelled using mean of parameter values from n=.. parameter sets, or mean of modelled mass balance using n=.. parameter sets). Please specify what “Historical” and “Modern” calibration refers to in the caption. Specify (Hugonnet et al. 2021) after modern data from 2000-2019. Also, if understood correctly from the methods, the historical calibration also uses the modern data, please specify this.

Summary, p.10-11:

p.10: “In summary, ~85 years of observations show a clear shift from a likely near-equilibrium state prior to 1957, to ongoing and accelerating mass loss throughout the late 20th and into the 21st century.” -> The latter part is shown by modelling, not by observations. Please rewrite. And be more specific than observations: geodetic mass balance? Since this is a summary, mention again the names of the glaciers.

p. 11: “Projections leveraging” -> Add “projections from a glacier evolution model”

p. 11: “and project this acceleration to continue through much of the 21st century—decades later than previous studies suggest” -> This is a bit unclear: “show a later onset of mass loss acceleration than previous studies suggest”?

p. 11: “other large glaciers (>50 km²) in the Wrangell Mountains” -> Reference?

p. 11, last paragraph: State you contribution in this paragraph, e.g. “We have shown that such imagery can provide unprecedented insights into long-term glacier mass losses and their mechanisms

and provide valuable data that improves projections of glacier change in both the near and distant future.”

Methods:

p. 11, Methods: Consider adding a section header before sections “Historical imagery” and “Historical DEM generation”, in the same manner as the “Additional DEMs” header. Here, it could also be useful for the reader to navigate if the DEM years are specified in the headers, e.g. “Additional DEMs (1938, 2004, 2012, 2023)”, or perhaps in subsections, e.g. “Washburn DEM (1938)”.

p. 12, Methods, Historical DEM generation, paragraph 4: Reference to Fig. 1, add subplot numbers c, d, e.

p. 12, Methods, Historical DEM generation, paragraph 4: The statement “which constrained the final DEMs to the ablation areas of Kennicott and Root Glaciers” is somewhat confusing since the DEM extents in Fig. 1 c and e also partly cover the accumulation area. Do you mean that reliable elevation change estimates could only be determined from the ablation areas (Fig. 2 a-e)?

L379. You need a larger buffer in the terminus area probably?

p. 13, Methods, Aster DEM: “adjusted long-term (1978-2012) thinning rates” -> Consider specifying that these thinning rates are derived from differencing 1978 and 2012 DEMs since the 2012 DEM is not presented until the next paragraph.

p. 13, Methods, National Park Service DEM: “from 1,250 images collected” -> specify how these images were collected.

p. 14, Glacier outline digitization, paragraph 1: “Outlines were produced for [...] two time periods (historical and modern).” What does historical and modern refer to? In the rest of the manuscript, “modern” and “historical” refer to datasets used for model calibration. Please provide the years of the outlines here instead. Same for paragraph 2 of the same section where it is referred to “modern extent”. be specific on years with outlines and which outlines used.

p. 14, Glacier outline digitization, paragraph 1: “Accumulation areas were excluded during outline digitization due to poor image contrast on snow and a lack of suitable ground-control points which encumbered DEM generation in these regions: above the uppermost icefalls on Root Glacier, the Gates and LaChapelle tributaries, and the edge of the aerial photograph extent on the main branch of Kennicott Glacier (Fig. 1a).” Consider moving the description of the specific areas to the first time this is mentioned, under “Historical DEM generation”, last paragraph.

p. 14, Glacier outline digitization, paragraph 1: “Nonetheless, our digitized outlines (Fig. 2g) represents over 95% of all mass loss on Kennicott and Root Glaciers from 2000-2019 (Hugonnet et al., 2021); we thus assume that mass loss calculations are representative of the entire area of Kennicott and Root Glaciers.” This statement/reasoning is not clear (please see comment on Fig. 2).

p. 14 Glacier outline digitization, paragraph 1: “The terminus boundary outline includes Kennicott Glacier below its junction with Root Glacier, aligning in extent with the 1938 DEM coverage (Figs. 1a, 2g).” Which outline is referred to here? Do you mean that all outlines cover this part of the terminus? In general the descriptions on outlines are unclear.

p. 14, Glacier outline digitization, paragraph 2: “Calculations requiring glacier area depend on the considered elevation change time period to determine the boundary used to delineate valid pixels for the volume change (Eqn 1) and the area (e.g., Eqns 3 and 4).” This sentence is difficult to understand, consider reformulating.

p. 15, Volume, elevation and mass change: "Volume change rate ..", "Mass change rate .." and "Mean elevation change .." Suggest to use "The" in front, e.g. "The volume change rate .." to make this easier to read.

L401-405: it will impact results, show a comparison.

L408-412. Be specific, why not use outlines for each year and period? Do you not have them? How does it impact the results?

p. 15, Volume, elevation, and mass change: "A density of 900 kg m⁻³ was preferred to the commonly-used geodetic volume-to-mass conversion 850 kg m⁻³ (Huss 2013) because our study is constricted to ablation areas over long temporal spans, and thus not affected by snow or firn" -> Can be more clear. The term density conversion factor is better to use. This statement should be related to the choice of using ρ_{ice} in Eqn 2, rather than the choice of value for ρ_{ice} . Consider "The density of ice in Eqn 2 was used for volume-to-mass conversion since we consider mass changes over ablation areas where mass loss is limited to the melt of ice (rather than the combination of ice, firn and snow)", or something similar.

p. 15, Volume, elevation, and mass change: "where the area A aligns with V the temporal span." -> Suggest to write out the description of V_{dot} here to increase readability: "where the area A aligns with the time span of the volume change rate V_{dot} ". Also, consider some additional explanation of which area you are referring to here before Eqn 3, e.g. "The mean elevation change rate within a given glacier boundary is the volume change rate divided by the area within the glacier boundary?"

p. 15, Volume, elevation, and mass change: "Mass balance is simply the specific mass change rate: b_{clim_dot} " -> Why do you refer to the climatic mass balance rate here? Here, the considered mass balance rate is perhaps the climatic-basal mass balance rate because mass changes include both surface, internal and basal processes, unless you assume basal processes to be negligible (in which case that should be stated). A short explanation of the mass balance components that are accounted for should be added here, in relation to the choice of notation.

p. 15, Volume, elevation and mass change: The notation M_{dot} is used for the mass change rate. This is strictly the geodetic mass balance in mass units and should be denoted by B_{dot} . It is not clear why the geodetic mass balance rate becomes the climatic mass balance in Eqn 4, see above comment.

p. 15-16, Ice thickness and bed topography: This section describes ice thickness measurements from 2024 and derivation of the bedrock topography. Sections "Flux divergence and climatic mass balance" and "Driving stress" describe the use of ice thickness to compute flux divergence and driving stress and from the first sentences on p. 18 under "Driving stress". It is so that ice thickness is computed for each of the DEM years by subtracting the bedrock topography described in section "Ice thickness and bed topography" from each surface DEM, providing several ice thickness datasets from which ice thickness is retrieved in sections "Flux divergence and climatic mass balance" and "Driving stress"? Suggest adding a short explanation at the end of the "Ice thickness and bed topography" section describing this procedure and reiterating which years (DEMs) this is done for. This explanation could be before the "Flux divergence and climatic mass balance" section since ice thickness is used here.

Is the velocity representative of the period 1957-1962?

p. 17, Flux divergence and climatic mass balance: Again, it is not clear why the elevation changes are assumed to represent the climatic mass balance (and not the climatic-basal mass balance). In relation to Eqn 6, shouldn't this be the climatic mass balance of the flux zone, $b_{clim,zone_dot}$?

p. 17, Flux divergence and climatic mass balance: “This flux-gate method emulates previous literature (Anderson et al. 2021b) but is robust to specific flux gate placement, thus reducing bias and accurately capturing uncertainty in climatic mass balance results as it ensures that localized uncertainties in velocity, ice thickness, and surface elevation change are consistently accounted for regardless of particular flux gate positions.” Consider splitting this sentence in two, it is difficult to follow.

P18. Miss values in uncertainty part.

p. 18, Driving stress: Equation 7 is referenced differently from previous equations (Eqn 7 vs. 7).

p. 18, Driving stress and Extended Data Fig. 2: In Extended Data Fig. 2 it would be helpful to add the glacier outline from 1957/1978 in the subplots or at least specify which extents the change in driving stress is computed for in subplots f-k. I assume that the driving stress/change in driving stress is not plotted for areas with no ice? To my understanding the driving stress is computed over each glacier extent, but I assume that the change in driving stress is plotted for the minimum extent of the two years in f-k?

p. 19, Uncertainty assessments, Volume, elevation and mass change uncertainty: Eqn 10: Here b_{dot} is used for specific mass change rate instead of b_{clim_dot} . Should be consistent with Eqn. 4 and 6.

p. 19, Glacier evolution modelling: Please provide a short introduction to how PyGEM models mass balance and dynamics.

p. 19, Glacier evolution modelling, Historical glacier evolution reconstruction: “The historical reconstruction requires an estimate of the glacier mass and area from 1940 to initialize the simulation.” Shouldn’t this rather be “estimate of the glacier geometry (ice thickness and extent)” rather than “mass and area”?

p. 19, Glacier evolution modelling, Historical glacier evolution reconstruction: The explanations in this first paragraph are not entirely clear. Why was the consensus ice thickness estimate from Farinotti used? Due to the complete coverage over the glaciers? This should be clarified. Consider using “consensus ice thickness” rather than “ice volume”, since volume suggests a quantity integrated over the glacier. Please be more specific in relation to elevation change rates missing in accumulation areas and remind the reader: “Since elevation change rates from 1957-2012 could not be reliably determined in the accumulation areas of Kennicott and Root Glaciers, we estimated elevation-changes from the 1957 U.S. ...” How are elevation changes in these areas determined from the 1957 map, could you explain? It is also not clear how the statement “Glacier-wide mass change from 1940-1957 was assumed to be negligible given no surface elevation change at the terminus” relates to the rest of the paragraph, was the 2000 ice thickness adjusted to 1957 and then no changes were assumed between 1940 and 1957? In that case it is a bit confusing that it is stated previously in the paragraph that the elevation change rates from 1957-2012 are used to adjust back to 1940. Also, is this last statement justified? Surface elevation changes at the terminus were positive 1938-1957 (same magnitudes, but opposite of 1957-1978), wouldn’t you expect positive glacier-wide mass balance? Needs more justification and explanation of how this fits in with the reconstruction of the 1940s glacier.

p. 20, Glacier evolution modelling, Historical glacier evolution reconstruction: Regarding the calibration data used (ice thickness change from DEM differencing 1957-1978, 1978-2004, 2000-2019). Why were these periods chosen? For example, why not use 2004-2012 and 2012-2023? I assume the 2000-2019 data is Hugonnet et al. (2019), please specify.

p. 20, Glacier evolution modelling, Historical glacier evolution reconstruction: Last sentence in paragraph 2: not clear what mean refers to here. Is it the mean of each parameter value from the n=89 and n=22 combinations, or are you referring to the mean of the modelled mass change as the best estimate of historical glacier evolution?

p. 20, Glacier evolution modelling, Future glacier evolution: “each model run had 48 individual realizations” -> it is not clear which parameter values are used here. It seems from “using the calibrated sets of model parameters” that the model is run 89x48 and 22x48 times per glacier, or are future simulations run with mean parameter values (ref previous comment)?

Extended data:

Extended Fig. 1: The caption to Extended Data Fig. 1 states: “The ice thickness from this study is derived using the 2012 IFSAR DEM and corrected with elevation change from Hugonnet et al. (2021) to adjust (c) to the year 2000 to match (d) and (e).” This is confusing. From section “Flux divergence and climatic mass balance” and “Driving stress” it seems that ice thickness is computed using the bed topography and each surface DEM. From the caption it seems that only ice thickness in 2012 is estimated? Assuming this is only shown as an example/for comparison, this should be stated. The figure caption should state clearly why it is adjusted to 2000 (not clear that this is the representative year for the other estimates). Reference to a comparison between these products in the main or method text seems to be missing. Would suggest adding a sentence about the comparison in this figure somewhere in the methods where you state the purpose and interpretation of the comparison.

Extended Data Fig. 2: Please specify in the caption the outlines plotted in c and d - 1957?

Extended Data Table 1: Would suggest changing the order of the columns to “Product, Year, Source, Data” to increase readability (helps reader locate product). Also, suggest adding a new column with a letter code to differentiate between the products derived or used in the study, instead of using different fonts.

Citations

Check the order of citations. These are not in alphabetical order or ordered by year, e.g. two places in p2, Introduction, paragraph 3.

A reference seems also to be missing from the suppl material.

References

Huss, M., M. Funk, and A. Ohmura (2009), Strong Alpine glacier melt in the 1940s due to enhanced solar radiation, *Geophys. Res. Lett.*, 36, L23501, doi:10.1029/2009GL040789.

Ismail, M. F., Bogacki, W., Disse, M., Schäfer, M., and Kirschbauer, L.(2023) Estimating degree-day factors of snow based on energy flux components, *The Cryosphere*, 17, 211–231, <https://doi.org/10.5194/tc-17-211-2023>.